# Noninvasive electromagnetic source imaging of spatiotemporally distributed epileptogenic brain sources

Abbas Sohrabpour [1,2], Zhengxiang Cai [1], Shuai Ye[1], Benjamin Brinkmann [3], Gregory Worrell[3] & Bin He [1✉]

Brain networks are spatiotemporal phenomena that dynamically vary over time. Functional imaging approaches strive to noninvasively estimate these underlying processes. Here, we propose a novel source imaging approach that uses high-density EEG recordings to map brain networks. This approach objectively addresses the long-standing limitations of conventional source imaging techniques, namely, difficulty in objectively estimating the spatial extent, as well as the temporal evolution of underlying brain sources. We validate our approach by directly comparing source imaging results with the intracranial EEG (iEEG) findings and surgical resection outcomes in a cohort of 36 patients with focal epilepsy. To this end, we analyzed a total of 1,027 spikes and 86 seizures. We demonstrate the capability of our approach in imaging both the location and spatial extent of brain networks from noninvasive electrophysiological measurements, specifically for ictal and interictal brain networks. Our approach is a powerful tool for noninvasively investigating large-scale dynamic brain networks.

[1] Department of Biomedical Engineering, Carnegie Mellon University, Pittsburgh, PA, USA. [2] Department of Biomedical Engineering, University of Minnesota, Minnesota, MN, USA. [3] Department of Neurology, Mayo Clinic, Rochester, MN, USA. ✉email: bhe1@andrew.cmu.edu

Normal and pathological brain states, observed via behavior, are produced through the underlying function and dysfunction, of an individual's brain networks. Understanding the properties of these complicated spatiotemporal networks, is of the utmost importance for advancing our foundational knowledge of the human brain, as well as introducing and advancing novel clinical tools aimed at managing brain disorders and improving quality of life. Throughout the last century, noninvasive functional neuroimaging has, in one way or another, played a significant role in advancing our knowledge of human brain networks[1]. Given brain's organization, i.e. that spatially localized areas specialize in particular functions (functional segregation) and that the communication among these areas results in observed behavior (functional integration), the brain activity on large scales, can be modeled as a globally distributed spatiotemporal network[2]. Estimation of these spatiotemporally distributed networks, from noninvasive measurements, provides useful information about which brain areas are activated (or deactivated) during particular brain functions (normal or pathological), how extended these regions are, how their activity develops over time, and how multiple areas interact with each other during normal or pathological brain states. In other words, spatiotemporal imaging of brain activity provides a quantitative reconstruction of the underlying networks involved in a task or brain state. For instance, comparing different brain areas involved in externally evoked responses in autistic and healthy patients, can be used to identify potential areas involved in autistic dysfunction[3]. Employing appropriate functional imaging methodologies that provide meaningful and robust insight for a given study, is important.

An ideal functional imaging modality for studying large-scale dynamic human brain networks should have at least three characteristics, namely, high spatial resolution, high temporal resolution, and a wide spatial coverage. Most existing imaging modalities do not satisfy all the aforementioned criteria, simultaneously. Although, invasive brain monitoring approaches and recording devices typically provide detailed information with high spatial and/or temporal resolution, they normally lack spatial coverage and pose risks to subjects, due to their invasive nature. Invasive recording techniques are more commonly involved in clinical cases regarding dysfunction, but are rarely, if ever, involved in healthy subject recordings, as their introduction poses unwarranted risks, and perturbs the neural system by simply being present, making them unsuitable for general human imaging applications. Despite the need for a noninvasive imaging technology with high spatiotemporal resolution, no such technology currently exists, in a single modality. The availability of such an imaging technology would have a profound impact on brain research and the clinical management of a variety of brain disorders[4].

Of the many noninvasive modalities available, they typically factor into either electrophysiological, i.e. direct, or metabolic, i.e. indirect, in nature. Modalities such as functional MRI (fMRI) and functional near infrared spectroscopy fall into the metabolic category and measure the neural brain activity indirectly through the hemodynamic response (blood oxygenation and volume flow), and, consequently, have inherently low temporal resolution. On the other hand, the two major noninvasive electrical recording techniques, electroencephalography (EEG) and magnetoencephalography (MEG), have the capability of mapping brain activity with high temporal resolution[5,6]. Unfortunately, the spatial resolution of traditional scalp EEG/MEG measurements is limited, due to low signal-to-noise ratio (SNR), inadequate sensor numbers, and, most importantly, the smearing effect caused by volume conduction. Over the years, various engineering approaches have been designed to improve the spatial resolution

of these modalities, with the most promising advancements coming from a set of approaches called the electrophysiological source imaging (ESI) techniques[1,7–10].

In general, ESI involves modeling brain electrical activity as a series of equivalent current source distributions[9]. Typically, these approaches model the system as a series of linear problems, where reconstruction relies on varying the magnitude of a distribution of current dipoles located within pre-allocated grids that span the cortex[11]. While source imaging algorithms have been successfully developed and applied to localize the underlying brain activity, these conventional approaches suffer from a common shortcoming: they are incapable of objectively determining the source's extent on their own[12–16]. This is a serious shortcoming, as the ability to distinguish between the relevant brain activity and background activity (defined here as unrelated brain activity) is a necessary requirement for studying both normal and pathological brain functions. Different brain regions are functionally specialized to perform particular functions[2], thus, determining the spatial extent of these regions is indispensable, and highly desirable when imaging brain networks.

Although recent attempts have been made to address this issue, by estimating extended sources[17–23], these methods still require the application of post-hoc thresholds in order to distinguish between the true underlying brain activity and irrelevant background activity. These thresholds may lead to biases which might work within datasets with known ground-truth, but can produce highly variable results in novel circumstances or exploratory analyses. Some of these recently proposed methods generate relatively extended sources which better distinguish background activity from desired brain activity (compared to conventional approaches)[19], which in return, enables them to use more objective thresholding schemes such as Otsu's thresholding technique[24] to discard background activity. However, the dependence of these algorithms on post-hoc addendums suggests that the modeling framework needs improvement.

Our approach, which we have termed the fast spatiotemporal iteratively reweighted edge sparsity (FAST-IRES) technique, can fill this gap by objectively estimating extended sources and their time-course of activity as they vary over time (Fig. 1). FAST-IRES requires little to no input from the clinician or researcher, when determining the spatial extent of underlying sources, completely avoiding the issue of potentially biased subjective thresholds. Our method estimates focally extended sources by starting with a relaxed estimation and iteratively refining the estimate by penalizing source locations with smaller amplitude compared to source locations with higher amplitude. In order to do so, we formulate our algorithm as a series of convex optimization problems that can be solved easily and efficiently. Additionally, by computing internodal connectivity among the estimated nodes of activity, we can determine major functional nodes, leading to a more interpretable picture of the underlying dynamics of brain networks. Although numerous problems could benefit from the application of our proposed technique, human-epilepsy stands apart as a uniquely challenging option that provides a rigorous testbed due to the availability of ground truth measurements. Additionally, and equally importantly, is the fact that clinicians could benefit from an objective form of noninvasive source estimation.

Epilepsy poses an important challenge due to its wide spread, affecting about 0.8–1% of the world population[25,26]. Uncontrolled seizures, other than their obvious debilitating consequences, are life-threatening and increase the risks of sudden death in epilepsy patients[27]. Additionally, studies have shown that the longer a patient suffers from untreated seizures, the less likely it is for subsequent treatments to be successful[28]. While typical pharmacological treatments work in the majority of patients, roughly,

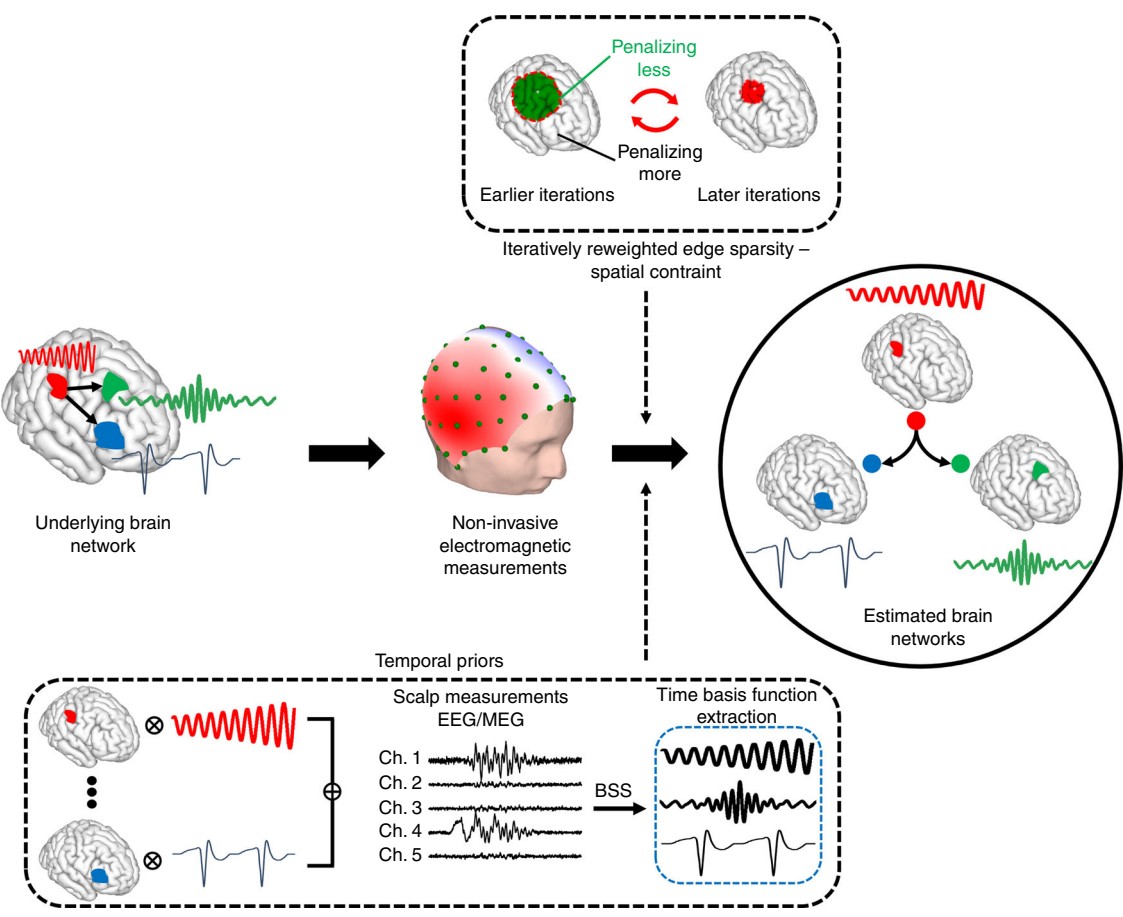

**Fig. 1 The concept of the proposed spatiotemporal source imaging approach.** Brain networks are modeled as focally extended sources that vary over time. The net-effect of these dynamics are recorded in EEG/MEG. Blind source separation (BSS) techniques applied to these measurements can delineate these underlying dynamics and serve as a temporal prior in the imaging algorithm. Spatial constraints that enforce the edge sparsity, i.e. clear distinction of activity and background noise, can be ensured by applying an iterative reweighting scheme in a data-driven manner to guarantee focally extended sources. Combining these data-driven priors into our imaging module, we can estimate underlying brain networks; the nodes and internodal connectivity (links) of these networks. Nodes are spatially extended regions in the brain and not focal points. Our spatiotemporal source imaging approach considers the functional segregation, i.e. spatially coherent regions in the brain specialized for specific functions, and functional integration, i.e. inter-regional communication and connectivity, of different brain regions.

one-third of epilepsy patients suffer from drug-resistant epilepsy[29,30], leaving surgery as a viable option for treatment[31]. Current imaging modalities cannot successfully determine the epileptogenic zone (EZ), which is defined as the minimum amount of brain tissue that needs to be removed in order to stop seizures, due to lack of spatial specificity[25,32,33].

For the reasons mentioned above, our approach is uniquely situated for aiding those with epilepsy. To both validate our approach, and to provide much needed insight into the network dynamics involved in epilepsy, we performed a comprehensive study of focal epilepsy patients using high-density EEG recordings. In this study, we rigorously and quantitatively validated our noninvasive source imaging results by comparing them with invasive intracranial electrophysiological recordings and surgical resection outcomes in the same patients.

By analyzing epilepsy networks with our proposed FAST-IRES framework, we demonstrate that the EZ can be determined objectively and noninvasively, with high precision, from scalp high-density EEG recordings. We also compare the performance of ESI when applied to both ictal recordings and interictal spikes to determine which feature might be clinically more relevant in determining the EZ. Note that, while developed and validated in an epilepsy framework, our proposed approach is by no means limited to epilepsy source imaging and is an efficient and general

source imaging approach from electromagnetic recordings, including EEG and MEG.

## Results

**Monte carlo simulation of extended sources.** The basic idea behind the proposed approach, is to estimate the spatiotemporal distributions of the underlying sources simultaneously, rather than estimating their underlying spatial distribution at every time-point. This allows for the development of more efficient and accurate algorithms and more precise modeling of the underlying brain processes. Underlying sources are assumed to be spatially focal, i.e. limited to local regions of the brain not being too focal and point-like, while not encompassing extremely large areas of the cortex. We refer to sources with this trait as *focally extended sources*. This property, most likely, is not specific to epileptic sources[34] and has been suggested in other large-scale phenomena, where extended cortical areas have to be synchronously activated to produce detectable signals at scalp-level measurements such as EEG and MEG[35,36]. Note that fine micro-scale organizations of the brain activity, are not perceivable in EEG/MEG or any surface recordings[35], and we are making no claims in this work to be able to recover such sources. The idea of a focally extended source is to model large-scale brain signals and organizations that are typically recorded in surface measurements such as EEG and MEG.

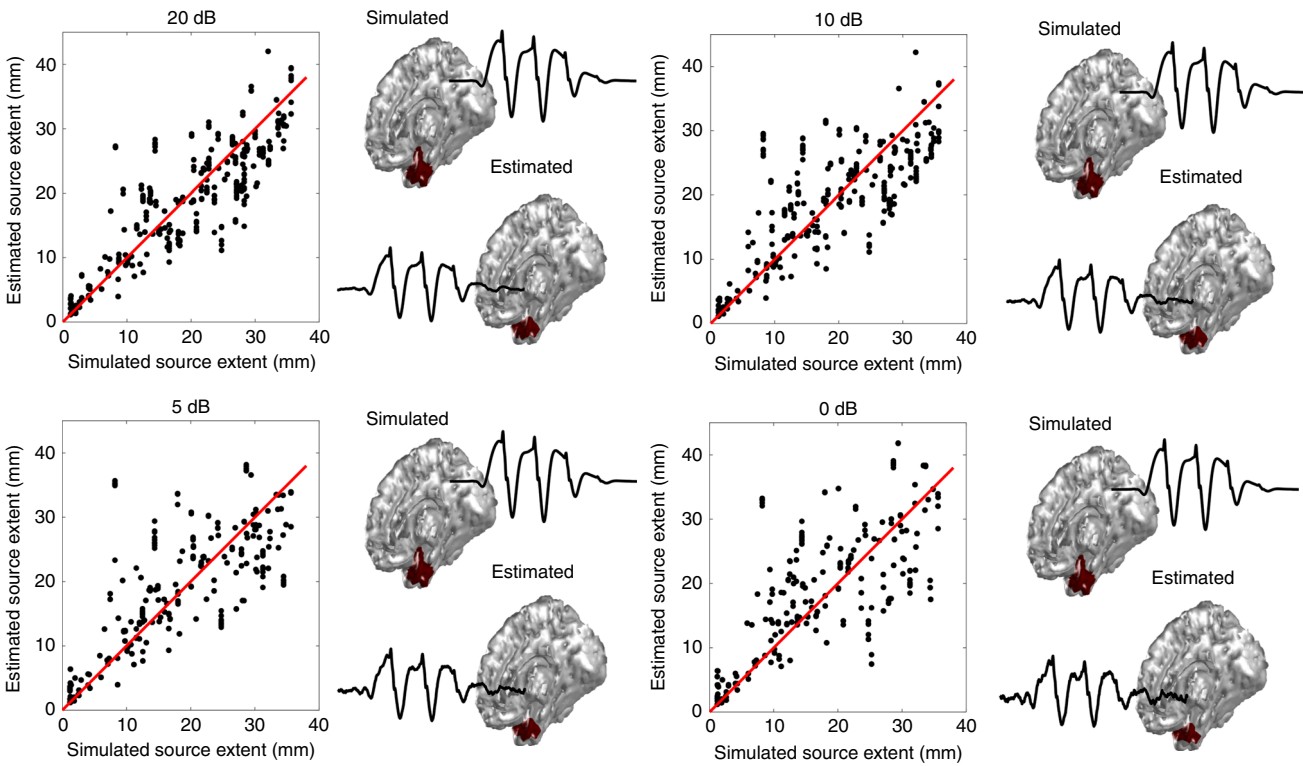

**Fig. 2 Monte Carlo simulation results of extent estimation.** The results of estimated extent from our FAST-IRES algorithm is plotted against the simulated sources' extent for four different SNR conditions. An example of a mesio-temporal source is also presented for all the SNR conditions, along its estimated time-course of activity. The red line in the four plots is the identity line and is provided for reference. Source data are provided as a Source Data file.

A key feature of our work is to capture this fundamental property of large-scale brain organization, mathematically, and to develop computational tools to solve these mathematical models. Each focally extended spatial source has its own unique time-course of activity corresponding to its change in amplitude over time. As the activity of all sources are linearly super-imposed to form the scalp potentials recorded by EEG, the source activities can be delineated from scalp measurements with blind source separation (BSS) techniques[37]. Additionally, spatial constraints, such as focality and the presence of distinct boundaries between activity and background noise, can be achieved by enforcing sparsity via iterative approaches, ultimately promoting focally extended sources. These spatial constraints and temporal priors, depicted in Fig. 1, can be implemented within a series of convex optimization problems which are easy and efficient to solve.

In order to thoroughly investigate the performance of our framework in reliably determining underlying sources' extent, we performed a Monte Carlo simulation (Fig. 2). In this simulation random locations on the cortex were selected and an extended source was simulated at that location, and the EEG signals generated by these source configurations were calculated. Figure 2 presents the results of our simulation along a typical example of a mesio-temporal source that has been localized in all SNR conditions. The estimated time-course of activity for this source matches the simulated time-course, nicely. A significantly high Pearson's correlation value of 0.88 was found for these results (for more details refer to Supplementary Notes 1 and 2, Supplementary Tables 1–3, and Supplementary Movie 1).

**Overview of clinical data analysis results**. With all modern forms of machine learning, or structured decomposition, discerning relevant feature vectors is paramount to quantitative assessment and interpretability of results. In our case, in order to

image epileptic activity, we first need to extract the known epileptic features from EEG recordings, namely, the interictal spikes and seizures. We studied 36 patients suffering from focal epilepsy, who underwent surgery as a treatment for their intractable seizures. As shown previously[38–40], high-density EEG recordings are important for achieving accurate solutions during ESI imaging. We developed a novel high-density (76 channels) EEG technique, which allowed us to record continuous EEG over multiple days in patients undergoing pre-operational monitoring, to capture both interictal events and seizures. We analyzed this novel data to estimate the EZ and to quantitatively compare our noninvasive ESI results to invasive iEEG findings and surgical resection outcomes in the same patients. Seizures were determined and marked by experienced epileptologists while spikes were extracted, for each individual patient, by the research team. Detailed information about the patients can be found in Supplementary Tables 4 and 5. We analyzed both seizures and interictal spikes, contrasting the quality and the concordance of the estimated solutions with clinical findings, for each feature (Fig. 3). Component analysis was used to reject noise as well as determine the time basis function (TBF) for the underlying sources (refer to Supplementary Methods for detail). The criteria for selecting TBF components was different for spikes and seizures, as explained later in the Methods section (Fig. 3, top panel). Once the temporal priors, i.e. TBF, were calculated, the source imaging process was performed for spikes and seizures (Fig. 3, middle panel) and the estimates from each feature were compared to surgically resected volume and/or intracranial electrodes denoted as seizure onset zone (SOZ) by epileptologists (Fig. 3, bottom panel). Detailed information about the implementation and mathematical derivation of our method is presented in the Supplementary Note 1 and Supplementary Methods, specifically in Supplementary Tables 1–3, Supplementary Figs. 1–3, and Supplementary Movie 1.

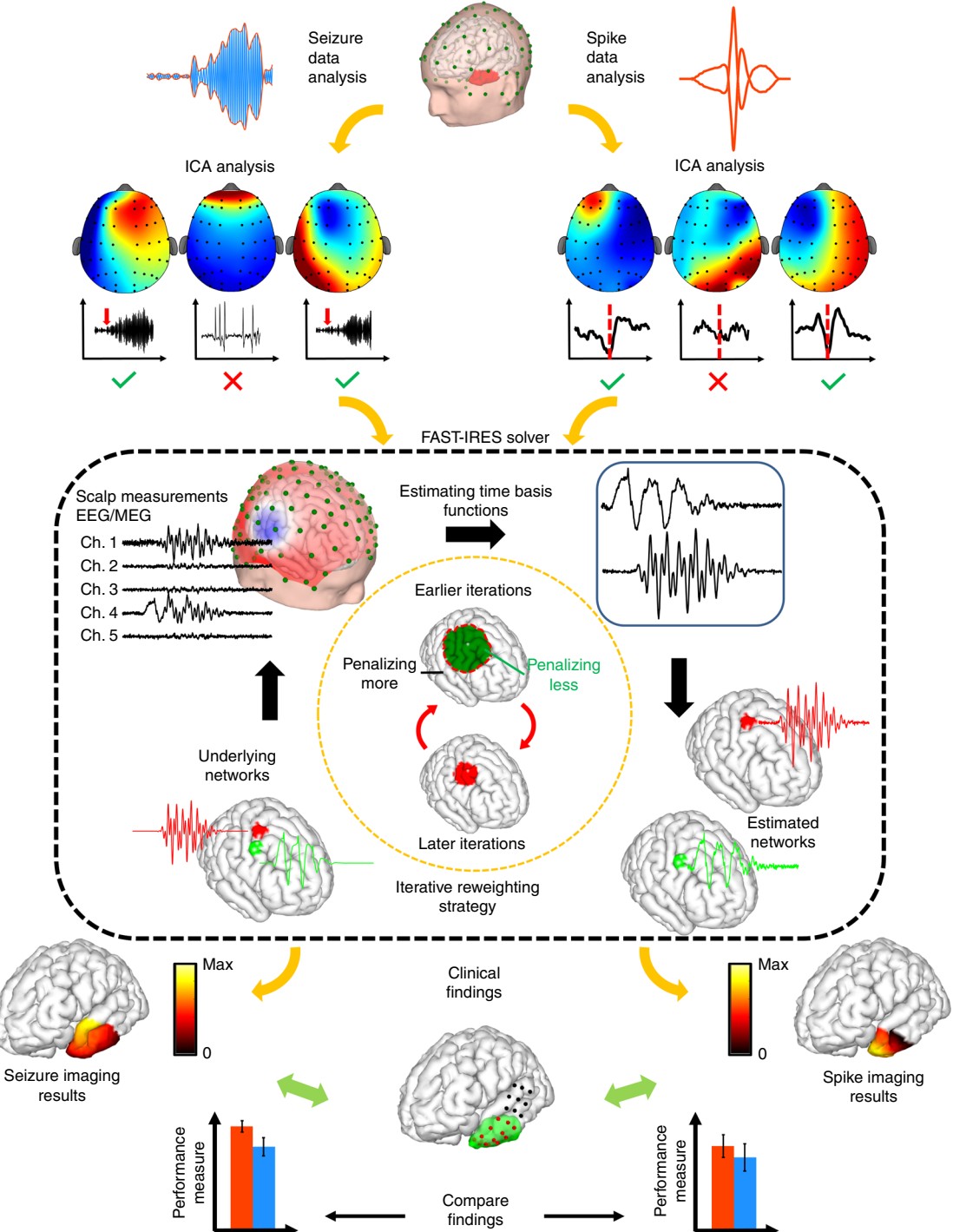

**Fig. 3 Overall study design.** In this figure the analysis pipeline is depicted. (Top) The main two arms of the study show how interictal spikes and seizures are extracted from EEG recordings, denoised, their time basis function determined, and input into the FAST-IRES solver. (Middle) The proposed FAST-IRES source imaging approach takes the spatial extent and focality of brain sources into account. The output of the algorithm is a spatiotemporal distribution of underlying brain sources, from which the epileptogenic zone (EZ) is extracted and compared to clinical findings, such as resection volume and seizure onset zone determined from intracranial EEG. (Bottom) Finally, the performance of epilepsy features, i.e. estimating the EZ by imaging interictal activity and ictal activity with FAST-IRES, is evaluated by comparing the estimated EZ to clinical findings.

**Spike imaging**. Spikes are generally believed to arise from the irritative zone, which is not the same as the SOZ[41]. While the irritative zone and SOZ are straightforward to define and easy to assess, the EZ has proven elusive. Clinically, the EZ is defined as the minimum amount of tissue which, if resected, results in seizure-freedom. This definition is abstract and difficult to assess. The closest measurable feature to the EZ is most likely the

surgical resection volume in patients who become seizure-free after surgery. The surgical resection volume can be thought of as the pseudo-EZ, or the true EZ, for the purposes of comparison in this paper. While prior literature demonstrates unequivocally the concordance of spike-imaging results and clinically determined EZs[42–44], the relationship between spike-imaging findings and imaging results from ictal recordings, remains uncertain. We have

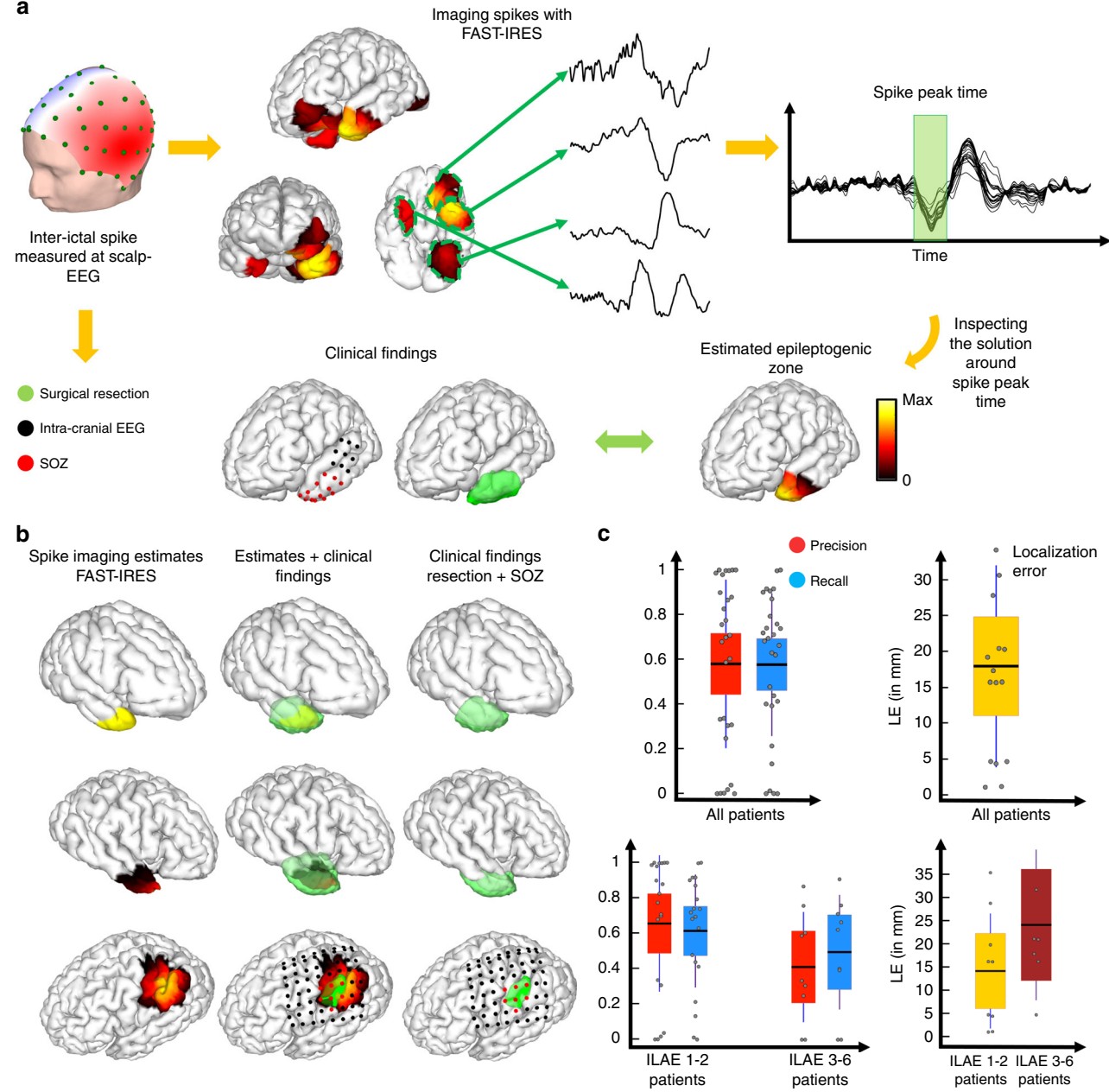

**Fig. 4 Spike imaging overview and results. a** The output of FAST-IRES is a spatiotemporal distribution where spatial distributions correspond to a time-course of activity. To determine the epileptogenic tissue, the source signals' energy is calculated around the spike peak-time and compared to clinical findings for validation. **b** Examples of spike-imaging results along the clinical findings in the same patients. **c** Quantitative results of spike-imaging results for all patients (top) and separated based on surgical outcome (bottom). Note that while the color scheme distinguishes precision vs. recall in these patients, for localization error it is used to denote seizure-free from non-seizure-free patient groups (bottom). Each gray circle corresponds to individual patient's data. The horizontal black bar indicates the mean, the color bars indicate the 95% confidence interval for the mean and the dark vertical bars indicate the standard deviation. To compute precision ($n = 29$, $0.58 \pm 0.38$) and recall ($n = 29$, $0.58 \pm 0.32$), 29 data points were available in total ($n = 29$). The same analysis for seizure-free patients yielded higher precision ($n = 20$, $0.65 \pm 0.39$) and recall ($n = 20$, $0.61 \pm 0.32$) compared to the precision ($n = 9$, $0.41 \pm 0.31$) and recall ($n = 9$, $0.49 \pm 0.32$) in the non-seizure-free group. The localization error ($n = 16$, $18.1 \pm 14.08$ in mm) was calculated from the data of 16 patients ($n = 16$). The localization error ($n = 9$, $13.9 \pm 11.96$ in mm) in seizure-free patients was smaller than the same value ($n = 7$, $23.5 \pm 15.7$ in mm) in non-seizure-free patients. Reported values are (mean ± standard deviation). Source data are provided as a Source Data file.

investigated this relation in the present study in order to determine which, if either, of these features can provide more relevant information about the EZ.

Spike waveforms were selected as the 2-s symmetric windows surrounding the spike peaks in each patients' EEG recordings prior to surgery. These waveforms were then averaged, and

source imaged (Fig. 4a). More details on component analysis and selection are presented in the Methods section.

After the averaged spikes and their corresponding TBFs are input to the FAST-IRES solver, the output is the distribution of the brain electrical activity that varies over the 2-s interval. The results are averaged across a 40 ms window around the spike-

peak time to obtain a single source distribution, which is compared to the clinical findings in each patient (Fig. 4a). A few examples of spike-imaging results are presented in Fig. 4b, overlapped with the corresponding clinical findings for each case. On average a precision and recall of 0.6 is observed for spike-imaging results (Fig. 4c). This roughly translates to spike-imaging estimates of the EZ having a 60% overlap with the resected tissue. This indicates that the EZ can be estimated with spike-imaging results, specifically when considering that surgical resections are typically large enough to ensure that all of the epileptogenic tissue is removed (hence they over-estimate the true EZ). The localization error, which indicates the distance of the estimated EZ to SOZ electrodes, is on average about 18 mm for spike-imaging results (compared to 6 mm in seizure imaging, Fig. 5c). This falls in line with the fact that spikes and seizures do not originate from the exact same regions in the brain, hence conveying different information about the underlying epilepsy circuits.

No statistically significant differences were observed between the seizure-free and non-seizure-free groups (Fig. 4c). The implications of this observation are discussed later in the "Ictal imaging vs. spike imaging" section. Supplementary Movie 2 presents the estimated spike source on the cortex throughout a 0.5 s window around the spike peak time in one patient. An advantage of this computational approach is that spike generation and propagation can be visualized clearly, and a wealth of information can be extracted by observing the video.

**Ictal imaging**. While interictal spikes have been analyzed over the years, seizures (ictal recordings) are typically left unanalyzed. This is partially because ictal recordings are very noisy and often include a large number of undesirable artifacts, such as eye movement, blinking, muscle and movement artifacts. Furthermore, most source imaging algorithms were developed to localize and image sources corresponding to a spatial map or spatiotemporal distributions of a relatively short segment. As a result, imaging raw ictal recordings is extremely difficult. Visually inspecting raw EEG recordings does not provide accurate information about the origin and extent of the underlying seizure-generating tissue. Such impediments in the quantitative study of ictal recordings is a major motivation for developing ictal-imaging techniques. Such quantitative approaches can provide a wealth of clinically relevant information about the origin and extent of seizure generating networks.

As before, defining the relevant features within ictal activity is foundational for its successful analysis. Historically, strong rhythmic signals have been attributed as the main characteristic of ictal activity with pronounced changes in frequency content coinciding with seizure onset. Additionally, it has been shown that the lobe from which the seizures arise is correlated with the frequency of ictal oscillations[45]. In light of these findings, many methods have been proposed which attempt to classify seizures by their origin in the brain, i.e. mesial vs. neo-cortical temporal seizures. Unfortunately, these gross attempts at classification are far too broad to accurately localize ictal activity for each individual patient. In an attempt to add nuance to their estimates, some studies have further attempted to localize the dominant seizure frequency of the ictal signal at and around the seizure onset[46–49]. The main shortcoming of these methods is that in order to remove noisy artifacts and improve the SNR, the signal becomes much too smooth (by averaging for instance). Additionally, some of the earlier studies used dipole fitting to localize SOZ, ignoring all information regarding the spatial extent of the sources, given that seizures propagate fast and are not generated at extremely focal sources.

Recently two works from Ding et al.[50] and Yang et al.[40], proposed a component-based analysis to remove artifacts and improve SNR in ictal signals. Yang et al.[40] proposed that by performing independent component analysis (ICA), major artifacts can be rejected and components showing a similar trend of spectral and temporal evolution to the raw ictal EEG recordings can be extracted and analyzed as seizure-relevant signals. They showed significantly improved results and also demonstrated that their proposed method has considerable advantage compared to methods proposed previously[40]. The TBF estimation and ICA denoising procedure, employed in this work are based on the positive results obtained in these prior works.

Using our proposed algorithm, we imaged the first few seconds (3–5 s) of seizures and estimated the distribution of electrical activity in the brain over this time interval, subsequently band-pass filtering our solutions at the dominant ictal rhythm. To find the dominant ictal rhythm in each patient, we compared the first 5 s spectrogram of the ictal signal after seizure onset to the spectrogram of the 5 s interval prior to ictal onset, and selected the frequency band where the two spectra were most different (Fig. 5a).

Examples of applying this method to ictal recordings to estimate the EZ are presented alongside the clinical findings in these patients (Fig. 5b). The precision and recall for our estimated EZs are both high, about 0.75, demonstrating that our inferred sources neither underestimate nor overestimate the true under-lying EZ (Fig. 5c). This is due to the fact that precision and recall are normalized metrics describing the overlap of source and ground truth, i.e. surgically resected volume, in relation to estimated source size or ground truth. As a result, if a source underestimates the extent of the surgical volume but falls perfectly within it, only one of these values will be high. Conversely, if the estimate overestimates the surgical volume size and completely contains the volume, still only one of these values will be high. Only, when the estimated source and surgical volume are roughly the same size and overlap nicely, will both of these values be high. Additionally, the localization error, in patients who underwent iEEG recordings, is around 5 mm, which is already near the iEEG electrode resolution limit (defined as half of the smallest distance between iEEG electrodes, which is typically 10 mm in our study). When the results are plotted for both the seizure-free and non-seizure-free groups of patients (international league against epilepsy (ILAE) 1 and 2 vs. ILAE 3–6), no statistically significant difference is observed (Fig. 5c).

In 87% of patients who underwent invasive iEEG recordings, deep electrodes were implanted to determine if deeper structures were involved in epileptogenesis or not, which resulted in deep structures such as the hippocampus being resected (refer to Supplementary Table 4 for more details). Nevertheless, the small localization errors indicate that using our proposed method, even deep sources can be delineated noninvasively. This finding is supported from other studies as well[5,6].

Two videos, showing the estimated epileptic activity of two patients, are presented as examples in Supplementary Movies 3 and 4, demonstrating seizure dynamics and propagation. As observed in Supplementary Movie 3, a virtual cut can provide a view of deeper brain structures (such as the mesio-temporal view presented in this video). Additionally, we calculated the correlation of estimated time-courses of activity, within the lateral portion of the anterior temporal lobe and the deep structures near the hippocampus and para-hippocampal cortex, and compared them to intracranial traces recorded in this patient. Our results show a correlational value of 0.61 between estimated time-courses and iEEG recordings in these two regions (Supplementary Movie 3 depicts these results as well). Videos

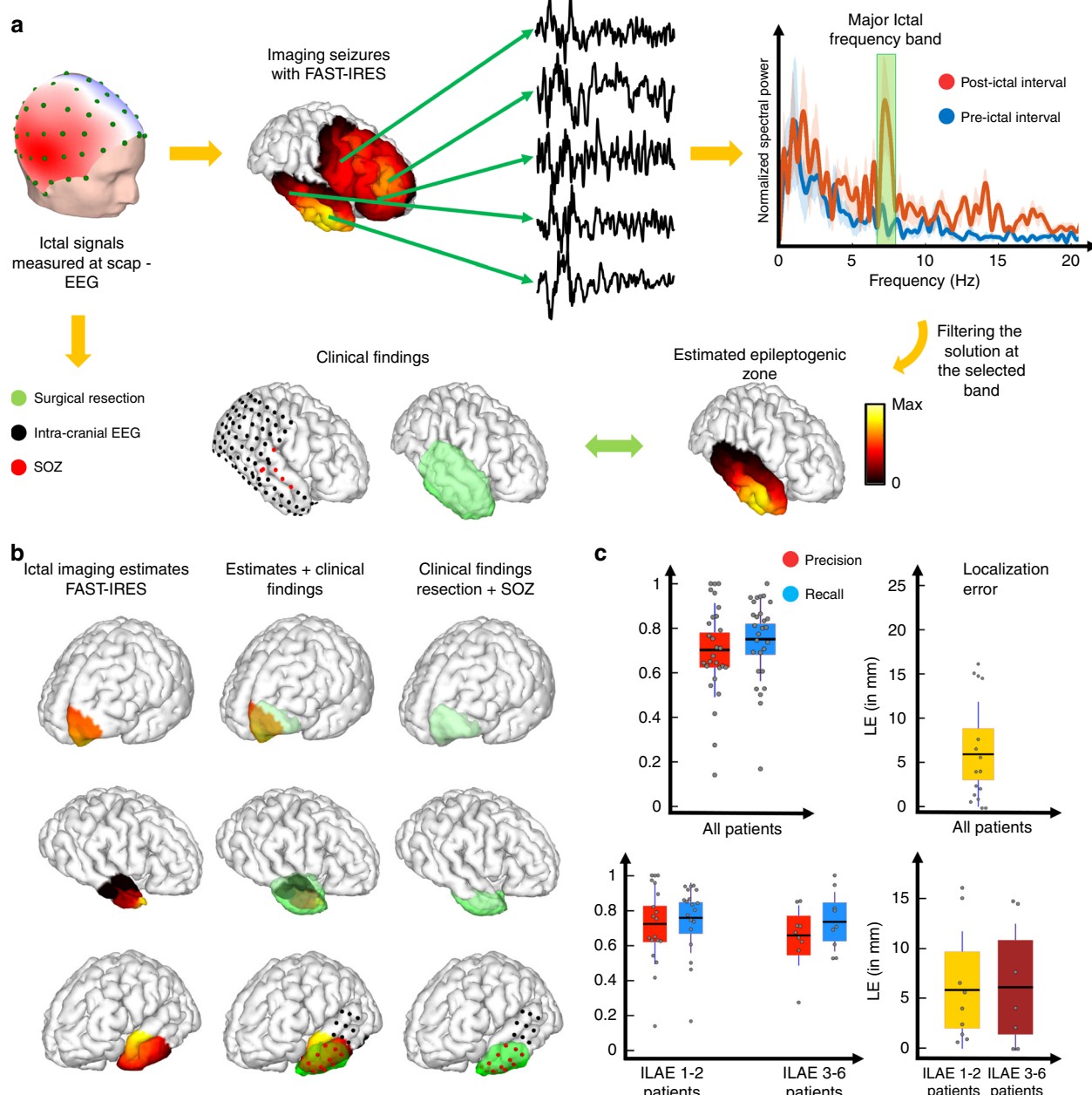

**Fig. 5 Ictal-imaging overview and results. a** The output of FAST-IRES is a spatiotemporal distribution where spatial distributions correspond to a time-course of activity. To determine the epileptogenic tissue, the dominant ictal frequency is determined and the source signal distribution is filtered at this frequency; then the energy of the source signals is calculated at this frequency and for a 1-s interval at seizure onset, to determine the epileptogenic tissue. **b** Examples of ictal-imaging results along the clinical findings in the same patients. **c** Quantitative results of ictal-imaging results for all patients (top) and separated based on surgical outcome (bottom). Note that while the color scheme distinguishes precision vs. recall in these patients, for localization error, it is used to denote seizure-free from non-seizure-free patient groups (bottom). Each gray circle corresponds to individual patient's data. The horizontal black bar indicates the mean, the color bars indicate the 95% confidence interval for the mean and the dark vertical bars indicate the standard deviation. To compute precision (0.7 ± 0.21) and recall (0.75 ± 0.19), 28 data points were available in total ($n = 28$). The same analysis for seizure-free patients yielded higher precision ($n = 19$, 0.72 ± 0.23) and recall ($n = 19$, 0.76 ± 0.2) compared to the precision ($n = 9$, 0.66 ± 0.17) and recall ($n = 9$, 0.74 ± 0.17) in the non-seizure-free group. The localization error ($n = 16$, 6 ± 5.8 in mm) was calculated from the data of 16 patients ($n = 16$). The localization error ($n = 9$, 5.9 ± 5.8 in mm) in seizure-free patients was smaller than the same value ($n = 7$, 6.1 ± 6.3 in mm) in non-seizure-free patients. Reported values are (mean ± standard deviation). Source data are provided as a Source Data file.

like these can help physicians to better interpret and analyze seizure dynamics and propagation as well as pertinent information about its onset. This virtual inspection of deeper brain structures is one of the major benefits of our noninvasive ESI approach that is beneficial for studying epileptic networks. It can also be employed to study brain networks in general.

**Connectivity imaging.** FAST-IRES provides a spatiotemporal estimate of underlying brain sources. This implies that in addition to the location and spatial extent of underlying brain activities, the time-course of such activities can be estimated, as well. Subsequent network or connectivity analyses can be performed based on FAST-IRES results. We have outlined and

performed a connectivity analysis based on our FAST-IRES results which bears novel ideas of how to potentially implement such analyses in tandem with the proposed framework. Refer to Supplementary Note 3, Supplementary Figs. 4–7, and Supplementary Table 6 for more details.

**Ictal imaging vs. spike imaging.** Significant differences were observed between spike and ictal imaging results that deserve further investigation. However, before discussing these issues, we should briefly introduce two additional measures that we have employed to quantify our results (mathematical details in the Methods section). Both of our normalized ratios, i.e., precision and recall, should ideally be 1, so in order to make comparison easier and more intuitive, we use the geometric and harmonic means of precision and recall. These means would be small and close to zero if either of these values is close to zero, only attaining high values when both measures (precision and recall) are close to 1 (the ideal case).

The results of seizure imaging are statistically significantly superior to those of spike imaging in terms of both localization error, geometric mean and harmonic mean (Fig. 6a, b, detailed statistical results and values are presented in the Supplementary Table 7 and Supplementary Fig. 8, geometric mean: 0.7, harmonic mean: 0.68). Upon further investigation we found that this difference (between spike and seizure imaging results) only existed in patients who were not seizure free post operation (Fig. 6a, b). This can most likely be attributed to the fact that some patients had multiple types of spikes, where one type could be contralateral to the affected side (Fig. 6c), while seizures consistently occurred on the same side as clinical findings. In other words, when a spike type, i.e. cluster of spikes having the same spatiotemporal characteristics, is discordant with clinical findings, for instance when it is contralateral to the resection site, the precision and recall decrease and localization errors increase, resulting in worse performance metrics for spike analysis results compared to seizure analysis results. In our study, only 18% of seizure-free patients demonstrated inconsistent spike types, while this value was close to 60% for the non-seizure-free cohort. This suggests that when inconsistent spike types are present, merely relying on spike imaging results can be misleading and the electrophysiological assessment of seizures must be considered. A detailed discussion of this point is reserved for the Discussion section.

We further investigated the difference observed between spike and seizure analysis, by comparing consistent spikes to seizures in our dataset. Consistent spikes were defined as spikes that were ipsilateral to the resection side. Our results indicated that no significant differences could be perceived between consistent spikes and seizures. We also investigated the use of a boot-strapping technique for spike averaging on the quality of results and found no improvement. Results and statistics are presented in Supplementary Notes 4 and 5, Supplementary Figs. 9 and 10 and Supplementary Tables 8 and 9. Note that even when seizures propagated to the contralateral side (which is common in seizures) both our post-hoc frequency analysis and our connectivity analysis could still pinpoint the correct SOZ (Fig. 6d).

Additionally, we have also looked at the difference between our connectivity and ictal-imaging results (the driving node is selected as the estimated EZ in the connectivity imaging) and found no statistically significant difference among the two groups. This indicates that while the connectivity imaging approach provides a more analytical framework, it did not improve the ictal imaging results significantly. The statistical tests and values are reported in Supplementary Table 6 and have been additionally, depicted in Supplementary Fig. 7.

**Investigating extent estimation in empirical data.** In order to directly compare our results with clinical findings, we calculated the size of our estimated epileptogenic tissue using seizure imaging analysis and spike imaging analysis and plotted these values against resection size. Significant correlations are observed. Please refer to Supplementary Note 6, and Supplementary Fig. 11 for more details.

## Discussion

The presented results firmly demonstrate that the location and extent of underlying brain networks can be determined in a precise and objective manner, noninvasively. Our proposed approach significantly extends the library of existing imaging/monitoring approaches, with the potential to be the main source of information in some cases. While, our method, was motivated by and tested in the framework of epilepsy imaging, we would like to emphasize that it has the capability of studying other normal and pathological brain functions, as underlying networks constituting these brain states are spatiotemporal processes that vary over relatively short time intervals[1].

In addition to our spatiotemporal approach towards brain's spatiotemporal processes, the mathematical assumptions of our method are based on physiological and physical principles which are fundamental, and therefore, general. This generality of assumptions ensures the aptness of our approach to study various large-scale network phenomena in the brain. The main assumptions of our approach pertain to the temporal and spatial aspects of brain networks. Firstly, we assumed that underlying sources' time-courses of activity can be delineated on scalp recordings, which was subsequently used to form our temporal priors. This is due to the linearity of the volume conduction model as necessitated by Maxwell's equations. Secondly, we assumed brain sources to be focally extended sources, which enabled us to impose spatial constraints on solutions. This assumption is backed up by physiological recordings and studies, suggesting that, a considerable amount of cortical tissue is activated during different brain functions and states; it is such large-scale phenomena that EEG/MEG sensors record during epilepsy-related brain activity (micro-scaled brain organization are not considered)[35,36]. Our proposed framework, which is constituted on these fundamental properties is therefore applicable to study other normal and pathological large-scale brain networks that are, inherently, distributed spatiotemporal processes.

It is apt to be reminded that our proposed approach like any model, is based on a theoretical framework and some model assumptions. For instance, we chose an $L_1$ optimization framework to formulate the inverse problem while another method might formulate the problem within a Bayesian framework of analysis. These choices are inevitable and are not the kind of subjectivity that must be avoided, as they are the product of a model-driven effort. However, based on the choices and assumptions different models make, subsequent steps of the analyses they must perform and the characteristics of the results they obtain, will differ. FAST-IRES has three significant advantages compared to existing approaches. Firstly, it can, in itself and through the process of iterative re-weighting, distinguish signal and background without the need of any post-hoc analysis. Secondly, its hyper-parameters are easy to tune and track. For instance, the hyper-parameter $\alpha$ which balances between sparsity and edge-sparsity of the solution can be tuned by the L-curve approach. This easy and intuitive method of tuning a hyper-parameter in itself is a major advantage of our proposed method compared to some more mathematically based assumptions, such as independence of variables in some Bayesian methods or hyper-parameters decided by trial and error or expert opinion. Thirdly, our approach does not pre-parcellate the brain[19], in a data-driven

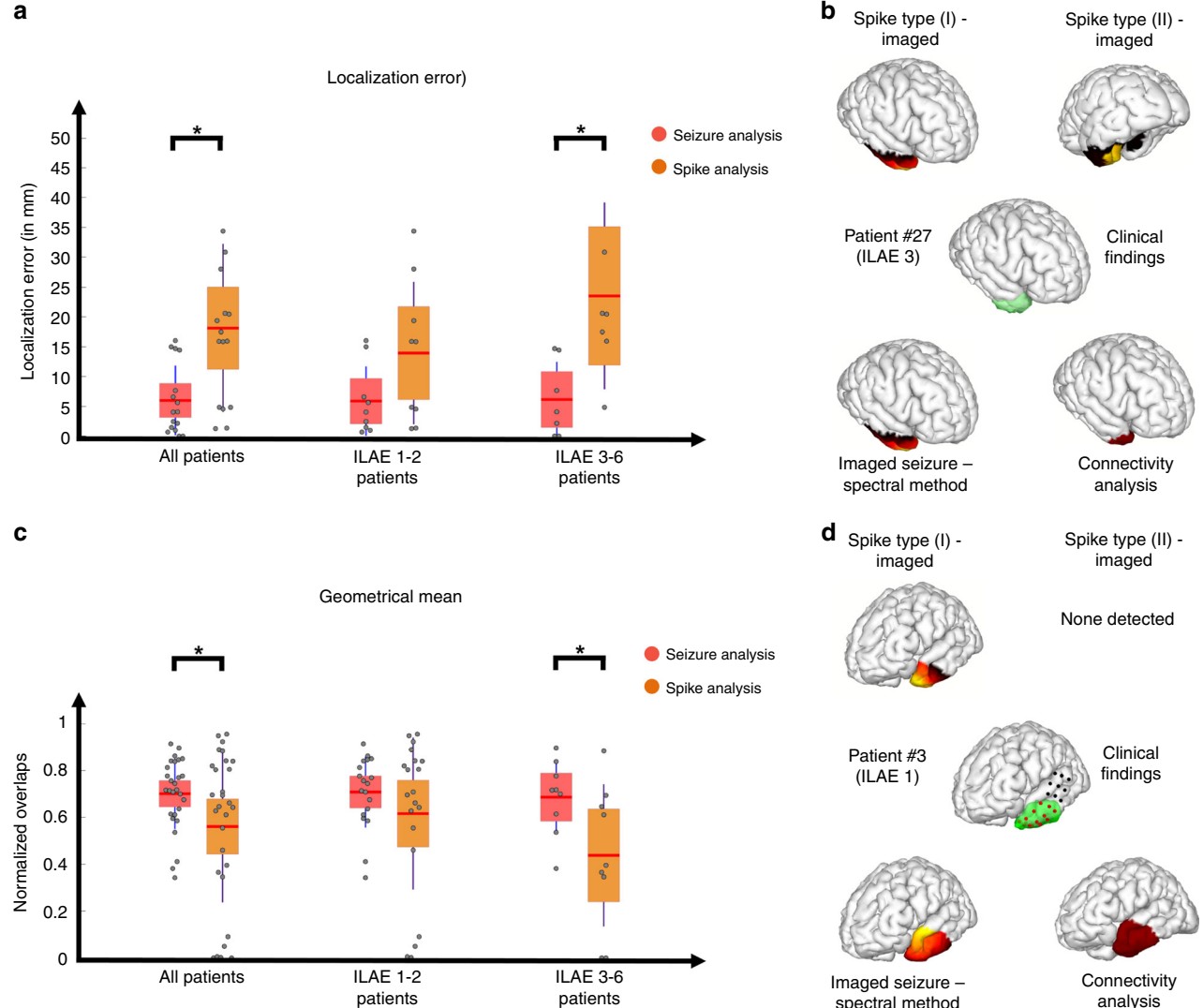

**Fig. 6 Comparing spike imaging and ictal imaging results. a** Localization error for spike and ictal imaging results are presented. Each gray circle corresponds to individual patient's data. The red bar indicates the mean, the color bars indicate the 95% confidence interval for the mean and the dark bars indicate the standard deviation. The localization error in all patients ($n = 16$) showed a significant difference (*$p < 0.05$) between the ictal and interictal imaging results (one-sided permutation test, $p = 0.0015$). Same results were observed for non-seizure-free patients ($n = 7$, one-sided permutation test, $p = 0.004$), while seizure-free patients did not show such a strong trend ($n = 9$, one-sided permutation test, $p = 0.0476$). **b** Geometric mean for spike and ictal imaging results; organization is same as in **a**. The geometric mean in all patients ($n = 29$ for spike analysis and $n = 28$ for ictal analysis) showed a significant difference (*$p < 0.05$) between the ictal and interictal imaging results (one-sided permutation test, $p = 0.0195$). Same results were observed for non-seizure-free patients ($n = 9$, one-sided permutation test, $p = 0.0236$), while seizure-free patients did not show such a trend ($n = 20$ for spike analysis and $n = 19$ for ictal analysis, one-sided permutation test, $p = 0.14$). **c** Spike and ictal imaging results in one patient along the clinical findings in this patient. This particular patient had two major spike types and belonged to the non-seizure-free group. **d** Spike and ictal imaging results in one patient along the clinical findings in this patient. This particular patient had one major spike type and belonged to the seizure-free group. Source data are provided as a Source Data file. Color scales and definitions for **c**, **d** are the same as Fig. 5.

or pre-determined manner, to estimate spatial extent (refer to Methods for more technical details).

Our results were rigorously validated by imaging epilepsy networks and were highly concordant with invasive clinical findings in these patients. Successful application of this technique may significantly impact clinical practice, as a wealth of information about underlying networks can be provided to physicians. For instance, providing a video of where seizure onset is located and how it propagates to other brain regions is highly helpful in determining the EZ, planning for surgery, or forming a hypothesis for iEEG electrode placement[51], or even placing responsive neuro-stimulation (RNS) stimulators. Furthermore, since the video-EEG monitoring of patients who are candidates of surgery

(to surgically remove the drug-resistant epileptogenic tissue), is a common pre-surgical medical routine[52], the high-density EEG approach we have introduced for ictal ESI imaging can be used to provide noninvasive, accurate, and reliable information about the epilepsy networks, without disrupting the clinical workflow. Actually, such high-density EEG pre-operational monitoring has recently been implemented in Mayo Clinic, Rochester, and is provided as an option to each patient for their routine epilepsy management and monitoring.

Our technique should not be seen in the light of competition but rather collaboration with existing imaging techniques to improve the management of epilepsy and maximize treatment benefits for epilepsy patients. It is not uncommon for epilepsy patients to

undergo invasive iEEG studies before or during the main surgery to determine where the seizures are arising from, as iEEG studies are considered the gold standard in determining the SOZ[53]. However, before undergoing such procedures it must first be determined if such studies are beneficial or not; additionally, a clear hypothesis as to the location and extent of the regions involved in seizure generation and propagation is required[51]. Currently, information from patient and family history, semiology, structural images like MRI, and EEG studies are considered to form this hypothesis, directing the placement of electrodes and surgical decisions. Our approach can offer a wealth of assistance in determining the SOZ and EZ noninvasively by providing reliable information about possible brain regions involved in epilepsy generation and propagation as well as the spatial extent of these sources. This information can be used to guide the placement of intracranial and stereo-EEG (sEEG) electrodes or even devices that use electrical stimulation to stop seizures, such as the RNS device.

Invasive measures come with increased risks of infection, complications and costs[54,55] and with the advent of more minimally invasive approaches to treat epilepsy, such as laser interstitial thermal ablation (LITT) therapy[56] and the RNS therapy, there is an unmet need to advance noninvasive imaging tools. These noninvasive approaches, e.g. FAST-IRES, can provide enough information to move epilepsy treatments away from invasive approaches towards less invasive and even potentially noninvasive monitoring.

An important observation in our results was that seizure imaging provided more accurate estimates of the EZ, compared to spike imaging. The reason for this difference in performance was that spikes were more likely to be inconsistent; meaning that patients could have multiple types of spikes, where some types might not be consistent with clinical findings (for example, two different spikes could be on opposite sides). This phenomenon of inconsistent spikes was more prevalent in patients who did not become seizure-free post-surgery (who were scored as ILAE 3–6, refer to Methods for definition) compared to seizure-free patients (ILAE 1 and 2 individuals). For instance, in our dataset, 5 patients out of the 22 seizure-free patients had multiple spike types of which 4 were spatially inconsistent spikes (spikes not arising from the same side or lobe, or apart more than 2 cm) while 9 out of the 12 non-seizure-free patients had multiple spike types, of which 7 were spatially inconsistent. In other words, only 20% (4 in 22) of seizure-free patients had multiple inconsistent spike types compared to about 60% (7 in 12) for non-seizure-free patients (in our dataset). This is possibly why we observed a markedly significant difference between ILAE3–6 patients' spike and seizure analysis while no such difference was observed among ILAE 1 and 2 patients in the two groups. Additionally, our post-hoc analysis of consistent spikes (spikes that were ipsilateral to the resection side) also supports this hypothesis, as consistent spikes bore no statistically significant difference with ictal imaging results (refer to Supplementary Note 4, Supplementary Fig. 8, and Supplementary Table 9). Thus, it is fair to conclude, that inconsistent spikes are ground for further investigation, and as a result seizure imaging must be performed in these patients to unequivocally determine the affected side and the EZ. This has been observed in the clinical literature; if patients have bitemporal activation, in temporal lobe epilepsy cases for instance, it is more likely to expect unfavorable results and chances of seizure-freedom are reduced significantly[57]. However, our data suggest that if the patient has consistent spikes, spike imaging might provide accurate information about the EZ on its own, without foreseeable benefits from seizure imaging. This indicates that spike imaging is a useful technique for imaging epileptogenic tissue but in cases of multiple inconsistent spike types, other relevant clinical information such as seizures, need to be considered and analyzed carefully.

No statistically significant difference was observed between the seizure-free patients and non-seizure-free patients (in seizure imaging analysis) in terms of performance (in estimating the EZ), even though a reduced performance was observed for ILAE 3–6 patients compared to ILAE 1 and 2 patients (although, the geometric mean of the two groups was different in spike imaging with a $p$-value of 0.08, this, we believe, is due to the spatially inconsistent spike types). The majority of our non-seizure-free patients are ILAE 3 and 4 patients who saw marked seizure reduction after surgery, but due to non-significant differences (compared to seizure-free patients), it is impossible to comment on the efficiency of resected volume in these patients (whether resection size was large enough in non-seizure-free patients or not). In this aspect, our algorithm has not been able to provide additional insight to physicians, as the physicians were confident enough about the EZ in these patients to suggest and perform surgery.

Furthermore, analysis of variance (ANOVA) for our results and different sub-populations of our patients, e.g. temporal-lobe cases vs. extra-temporal-lobe cases, revealed no significant differences between any sub-group of patients (detailed analyses are presented in Supplementary Note 7 and Supplementary Tables 10 and 11).

Determining the success of surgery or what risk factors will impact the outcome is an ongoing topic of research, without any final verdict on a biomarker that can distinguish perfectly between successful and unsuccessful cases[25,31,33,58,59]. It is possible that multiple factors will have to be considered in addition to electrophysiological signals. Research into high-frequency oscillations[60,61], connectivity[62], and other possible candidates will have to be considered (or even combined) to obtain better results. While we used a Granger causality analysis to determine the driving node, we have not looked at other possibilities, such as phase-locking values and indices[63]; these measures might provide additional information that could help estimate the EZ, as well as distinguishing between responders and non-responders (to treatment).

To conclude, our work demonstrates the merits and benefits of noninvasive source imaging approaches and the extent which they can be used to image brain networks. In other words, our approach can localize network nodes, determine the spatial extent of these nodes, estimate the temporal variation of nodal activities, and compute the internodal connectivity and dynamics, only from noninvasive EEG recordings. We also demonstrated that epilepsy networks can be successfully imaged to determine the EZ in individual patients with results that are in concordance with invasive clinical findings. Specifically, ictal imaging was observed to be superior to spike imaging, when patients have multiple spatially inconsistent spike types in their EEG recordings.

## Methods

**Ethics statement**. Our clinical studies, recordings and data analysis were approved by and performed in accordance with the regulations of the Institutional Review Boards (IRB) of Carnegie Mellon University, Mayo Clinic, Rochester, and the University of Minnesota. Patients gave their informed consent to participate in this study.

**Patient information**. A total of 36 patients were included in this study. These patients all underwent surgery. In 34 of these patients, seizures were recorded preoperatively in high-density EEG (76 electrodes) and 35 patients had interictal spikes in their EEG recordings. These patients were scored based on the ILAE system by the physicians and were monitored for an average of 18 months (follow-up duration). All patients had at least 1 year of follow-up (refer to Supplementary Table 4 for individual's clinical information). Twenty-one patients were scored as ILAE 1 (completely seizure-free), 3 were scored as ILAE 2 (no seizures, only auras) during a 19-month follow-up period, and 12 were scored ILAE 3–6 (non-seizure-free; detailed explanation in Supplementary Table 4) during a 15-month follow-up period. The surgical resection boundaries and volume was available in 30 of these patients, for which precision and recall was calculated. This information was extracted from the post-operative MRI in all relevant patients; except for two anterior temporal lobe resections where post-operative MRI was not available. In

these cases, reports clearly provided the dimensions and location of the resection, so the resection boundary could be reconstructed. In six other patients, only intracranial electrode locations were available from CT images. These subjects were not excluded, as the SOZ was determined based on the iEEG studies in these patients. In total, 16 patients underwent iEEG study and localization error was calculated in these patients. Detailed information about the clinical information of patients included in our study, is summarized in Supplementary Tables 4 and 5.

**Algorithm outline**. The basic mathematical principles of the FAST-IRES algorithm is based on our work reported previously[64]. That version of the algorithm assumed static sources and was not suitable for studying dynamic brain processes and networks. In our new approach we have modified and enhanced the algorithm to be suitable for imaging sources changing over time (spatiotemporal processes). We proposed an efficient algorithm to solve the optimization problem formulated by our ESI approach. The basic idea of this algorithm is that sources recorded with EEG/MEG are not extremely focal but *focally extended*, as EEG/MEG signals are the superimposed post-synaptic potentials of many synchronous neuron ensembles. Additionally, the time courses of activity of these sources are detectable on the scalp as the sources are superimposed on each other over time. Based on these premises we assumed that if the TBF of the underlying brain activities are delineated from the scalp recordings through component analysis, each row of the estimated TBF corresponds to a spatially extended focal source (mathematical details are presented in the Supplementary Methods). The type of optimization problem that needs to be solved based on these assumptions is of the following form:

$$\mathbf{j}^L = \underset{\mathbf{j}}{\operatorname{argmin}} \sum_{i=1}^{N_c} \| \mathbf{W}_{d,i}^{L-1}(\mathbf{V}\mathbf{j}_i) \|_1 + \alpha \sum_{i=1}^{N_c} \| \mathbf{W}_i^{L-1}\mathbf{j}_i \|_1$$
$$\text{subject to } \operatorname{Trace}\left\{ (\phi(t) - \mathbf{K}\mathbf{j}\mathbf{A})^T \mathbf{\Sigma}^{-1}(\phi(t) - \mathbf{K}\mathbf{j}\mathbf{A}) \right\} \leq \beta^2 \quad (1)$$

where $\phi(t)$ is the scalp potential (or magnetic field) measurements over the interval of interest (an $E \times T$ matrix where $E$ is the number of measurements and $T$ is the number of time points in a given interval), $\mathbf{K}$ is the lead field matrix (an $E \times N$ matrix where $N$ is the number of sources), $\mathbf{j}$ is the unknown current density of the brain regions (an $N \times N_c$ matrix, where $N_c$ is the number of TBFs), $\mathbf{A}$ is the time course activation matrix (an $N_c \times T$ matrix) or the TBF which is given by, $\mathbf{A} = [\mathbf{a}_1(t), \mathbf{a}_2(t), \ldots]$, $\beta^2$ is essentially the noise power, to be determined by the discrepancy theorem, $\mathbf{\Sigma}$ is the covariance matrix of the noise to be determined from the baseline activity, $\mathbf{W}_{d,i}^{L-1}$ and $\mathbf{W}_i^{L-1}$ are the weights pertaining to each $\mathbf{j}_i$ and are updated with the same rule determined in IRES iterations[64], $\mathbf{V}$ is the discrete gradient operator, $\alpha$ is the hyperparameter balancing the two terms in the regularization term which will be tuned using the $L$-curve approach, and $L$ is counting the iteration steps (we implement an iterative approach to determine the extent of underlying sources objectively without applying subjective thresholds).

We proposed an efficient algorithm (coded in MATLAB) that can solve optimization problems proposed in Eq. (1) using basic convex optimization tools in an efficient and optimal manner (refer to Supplementary Note 1 and Supplementary Methods for mathematical and implementation details).

**FAST-IRES principles, limitations, and parameters**. The key principles of FAST-IRES can be summarized into four main ideas. Firstly, minimizing source edges with an $L_1$-norm regularization term. The $L_1$-norm of the spatial gradient, which is the difference between the amplitude of two neighboring sources, i.e. the edges, is minimized. This allows the solution to make sudden changes or jumps at limited locations or edges. This is due to the nature of $L_1$-norm where sudden changes of amplitude are allowed at limited number of vector elements as opposed to $L_2$-norm which enforces smooth changes. The $L_1$-norm basically allows a piecewise homogeneous solution, as changes in amplitude in the activated region are discouraged (to minimize edges). Secondly, minimizing the solution's $L_1$-norm also enforces sources to have zero background, discouraging constant backgrounds (a constant background has a zero gradient and is not perceived by the edge sparsity term so it cannot be excluded by the edge-sparsity enforcing term, alone). Thirdly, a parameter (i.e., $\alpha$) needs to be defined to balance the two terms of the regularization terms. An L-curve approach can objectively achieve this goal. Fourthly, a series of iterative re-weightings are performed, in which, the two terms of the regularization function are weighted. These weights are updated at each iteration based on the obtained solution. These weights systematically and without the intervention of an operator, slim down the solution to an extended source that fits the measurements.

The reason why these iterations will not result in an overly focused solution, is because of the edge minimization term. This term prefers extended sources and potentially prefers a constant value so that there are no variations in the solution, but it has to allow changes as the measurements have to be fitted as well. In this manner a balance is reached, and the continuation of iterations will not result in overly focused solutions. The weights also ensure that the solution and its edges are sparse and use solution amplitude (and edge amplitude) to guide the algorithm to converge to an extended solution, more easily.

The number of iterations in our algorithm (for this iterative re-weighting scheme), and most algorithms we are aware of, are difficult to pre-determine, but, the process can continue until solutions converge; that is, the relative change of the solution, normalized by the solution at the previous iteration, does not exceed a

pre-set value such as 0.0001 (default value used in our codes). Our experience with simulation and data suggests that within a few iterations, the solution converges and stops changing any further, and continuing the iterations will not shrink the solution, hence, making it overly focal; consequently, even if a different tolerance was chosen, e.g. $10^{-6}$, the solution would not change much (refer to Supplementary Fig. 12). Thus, we do not believe, that iteration numbers after the solution converges, affect the extent of the sources much, if any at all.

The weights are initiated, for solution and gradient, as identity matrices, as no a priori knowledge of sources is available. The weights' initialization does not affect the solutions. The choice of weights being reciprocal to the inverse of source amplitude, is not based on heuristics, and is a direct result of approximating the "$L_0$-norm" with a logarithm function (explained and detailed in refs. [64,65]).

To assess the effect of L-curve on FAST-IRES estimates, it must be acknowledged that the parameter $\alpha$ affects source estimates as it provides a balance between the two terms of the regularization. However, this parameter can be tuned using an L-curve approach. The L-curve approach is based on a Pareto optimality idea that both terms of the regularization should be as small as possible, basically selecting the $\alpha$ value corresponding to the knee of the L-curve. This approach in itself is not subjective. However, it is possible that due to noise or other unpredictable factors, the curve changes a little. This is a possibility that cannot be ruled out. We have provided an example in Supplementary Fig. 13 showing that even changing the $\alpha$ by a factor of 10 near the bend of the L-curve does not affect the solution much, indicating that the algorithm is robust against variability in choosing $\alpha$, as iterations can gradually compensate for such variabilities. The presented framework provides an objective measure to select $\alpha$, and this, in itself, is not problematic. As long as these choices are warranted within the model's framework and are theory-driven, and do not affect the subsequent tuning of other parameters in the model, they are not subjective. We believe that our proposed algorithm is robust to moderate variations of $\alpha$ and employing the L-curve method can systematically reduce the ranges of $\alpha$ that needs to be considered; hence it does not appear to seriously affect source estimates.

It is obvious, that this approach, like any other, is susceptible to vary with noise (as our simulations also indicate that source extent estimation does not exactly fall on a line, albeit with very high Pearson's correlation values). Changing the solution size and shape a little bit will not affect the regularization terms and may still fit the measurements well enough, so the solution is not unique within a given noise level. Thus, depending on the measurement's noise level, we can have errors in estimating the source extent, hence the variability observed in our simulation results and general decline in the algorithm's performance with increasing noise. These are the basic ideas and key parameters of the proposed approach and some of the potential limitations of FAST-IRES. Our algorithm provides a relatively simple, yet effective framework to estimate underlying sources' extent.

**General data recording and processing routines**. The patients underwent 76-channel EEG recording in the evaluation phase prior to surgery. The 76 electrodes were glued individually based on a 10–10 montage with the reference electrode at CPz (a common average reference was used for analysis and source imaging later in our analyses). The EEG recordings were sampled at a 500 Hz sampling rate, using the Xltek EEG amplifier (Natus Medical Incorporated, CA, USA), and were filtered with a high-pass filter above 1 Hz to remove spurious slow activity and possible DC shifts in the data. For ictal signals a low-pass filter with high-frequency cutoffs at 20–30 Hz was applied (as ictal signals contain a lot of muscle artifacts and are noisy) while a low-pass filter at 50 Hz was applied to interictal spikes.

For each patient an individualistic head model was built using subject's individual MRIs to form the lead-field matrix, denoted by $K$ (which is later used in solving the inverse source imaging problem). A three-layer boundary element method (BEM) model was employed[13,66] to solve the forward problem and obtain the lead-field matrix[1]. This BEM model consisted of three layers representing the brain, skull, and skin with corresponding electrical conductivities of 0.33, 0.0165, and 0.33 s/m, respectively[67]. The cortical current density (CCD)[68] model is used to develop the FAST-IRES approach.

**ICA and component selection**. In order to denoise the data and extract the TBF, we applied ICA to our EEG signals. The manner, in which ICA was applied to interictal spikes and ictal signals was different.

The ictal recordings were initially filtered between 1 and 30 Hz to remove high-frequency noise, as well as enormous muscle artifacts which spectrally span broad frequency bands. Ictal dominant frequencies are mainly in the lower frequency range, rarely exceeding 15 Hz. After initial filtering, channels that showed huge movement artifacts (or other high amplitude artifacts) were interpolated with the recordings of the four closest neighboring channels and in rare cases noisy channels were removed (2 patients and up to 4–6 electrodes).

An interval of 10 s prior to seizure onset (where seizure onset time was marked by trained epileptologists), up until seizure termination time, was selected for component analysis in each seizure (in two patients where seizures lasted for more than 2–3 min only the first 2 min were selected). Typically, intervals of 30–90 s were selected for subsequent ICA analysis (which is the typical seizure length in our cohort of patients).

After the intervals were selected for each seizure in every patient, they were fed into the ICA module available in EEGLAB[69]. An independent component decomposition of ictal data using the logistic infomax ICA algorithm was

performed, using the EEGLAB toolbox (version 14.1.1b)[70]. ICA decomposes the ictal data recoded in EEG into spatially fixed and temporally independent sources.

After components were formed, artifact components such as eye movement or eye blinks which have distinct spatial (high activity in the anterior electrodes) and temporal features (sharp high amplitude spikes at blinking or eye rolling moments), were visually identified and rejected. Movement artifacts demonstrate abnormally invariant wide-frequency activity and occasionally abnormally spatially focused topographical potential scalp maps. These well-known artifacts were rejected to form clean and denoised EEG recordings.

Furthermore, in order to identify the seizure-relevant components, independent components (ICs) that demonstrated high correlation with the temporal evolution of the seizures, i.e. silent or less activity before seizure onset and high activity after seizure onset time, were selected. It is the time courses of these components that were used as the TBF, which is subsequently fed into the FAST-IRES algorithm.

For interictal spikes, we concatenated spikes that were selected in an individual patient and performed the ICA analysis on this concatenated signal. An interval of 2 s around the spike peak was selected for each spike. In patients, where multiple spike types existed, we performed the process independently for each spike type. The general process followed for analyzing spikes was similar to that of seizures with two major differences; firstly, we filtered interictal data between 1 and 50 Hz, and secondly, after obvious artifact components were rejected, the time-course of activities were averaged (basically every two-second interval corresponding to a single spike, was selected and averaged). We then inspected these averaged IC time courses and included them as spike-related components if and only if they showed a marked increase of amplitude at the peak-time (compared to baseline). Basically, as an averaged interictal spike will show marked increase of amplitude around its peak-time (compared to its baseline), it is reasonable to expect and enforce the same criteria for the IC time-courses that are to be selected as spike-related TBFs. It is these averaged time-courses along the averaged spike, that are then input into the FAST-IRES for the source imaging problem to be solved. The number of ICs used for spike and seizure analyses are presented in Supplementary Table 12.

A noteworthy difference between the approach presented here and that of Yang et al.[40] is that, only the TBF's were selected here, i.e. $\mathbf{A}$ in Eq. (1), and the columns of $\boldsymbol{j}$ corresponding to each row of $\mathbf{A}$ were then estimated using FAST-IRES, while, Yang et al.[40] chose a different approach. They used the spatial topographical maps obtained from ICA and mapped those scalp topographies to the source space only to later recombine them based on each component's time-course of activity to obtain the source's spatiotemporal distribution.

The ICA component selection, and consequently the number of ICs, is based on objective measures, in principle. Additionally, as each ICA component will be a TBF element, the spatial distribution of that TBF element will be computed for that temporal component individually, not affecting the spatial distribution and extent of other components; so, missing a component will not affect the spatial estimates of other components. In this sense, the choice of TBF does not affect the extent of the source. It is, however, possible that missing a component might affect the overall estimate of more complex sources such as seizures. That is why the component selection is not arbitrary. Selected components are time-locked to events observed on the scalp, i.e. spike-peak and seizure onset. This automatically removes some noisy components, however, to speed up the process noisy components such as eye blinks, movement and muscle artifacts, etc. were removed by visual inspection. These components are easy to detect and non-controversial to eliminate, as is common with artifact removal in most EEG pre-processing pipelines (refer to Supplementary Fig. 14 for an example of denoising effects of ICA analysis). It is possible, in principle, to automate this whole process to minimize the effect of human interactions during this process, however, this is a project of its own and not within the scope of the current work. When doubtful if a component is signal-related or not, the component was included to avoid mistakenly discarding signals of interest (potentially introducing more noise into the system). It is acknowledged that this might affect overall estimation results, therefore, ICA must be applied carefully.

**Spectral analysis**. After denoising the ictal signals and forming the TBF, the FAST-IRES algorithm was applied to this denoised ictal signal for the first 3–5 s post seizure onset. In this manner, a spatiotemporal distribution of the seizure within source space was obtained during the initial phase of seizure onset. In line with our expectations of seizure propagation and rhythmicity, the distribution fluctuates constantly and propagates rapidly, quickly involving regions that are not in the SOZ or even spatially adjacent to it. In the case of temporal lobe epilepsy patients, activity could propagate to the contralateral lobe in a matter of a few milliseconds, making the task of determining the SOZ from this source space distribution difficult. In fact, 13 patients in our studied cohort indicated secondary generalized seizures or propagation to the contralateral side after seizure onset (earliest, after a few seconds after onset). In the IC selection process, contralateral components were observed and these components were ordinarily included in the TBF as seizure propagation to the contralateral side is expected and observed (refer to Supplementary Fig. 15 for a more illustrative example).

Following the literature in ictal source imaging, it seemed natural to filter the solution at the dominant frequency to obtain a more spatially relevant signal. To this end, the average spectral power of the pre-ictal period over all channels (a 5 s period right before seizure onset) was plotted against the same quantity during the initial 5 s right after seizure onset. These pre-ictal and ictal periods were further

divided into five non-overlapping 1 s intervals where the average spectrum was calculated in each period and averaged together to finally obtain two average spectrums; one average pre-ictal spectrum and one average early-ictal spectrum. By comparing these two spectrums the frequency band that showed distinct differences between the two spectrums (a significant increase of power compared to pre-ictal spectrum) was selected as the dominant frequency band of analysis. This frequency band was selected based on the criteria that its spectral power had to be larger than the pre-ictal spectrum and only the single most powerful lobe in the spectrum was selected (basically side-lobes of the ictal spectrum were rejected even if significantly larger than pre-ictal spectrum to ensure a narrow-band selection). The solution was filtered to keep all of the frequencies in this band and reject the frequencies outside. Subsequently the average power (energy) of this filtered solution during a 1-second interval at the beginning of seizure onset was calculated and visualized. This is best depicted in Fig. 5a. To further assess the effect of this window size on the proposed analyses, different window sizes were tested (refer to Supplementary Fig. 16 for two examples). Window sizes cannot be too large, as there is the risk of capturing propagated ictal activity. Furthermore, given that the dominant frequency of the analyzed seizures is typically within the 1–4 Hz range, including short windows will not capture full oscillations of low-frequency waves and might negatively affect results.

**Defining performance metrics**. In order to compare the shape and relative position of the estimated sources with the ground truth, an overlap metric is used. The amount of overlap between the estimated sources and the resection area is calculated and divided by either the resection area or the estimated source's area to derive a normalized overlap ratio (NOR). This NOR shows how well the two distributions match each other, with ideal values for both ratios being 1. If an overestimated or under-estimated solution is obtained, one of the two measures will be close to 1 while the other decreases significantly. These two NORs are also referred to as precision and recall in most statistical and computer science literature. Estimates were computed on the cortical surface, so surgical surfaces extracted from post-operational MRI had to be projected to the cortical surface for calculating NORs, i.e. precision and recall (Supplementary Fig. 17 depicts two examples).

Another performance metric used in this paper is the localization error (LE), which is defined as the average minimum distance between every SOZ electrode (computed in patients who underwent iEEG implantation) and the EZ estimated from the solution (either using spike imaging, ictal imaging or connectivity imaging). The distance is calculated to the boundary of the estimated EZ. The amplitude of estimated solutions by FAST-IRES do not vary much within the activated region, and SOZ electrodes cover (or are close to) regions where solutions' amplitudes reach maximum (refer to Supplementary Fig. 18 for a few examples).

The geometric mean between two numbers $a$ and $b$ is defined as $\sqrt{a.b}$, and the harmonic mean is defined as $\frac{2a.b}{a+b}$. As it can be seen in these formulas, if either value is low or close to zero, the mean value will be close to zero, and only when both values are high, will the means be high. These are appropriate measures to check if both precision and recall are attaining high values (remembering that precision and recall are between 0 and 1).

**Comparing FAST-IRES time-courses with intracranial traces**. In order to assess how close our estimated time-courses resemble true time courses, i.e. the time-course of activity of underlying sources, an additional analysis was performed in one patient. In this analysis the estimated time-course of activity from FAST-IRES were averaged in the lateral anterior temporal region and the deep para-hippocampal cortex, where intracranial electrodes were placed and were ultimately determined as SOZ. These averaged time-courses were correlated to the averaged intracranial traces of these two regions and an average correlation of 0.6 was achieved for both regions. This process was repeated for all the three seizures that were analyzed in this patient and for every EEG seizure, the extracted time-courses were compared to three randomly selected seizures recorded by the iEEG electrodes. The analysis was done in the 1–5 Hz frequency band which is the dominant frequency of this patient's ictal activity. The details of these analyses are presented in Supplementary Table 13 and Supplementary Fig. 19, as well as Supplementary Movie 3.

While these results are positive and in line with our simulation results (that underlying sources' time-courses of activity can be estimated reliably), they are limited by the fact that intracranial recordings and EEG recordings were not recorded simultaneously. While these seizures are typical and on average should represent similar spatiotemporal processes, further investigation are necessary to determine the validity of this observation — a work we intend to undertake in our future endeavors.

There is some preliminary evidence that such accurate estimations are possible. In two recent publications[5,6], estimated time-courses of activity from scalp measurements were compared to deep intracranial recordings and moderate correlations (comparable to our results) were shown. These two studies benefitted from simultaneous intracranial and scalp recordings and are highly suggestive of the possibility to delineate underlying sources' activity even from deep tissues within the brain.

**Statistical tests and reproducibility**. To determine the statistical difference among the performance metrics of spike and ictal imaging estimates, a permutation test was employed. Given that the performance metrics studied here were less likely

to have come from Normal distributions and that sample size was limited, we preferred to use the permutation test. The two groups compared to each other were the seizure analysis results and spike analysis results. In a shuffle of $10^4$ times, the labels (seizure or spike analysis results) of the normalized measures, i.e. precision, recall, geometric mean, and harmonic mean, were shuffled and the difference of their means were compared to the distribution of the $10^4$ shuffled samples to determine its relative position in the distribution[71]. The level of significance was set at $p < 0.05$ in our study. We did perform other two-sided statistical test, such as Welsh's two-sample $t$-test and Wilcoxon rank sum test[72], and were able to reach similar results for observed differences in the geometrical mean of spike and seizure-imaging analysis results, specifically in non-seizure-free patients (results are not reported). The analysis was performed in MATLAB.

**Weighting strategy for spike analysis.** Some patients had multiple types of spikes. Spike types were determined by the topographical potential scalp maps of the spikes at their peak. When analyzing these types separately, different performance metrics are observed for different spike types. For instance, a spike ipsilateral to the affected side will have much better performance metrics (higher precision and recall and lower localization error) compared to a spike on the contralateral side. In order to report a single number for each patient (so as to compare patients in a fair manner), we used the weighted average of the performance measures obtained for each spike type. The weights were determined based on the number of spikes in each group. For instance, if 100 spikes were detected in a patient and 30 of these were type-I and the rest type-2 (naming is arbitrary), when combining the performance measures of the two groups, group 1 measures were weighted by 0.3 and group 2 measures were weighted by 0.7.

One question that emerges from our analysis is: how much do our results depend on these particular weights? To answer this question and ensure that our results are independent of this choice, we performed an analysis in which the weights were randomly chosen in a uniform manner between 0 and 1 (weights are between 0 and 1 and should add up to 1). We repeated this process (comparing spike and seizure results and obtaining a $p$-value) 10,000 times, and each time the statistical tests were repeated with these new measures (or rather measures combined with new weights), to obtain the distribution of these $p$-values. Results are depicted in Supplementary Fig. 20, for two-tailed Welch's $t$-test. As it can be seen in this figure, most of the $p$-values calculated in this manner are skewed towards 0, and over 74% of the $p$-values were smaller than our significance level of 0.05. This clearly shows that the obtained results hold true, regardless of the choice of weight. While weighing the results based on the relative number of occurrences seems intuitive, there is always the chance that some spikes are missed and not counted, resulting in different weights from what we have assigned (although this ratio is less likely to change once many spikes are seen). In any case, this analysis clearly rejects the possibility of such effects.

**Reporting summary.** Further information on research design is available in the Nature Research Reporting Summary linked to this article.

## Data availability
The data that support the findings of this study are available from the corresponding author upon reasonable request. A reduced and de-identified subset of data that could be safely shared is deposited in https://doi.org/10.35092/yhjc.11996931. A source data file is provided for Figs. 2, 4c, 5c, and 6a, b and Supplementary Figs. 6c, 7a,b, 8, 9, 10, 11, and 20a–d.

## Code availability
Scripts and codes used for analysis, with sample data, are available at https://github.com/bfinl/FAST-IRES. Our developed scripts are written and developed in MATLAB (The MathWorks, Inc., MA, USA), mainly on version 2013B and also tested on 2018a/b. We employ CURRY 8 (Compumedics, NC, USA) and the EEG Lab toolbox (version 14.1.1b) for visualization and some simple pre-processing analyses (https://sccn.ucsd.edu/eeglab/index.php), as well as the eConnectome toolbox (version 1.0 beta) for connectivity analysis (https://www.nitrc.org/projects/econnectome). Additionally, some MATLAB plugins (freely available online) were used for visualization purposes; these include the Egg Head Plot toolbox (https://education.msu.edu/kin/hbcl/software.html), the Bounded Line toolbox (https://www.mathworks.com/matlabcentral/fileexchange/27485-boundedline-m), and the NotBoxPlot toolbox (https://www.mathworks.com/matlabcentral/fileexchange/26508-notboxplot).

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

## Acknowledgements

This work was supported in part by NIH grants NS096761, EB021027, MH114233, AT009263. The authors are grateful to Dr. Yunfeng Lu and Mr. Daniel Suma for discussions and manuscript editing, as well as Ms. Cindy Nelson for assistance in data collection.

## Author contributions

Study design: B.H.; Conceptualization: A.S. and B.H.; Methodology: A.S. and B.H; Formal analysis: A.S. and Z.C.; Investigation: A.S., Z.C., S.Y., B.B., G.W., and B.H.; Resources: G. W., B.B., and B.H.; Writing—original draft: A.S.; Writing—review & editing, A.S. and B. H.; Supervision: B.H.

## Competing interests

A.S. and B.H. are co-inventors of a US patent application on imaging algorithms used in this investigation, that was filed by the University of Minnesota. The authors declare no other competing interests.
