## [Peer Review File · Nature Communications]

Reviewers' Comments:

Reviewer #1:

Remarks to the Author:

According to the abstract, in this paper the authors propose a new inverse method in particular to estimate the spatial extent of neural sources from EEG data. My problem is that I couldn't find that this claim is backed from any of the results. I couldn't even find results where the estimated spatial extent in units of mm^2 was presented. I have only seen results for the average overlap with the "true" activated area. Here is the problem: if an estimate of the localization is reasonable (and that wouldn't be new in this context), then I could, hypothetically, always assume a fixed spatial extent of, say, 1cm^2 , and the overlap would probably come out reasonable, too. If the authors would simulate a single dipole plus additive noise, I guess that such a distributed source would come out as a result. That's just in the nature of this method which assumes focal extended sources. The effect of spatial extent on the measured potential is known to be extremely small for relatively focal sources. Therefore I doubt that the results presented here have anything to do with the actual extent of the sources.

There are two things missing in this paper:

a) simulations where the true extent is varied starting from zero extent to large extent, and b), presentation of results of estimated and true extent for the empirical data for all individual subjects. For the latter, at least a significant positive correlation between estimated and true extent would be justify the claim that extent is observable at all.

Reviewer #2:

Remarks to the Author:

This paper provides a noninvasive source imaging methodology to localize spatio-temporally organized brain sources from EEG. The methodology is mainly based on the authors' earlier work, Fast Spatio-Temporal Iteratively Reweighted Edge Sparsity (FAST-IRES), for source localization. The authors apply their methods to a large cohort of patients with epilepsy, in both ictal and inter-ictal conditions, and validate the noninvasive estimates of the epileptogenic zone with those obtained by iEEG and expert annotations.

Overall, the paper is well-written, the methodology is sound, and the contributions are significant. I have a number of specific comments that would like the authors to address:

Major comments:

1) The title of the paper is too general given the scope of the contributions. First, while the proposed algorithms are only validated on data from epilepsy, there is no mention of epilepsy in the title. I understand that the algorithms can potentially be applied to different paradigms (e.g., sensory or cognitive experiments), but without the authors validating their algorithms on at least one more non-epilepsy paradigm, the current title would be too general. Second, the notion of "brain networks" in the title is a bit misleading, as the authors later on show that the results using DTF analysis (Fig. 5) do not significantly improve over the simpler ictal imaging (Fig. 4) (Also see my comments 2 and 3 below

on the focal assumptions and connectivity analysis). I suggest that the authors use a more specific title that conveys the contributions accurately. A suggestion is: "Noninvasive Electromagnetic Imaging of Spatio-Temporally Distributed Epileptogenic Brain Sources".

2) On page 7, the authors give reference to [36] as evidence showing that activity of areas up to 400 mm² can be thought of as uniform. However, this conclusion is misleading. In fact, this reference argues for the converse: that a "detectable" evoked response by EEG/MEG requires the uniform activity of 40-400 mm² area of the cortex (not that the actual neural activity is 40-400 mm² in extent). In [36], it is indeed mentioned that these estimates are much larger than the actual tonotopic and amplitopic structures of the auditory cortex. The authors need to clarify that while this assumption may hold for epilepsy (where a large area of cortex exhibits uniform activity), it is not suitable for sensory modalities such as vision, audition and somatosensation where areas as small as 10 mm² get activated (e.g., tonotopy).

A similar statement appears on page 17: "This assumption is backed up by physiological recordings and studies, suggesting that in many species including human beings, a considerable amount of cortical tissue is uniformly activated during different brain functions and states." which needs to also be fixed accordingly. This also relates to comment 1 in regard to the title: brain networks in the spatial scales of < 40-400 mm² (e.g., within the somatosensory or auditory cortex) may not be revealed using this methodology.

3) While the DTF analysis sounds interesting, there is a major concern regarding its validity. Inverse solutions of EEG/MEG can never fully segregate the actual contribution of the sources. That is, the inverse operator, say F , acts on ϕ , which itself is given as $K_j A$. Therefore, unless F fully cancels K , and recovers $j A$ exactly (which is never the case) and ICA ideally factors all the temporal components with no leakage, the source estimates are a "mixture" of all true sources. As such, performing a causality analysis between source estimates (that are mixtures of all the true sources) is not reliable, as each source estimate contains elements of all the other sources as well! In my opinion, while the DTF analysis is interesting, it does not add to the already rich contributions of the authors, especially given that the connectivity analysis did not provide significantly better results than the simpler ictal imaging analysis. I suggest that the authors move the connectivity analysis to the supplementary material, and discuss the limitation I mentioned above in the context of source imaging.

Also, the statement on page 16 "This indicates that while the connectivity imaging approach provides a much more objective framework, it did not improve the ictal imaging results, significantly." is not precise. The connectivity approach relies on DTF analysis as well as correlation-based clustering which in turn pose extra assumptions on the analysis (e.g., AR order selection or number of clusters). Also, there may be situations that a unique causal source node is not evident from the connectivity analysis: consider 3 sources A, B, and C. If the DTF infers that $A \rightarrow B \rightarrow C \rightarrow A$, then none of these nodes can be selected as the EZ, and we are back in the ictal imaging case. I suggest that the authors revise this statement to "This indicates that while the connectivity imaging approach provides a richer analytical framework, it did not improve the ictal imaging results, significantly."

4) The authors repeatedly mention "subjective thresholding" (e.g., pages 5, 6, 13, 14, 15 and 23) as a sub-optimal method of obtaining source or network estimates, which I fully agree with. However, it must be noted that any source localization or connectivity analysis algorithm requires "subjective" choices at some level, albeit not in the form of thresholding. For instance, the parameter α in Eq. (1) is chosen by the L-curve approach as a form of cross-validation. By changing the cross-validation cost function (instead of the usual squared error), one may find different results. Conversely, one can use thresholding but pick the threshold level "objectively" through cross-validation.

The subjective choice in the case of FAST-IRES is tuning by the L-curve approach (as opposed to other cross-validation criteria), or the use of AIC instead of FPE in case of DTF modeling, or the correlation-based clustering approach used to create hyper-nodes (instead of say, hierarchical clustering). Also, how is N_c (number of time basis functions or ICA components) chosen in FAST-IRES? From the explanation on page 25, it seems like some of the ICs that resembled artifacts were visually (i.e., subjectively) removed. Even if the ICs are chosen based on a data-driven method (with pre-defined parameters), it adds another subtle layer of subjectivity. So, I suggest that the authors clarify on pages 5-6, 13-15 and 23 that the objectivity in their approach is based on their modeling and theoretical assumptions, as opposed to subjective thresholding which is not model- or theory-driven.

Minor comments:

1) Page 3: In the sentence "...can be formulated as a globally distributed spatio-temporal network problem.", the words "formulated" and "problem" are not appropriate. I suggest changing it to: "...can be modeled as a globally distributed spatio-temporal network."

2) Page 4: Electro/magnetoencephalograms are the traces obtained by electro/magnetoencephalography. So, in defining the EEG and MEG acronyms, I suggest using "Electroencephalography" and "Magnetoencephalography", given the context.

3) Page 4: While the low spatial resolution of EEG is mainly attributed to volume conduction effects, MEG is blind to the conductance profile of the tissue. So, the statement that "Unfortunately, the spatial resolution of traditional scalp EEG/MEG measurements is limited, due to the volume conduction effect." is not correct. Technically speaking, if observation noise is zero, and there are enough EEG or MEG sensors, the spatial resolution of EEG and MEG can be arbitrarily high (i.e., solving $Kx = \phi$ for x , where the dimension of ϕ is much more than x), modulo the blindness of MEG to the gyri. So, I suggest that the authors change this sentence to: "Unfortunately, the spatial resolution of traditional scalp EEG/MEG measurements is limited, due to low SNR and having a small number of sensors compared to the number of sources".

4) Page 6: I suggest changing "...equally important" to "...equally importantly...". Also I suggest changing "non-subjective" to "objective".

5) Page 8: I suggest changing "...is paramount to success both quantitatively, and for the interpretability of results." to "...is paramount to quantitative assessment and interpretability of the results."

6) Page 10: In the statement "The implications of this observation are discussed later in the manuscript.", please mention what part of manuscript you are referring to (Ictal Imaging vs. Spike Imaging?).

7) Page 11: I suggest changing "...foundational for its successful study." to "...foundational for its successful analysis."

8) Page 12: I suggest changing "...demonstrating that our estimates are neither an underestimate nor an overestimate of the "true" underlying EZ" to "demonstrating that our inferred sources neither underestimate nor overestimate the "true" underlying EZ".

9) Page 14: I suggest changing "involve multiple columns of j " to "involve multiple time basis

functions". First, j is not defined here, and second, the problem is when one spatial region corresponds to multiple rows of the TBF matrix. So, it is useful to make it explicit in words.

10) how is the distance between the SOZ electrodes and the source estimates defined (on page 30)? Is this the distance between the SOZ electrodes and the peak activity of a focal region? or the boundary of a focal region? Please clarify.

Reviewer #3:

Remarks to the Author:

This article proposed by Sohrabpour and colleagues presents an impressive contribution in the context of high density EEG source localization in the context of interictal spikes and seizure source localization, while evaluating their results using a method previously proposed by their group, FAST-IRES method. The method consists in assessing the underlying time course of neuronal activity using ICA to define Temporal Basis Function, while optimizing a sparse criterion (L1 based) on both the amplitude of the sources, therefore allowing the reconstruction of locally spatially extended sources, which is indeed highly important in the context of epilepsy source imaging. While more details on the method and few simulations examples are provided in supplementary material, the core of this study is to apply this method on a very large dataset of epilepsy patients (37), for whom interictal spikes and seizures were localized for most of them. All patients underwent surgery with an average of 18 months postsurgical follow up, and the resected area was considered as the ground truth for source localization evaluation. Moreover 16 patients underwent intracranial EEG (iEEG) investigations, for which localization error between source localization and the Seizure Onset Zone channel.

Whereas there is a clear interest for such a detailed study and results are really encouraging, several important major issues should be carefully addressed before deciding whether this study could be considered for publication in Nature Communication.

Major concerns:

- The introduction and discussion are relatively long describing what is the standard procedure of presurgical investigation of patients with epilepsy, using non-invasive technique and notable EEG source imaging. Several items of such description are largely known by the clinical community and should probably be summarized. More importantly, the description of their proposed method and results, in comparison with other studies published in the field, using EEG or MEG, using interictal spikes or seizures, is quite brief and limited. It would be quite important to develop this important issue to position this work in regards with other similar studies. Several important references, especially regarding methods dealing with recovering the spatial extent of the sources in spikes and seizures, have been missed.
- The discussion should also include a more clinically oriented discussion of the results. From the patient selection table (Suppl Table S1), it seems obvious that, among the very large population studied, most patients were actually characterized by Temporal Lobe Epilepsy. Moreover, in TLE, EEG source localization is usually showing better performance than MEG (mainly radially oriented generators). The paper would therefore benefit from a more accurate clinical description of the patients investigated (Patients selection section) and of a description/discussion of the results depending on the type of epilepsy (TLE, TLE with hippocampal sclerosis only, extra-temporal epilepsy, MRI lesion, ...).
- Illustrations provided in the paper (except 1 video), only consist in temporal lobe epilepsy cases, it would have been interesting to see the performance of the method on extra-temporal epilepsy cases.
- The time period of few seconds selected for the localization of the SOZ seems relatively long.

Whereas this could be relevant for temporal seizures, localization frontal seizures that are known to propagate quite rapidly might be biased by such selection.

- From a methodological point of view, the authors are mentioning briefly in the introduction few other source localization methods able to recover the underlying spatial extents of the sources, while arguing that these methods are less clinically relevant because they depend on “subjective thresholds”. This is not completely true, since some quantitative approach for thresholding have been proposed for some methods, while for other it has been shown that because of the use of sparse constraints, spatial maps with localized with great contrast compared to other more standard linear methods (MNE, dSPM, sLORETA), therefore providing highly clinically relevant results. On this issue, the authors should also clarify how several parameters of their proposed methods, such as the number of ICA components, the number of iterations and stopping criteria, the L-curve and the way the weighting factor for both the source and their gradient were initialized and evolved from iteration to iteration. These are several important methodological parameters that could have an impact on the resulting spatial extent of the generators. This important issue, in comparison to other methods proposed in the literature, should be clarified and at least discussed.
- The author should clarify what was the space of possible spatial extents explored by their method during the iterative approach, what are the key parameters allowing to recover such spatial extent and finally, whether these extents could be in agreement with the large spatial extents involved during epileptic activity, as suggested from intra-cranial recordings.
- Overall the evaluation of the accuracy of the results is based on spatial overlapping measurements between the results of the proposed source localization technique (spikes, seizure, connectivity during seizure) with the resected area, using validation metrics of precision and recall. It was not clear to me whether the source space was a cortical surface grid or a 3D grid. Indeed if sources are estimated along the cortical surface, then this evaluation with the resection area would require a surface to volume transformation that was not mentioned. Moreover, it was not clearly explained how was the resected region segmented, and co-registered with the pre-surgical MRI. The authors should also discuss whether their evaluation could also be biased by possible co-registration errors, especially if non-linear co-registration was considered between pre-op and post-op MRIs.
- When comparing seizure localization to spike localization, the fact that several spike types were averaged (weighted average) even if coming from completely different topographies is seriously biasing spike localization results, which have been shown to perform quite well in other EEG or MEG studies. In clinical practice, one should really consider the spikes that are of primary importance and this will depend of a global clinical picture. Similarly, some patients might also exhibit seizures starting from the right or from the left independently. Did the author record such cases; the duration of the recording should be included for every patient. Could it be that the recording was not sufficiently long to record seizures from both side, in case of bilateral temporal epilepsy patients (if any?). To provide a less biased evaluation of spike versus seizure localization, only spikes showing a scalp topography in agreement with the seizure should be considered in the quantitative analysis (Fig. 6 and statistical tests in suppl. material).
- How many ICA components were selected for every spike or seizure? Was every seizure analyzed independently using ICA or where the seizure concatenated and at the end how were results combined to provide one seizure localization result for every patient. In line with this idea, even if ICA is indeed well cleaning the signal, at the end the algorithm is performing only one “averaged spike” source localization. Therefore, at the end, authors are considering only one source localization result for every spike type, therefore lacking statistics. Several authors have proposed the use of single spike source localization, or sub-averaged of clusters of spike, in order to provide statistics over several spike localization. In this context, the idea of proposing a measure of spike-to-spike reliability in the source space (after single spike source localization) could significantly improve source localization results.
- The proposed method and results on connectivity are of great interest and quite promising. However I would further clarify and rename these section “Connectivity studies during seizures”, in order to make sure not to confuse with the concept of resting state connectivity analysis, which is quite

important in our community. Because the method is completely driven by ICA and components able to extract signal with good SNR, I am not convinced this approach could be used in the context of resting state connectivity, but to study the flow of information during the seizure this is completely fair. However, in this regards, the problem of volume conduction that still exists after source localization, was not addressed (briefly in suppl material) and should be discussed.

Minor concerns

- There are few typos throughout the manuscript that should be considered.
- The concept of field of view, or scope, in the context of electrophysiology using invasive and non-invasive techniques is not very clear. It should be clarified or possibly another term could also be proposed to better describe this property.
- For measurement of Localization Error with one channel SOZ, it would be important to clarify what part of the source localization was considered to measure this distance, the main peak, the whole extended area? Were there cases of large intensity source localization results non covered of iEEG implantation?
- It could be of interest to illustrate how much ICA is able to reduce the impact of motion and muscle artifact during the seizure, which is a primary importance.
- For all video, a time scale should be added to the underlying time course to get a better idea of the scale of the signal (even if time samples are mentioned, a temporal scale is quite important). For the video illustrating reconstruction in deep mesial structure, it would be of great interest to compare the time course localized by the methods to real recordings, during typical spikes or seizures (even if these data were not recording simultaneously, it should be possible to find example that are likely to be quite similar). This would be important to judge the quality of the source reconstruction results in deeper structure.
- On a minor note regarding simulation results in suppl. material, it would have been quite interesting to see the behavior of the method in the presence of realistic noise exhibiting correlations in space and time, as opposed to white noise.
- Fig. 5b: there might be a typo in the title which should refer to "Connectivity Imaging Estimates" as opposed to "Spike Imaging Estimate"

Response to Reviewers

We thank the reviewers very much for their critical and constructive comments. We have undertaken a significant effort to substantially revise the manuscript including substantial additional simulations, new analyses and statistical tests on the empirical data, and addressing all the comments and suggestions raised by the reviewers. These resulted in the addition of Fig. 2 and updating Fig. 6 in the main body, adding 13 new figures and 7 tables to the supplementary materials and updating supplementary videos. As a result, we believe that the manuscript has been significantly improved and strengthened. The revised text in the manuscript and supplementary materials is printed in **RED** to make it easier to locate. We have also replied to the reviewers' comments one by one in the following (in **BLUE**).

Reviewer #1 (Remarks to the Author):

According to the abstract, in this paper the authors propose a new inverse method in particular to estimate the spatial extent of neural sources from EEG data. My problem is that I couldn't find that this claim is backed from any of the results. I couldn't even find results where the estimated spatial extent in units of mm^2 was presented. I have only seen results for the average overlap with the "true" activated area. Here is the problem: if an estimate of the localization is reasonable (and that wouldn't be new in this context), then I could, hypothetically, always assume a fixed spatial extent of, say, 1 cm^2 , and the overlap would probably come out reasonable, too. If the authors would simulate a single dipole plus additive noise, I guess that such a distributed source would come out as a result. That's just in the nature of this method which assumes focal extended sources. The effect of spatial extent on the measured potential is known to be extremely small for relatively focal sources. Therefore, I doubt that the results presented here have anything to do with the actual extent of the sources.

-- We have revised the manuscript substantially to better convey our findings that the proposed method provides an objective way of determining source extent ranging from focal sources to distributed sources with a variety of extents. We have clarified the concept on our novel source extent imaging, and the challenges the field is facing, with substantial additional simulation results and clinical results (in patients) to back up our claims. These new analyses are reported in the main body in Fig. 2, a section titled "Numerical Simulation Study" and another section titled "Comparing Extent Estimations of Empirical Data with Surgical Resection Size". Additionally, these extra analyses are reflected in supplementary Texts T2 & T5, Fig. S1 & S11 and Table S3.

As suggested by the reviewer, we have included new evidence (Monte Carlo computer simulations and estimated empirical extents of all subjects included in the study) to our manuscript to back up the claims we have made about extent estimation. We believe that these new pieces of evidence better clarify our claims. We would like to point out that overlap in itself is not an adequate measure (which is why we have looked at normalized overlaps in our study), since as the reviewer noted, a fixed small source will always have a relatively high overlap with a "true" source or surgical resection (particularly if the "true" source is fairly extended). However, the overlap normalized by

the “true” source’s extent and (simultaneously) the estimated solution’s extent will not be high, at the same time. These normalized ratios, also referred to as precision and recall, will both achieve high values, i.e. close to 1, *if and only if* the estimated source’s size is comparable to the true source’s size and completely overlaps with it; for instance, a small circle of 1 cm² might lie well within a resection area, hence obtaining a high precision value (as the overlap area is equal to the solution’s area resulting in a ratio of 1), but the recall value will be low (as the resection area is larger than the overlap area). Attaining high values for both the precision and recall, as described in this work, is a suitable measure to assess the efficiency of our proposed method. As a result, the presented overlap ratios, specifically the geometric mean of precision and recall (which attains high values, only if both, precision and recall have high values), has in effect quantitatively provided evidence for our claim. Therefore, a method that is sensitive to the underlying source’s extent is quite necessary if reasonable estimates of source sizes is to be obtained. It is necessary to note, that estimating the extent and shape of source distribution is not unique in a statistical sense, that is, depending on noise and the geometrical complexities of the cortex where the source originates from, the estimated source can make mistakes. Our simulation results, to follow, indicate that source extent estimation is accompanied by a variance. However, the overall trend characterized by the Pearson’s correlation, unequivocally shows that our imaging algorithm is capable of distinguishing between different source sizes on average, which is highly relevant for clinical applications such as determining resection size and area in focal epilepsy and/or brain mapping studies. It is also important to note that one of the benefits of using high-density electrode set-ups is that the scalp potential topographical maps can be sampled more comprehensively and the possibility of detecting more minor changes of scalp potential distributions due to smaller sources becomes more realistic. The benefits of employing high-density EEG recordings, compared to low-density EEG, has been indicated by multiple studies [1]–[4]. Some claims as to the limitations of source estimation might be due to limited and prior studies of EEG with earlier generations of EEG recording device. We do acknowledge, however, that small changes in source size could possibly be lost in noisy scalp potentials, which is why the extent estimation has a stochasticity to it, as indicated by our results. This point is emphasized in our manuscript. In our simulation studies we are using an EEG cap of 128 electrodes and our recordings from focal epilepsy patients have 76 recording channels.

There are two things missing in this paper:

a) simulations where the true extent is varied starting from zero extent to large extent, and

-- As suggested by the reviewer, we have conducted a series of additional Monte Carlo simulations. We varied source extent, i.e. average radius, from 0¹ mm to around 40 mm, and randomly selected 115 location on the cortex to simulate these extended

¹ It must be noted that the smallest source possible to place on the cortex is a dipole; a dipole by nature, is a representation of the activity of a small cortical area. In our model this means a single surface triangle, where the dipole is placed at the center, is activated (surface triangles tessellating the brain). The area of such triangular elements is about 1.5-3 mm² (equivalent to an average radius of 0.7-1 mm).

sources. The forward problem was solved (on a fine cortical model with 66,490 triangular surface elements) and noise was randomly generated and added to the simulated scalp potential recordings for a total of 10 times for each source configuration. The noise was a combination of real EEG signals (from resting state EEG segments without any epileptic activity) and white Gaussian noise. The inverse problem was solved on a coarser cortical model (31,530 triangular surface elements) and the estimated sources' extent are plotted against the simulated sources' extent for comparison. This whole procedure was repeated for 4 different signal-to-noise ratio (SNR) scenarios, namely 20, 10, 5, and 0 dB. The SNR is defined as the average signal power over all channels to noise power over all channels in a decibel scale. A total of 1,150 simulations were performed for each SNR condition. We further investigated the Pearson's correlation between the estimated and simulated extent values and found a significant correlational value of 0.88, 0.85, 0.83, and 0.78 for the 20, 10, 5, and 0 dB SNR cases, respectively (the p-value for these measures were close to zero and negligent). The correlation between time-courses of activity for the estimated and simulated sources were also calculated yielding high values over 0.98, indicating that our proposed approach is capable of performing reasonable spatio-temporal estimation. The 95% confidence interval for these correlational values is summarized in the following table, Table R1. We have also included some examples of the spatio-temporal source reconstruction in Fig. R2 to provide some reference.

Figure R1. Monte Carlo Simulation Results of Extent Estimation. The results of estimated extent from our FAST-IRES algorithm is plotted against the simulated sources' extent for 4 different SNR conditions. An example of a mesio-temporal source is also presented for all the SNR conditions, along its estimated time-course of activity. The red line in the four plots is the identity line and is provided for reference.

Table R1. Summary of Statistics (Monte Carlo Simulations)

SNR (dB)	Pearson's Correlation (between simulated and estimated extent)			Correlation between estimated and simulated time-courses of activity	
	Coefficient	95% Interval	P-value	Mean \pm std	95% Interval
20	0.879	[0.866 0.891]	0	0.998 \pm 0.0006	[0.997 0.999]
10	0.847	[0.830 0.862]	5.5×10^{-317}	0.997 \pm 0.0012	[0.995 0.999]
5	0.832	[0.813 0.849]	2.7×10^{-296}	0.994 \pm 0.0027	[0.988 0.999]
0	0.785	[0.762 0.806]	5.5×10^{-241}	0.983 \pm 0.0079	[0.967 0.998]

The results presented here are also reported in the main body and supplementary section of our manuscript (details to follow).

Figure R2. Monte Carlo Simulation Examples. Examples of three different sources estimated under the four tested SNRs is given in (a). All the simulation results for the 4 different SNR conditions are plotted simultaneously for a better comparison (b). The estimated time-course of activity from all simulation examples under each SNR condition is presented in (c). The thick line is the mean among all different conditions and the shaded area shows the 99th percentile of the distribution among the 1,150 simulation cases.

b), presentation of results of estimated and true extent for the empirical data for all individual subjects.

-- As suggested by the reviewer, we have plotted the estimated epileptogenic zone using both ictal analysis and inter-ictal spike analysis in all patients studied. Each dot, in the graphs depicted in Fig. R3, indicates the data from one patient. For each patient, the averaged inter-ictal spikes were analyzed (on average patients have 29 spikes), and the area of the inter-ictal solution was calculated. Additionally, for ictal analysis, the area of each seizure analyzed in a patient was calculated and averaged to generate one data point for the patient. Our results presented in Fig. R3, indicate that the size of the estimated epileptic region from seizure and inter-ictal spike correlates significantly with the resection size. The Pearson's correlation was found to be 0.61 for ictal estimates (p-value of 0.0006) and 0.48 for spike estimates (p-value of 0.009). We further divided the patients into seizure-free and non-seizure-free indicated by the red and blue color, and did not observe a marked difference between the extent estimation between the two groups. Pearson's correlation in effect determines how dependent two random variables are, and a correlation of 1 indicates that two variables are fully dependent on each other, which suggests that they may be driven by a common factor or variable. A correlation of 1 between estimated epileptogenic zone and resection region, would have indicated that these processes are driven by common factors, as they would have been dependent processes. However, planning for the surgery and determining the region to be resected is not a process that is merely dependent on electrophysiological recordings and interpretation; other factors such as ictal semiology, presence of lesions observed in MRI, etc. are considered. Consequently, this process is determined by many factors and not just electrophysiological recordings, although electrophysiological recordings play an important role, undoubtedly. As a result, we did not expect our estimates to show a perfect Pearson's correlation. Our results do show positive and significantly moderate values of correlation with surgical resection, indicating that our approach provides estimates that are relevant and related to the clinical routine. It is evident that most of our estimates are smaller than the resection, which is probably due to the well-known clinical practice of making resections large enough to ensure post-surgical seizure-freedom.

There are two outliers in Fig. R3, that deserve more attention (a black box is used to pinpoint these two data points). The first point is a seizure-free case where our algorithm suggests a larger epileptogenic tissue based on ictal analysis, and the other point, belongs to a patient who did not become seizure-free, yet our algorithm suggests a smaller resection area than the actual resection the patient received. In the first case, a focal lesion was observed in the MRI, and as such, the clinical team decided to only remove the lesion, while the actual seizure was originated from a larger brain tissue. Such factors, as mentioned before, are considered in determining the surgical strategy and boundary, in practice, and is why a perfect correlation should not be expected from electrophysiology-based methods. For the second case, our algorithm suggested a region which was not covered by the resection (it was still in the vicinity of the resection), thus, while the overall region suggested by the algorithm was smaller it proposed areas not covered by the resection.

This agrees with the prior explanation of how clinical decisions encompass a multitude of factors, still relying heavily on electrophysiological data, as the moderate value of correlational analysis suggests. Interestingly, and in line with our observations in this study, seizure analysis shows stronger correlation, compared to spike analysis, to

clinical findings, implying that ictal source imaging analysis might have a more positive impact in the clinical decision-making process.

This result combined with other results already reported in the manuscript indicates that our proposed method is capable of distinguishing between different source sizes, albeit, with existing stochasticity and reasonable variation due to the inverse process and noise.

Figure R3. Estimated and “True” Extent of Empirical Data. The area of resection and estimated sources’ area (for both inter-ictal and seizure analysis) are plotted against each other. Two examples of estimated epileptogenic tissue from ictal and inter-ictal imaging are provided as well.

For the latter, at least a significant positive correlation between estimated and true extent would be justify the claim that extent is observable at all.

-- We agree with the reviewer's comment and believe that the significantly high correlations provided in this response support the claim that information about the underlying source can be recovered, reliably. We have updated our manuscript to include these results in Fig. 2 in the main body, and Fig. S1, Fig. S11 and Table S3 in the supplementary materials. We also added these results and the relevant discussions in the body of our manuscript.

Specifically, in the Results section we added a section named "Numerical Simulation Study" and introduced the results as follows:

"In order to thoroughly investigate the performance of our framework we performed a Monte Carlo simulation in which random locations on the cortex were selected and an extended source was simulated at that location, and the EEG signals generated by these source configurations were calculated. A combination of real EEG noise (scalp potential recordings of actual human subjects during rest with no evident epileptic activity) and white Gaussian noise was added to the generated scalp potentials. Multiple noise levels were generated resulting in signal-to-noise ratios (SNRs) of 0, 5, 10, and 20 dB (1,150 simulations were performed for each noise condition). The sources were then estimated using our proposed algorithm and the estimated sources' extent was plotted against the simulated sources' extent to show how well source extent can be estimated by our proposed algorithm. Fig. 2 presents the results of our simulation along a typical example of a mesio-temporal source that has been localized in all SNR conditions. The estimated time-course of activity for this source matches the simulated time-course, nicely. A significantly high Pearson's correlation value of 0.88 was found for these results. More details about the simulation protocol, Pearson's correlational values and the related statistics, and more examples of estimated sources are presented in the supplementary materials (Table S3-S5, Fig. S1-S3 and Video S1)."

As well as in the supplementary text T2:

"In order to provide a thorough evaluation of FAST-IRES a Monte Carlo simulation was performed. Random locations were selected on the cortex as seed points, around which an extended source was created (115 location on the cortex). The extent size of these sources ranged from 0 mm to around 40 mm (each location was randomly assigned an extent). Once the extended sources were placed on a fine cortical model (66,490 triangular surface elements) the forward problem was solved to generate the scalp potentials of these sources and different levels of noise was added to this simulated EEG to obtain SNR levels of 0, 5, 10, and 20 dB. The noise was a combination of realistic correlated EEG noise (from non-epileptic periods of EEG recordings) and additive white Gaussian noise (equal power to realistic EEG noise). For each simulation case at an SNR, 10 noise realizations were analyzed (accruing to a total of 1,150 cases for each SNR level). The inverse was then subsequently calculated for these noisy scalp maps on a coarse cortical grid (31,530 triangular surface elements). The extent of the estimated sources was compared against the simulated sources extent and plotted against each other (Fig. 2 in the main body of this paper). Calculating the Pearson's correlational value and its related p-value, we found statistically significant and high correlational values, proving that our method robustly estimates underlying sources' extents (refer to Table S3 for quantitative and more detailed results).

It must be mentioned that as is evident in the results presented in Fig. 2 of the main text, our estimates show the stochastic nature of our algorithm (as in almost any source imaging algorithm), as our estimates do not fall exactly on a straight line. This variation is partly due to noise and partly due to the geometrical complexities of the cortex, specifically for larger sources. Fig. S1 shows some simulation examples.”

Additionally, we discussed the results of comparing empirical data extent size to resection size in a new section in the Results section titled “Comparing Extent Estimations of Empirical Data with Surgical Resection Size”, as follows:

“In order to directly compare our results with clinical findings, we calculated the size of our estimated epileptogenic tissue using seizure imaging analysis and spike imaging analysis (consistent spikes) and plotted these values against resection size. The results are presented in supplementary Fig. S11. The analyses revealed that a significant Pearson’s correlation of 0.61 (p-value of 0.0006) and 0.48 (p-value of 0.009) was observed between empirical results and resection size, for ictal and inter-ictal imaging results, respectively. Our results often indicated smaller sizes for the epileptogenic brain tissue compared to resection size, which is probably due to the “conservative” approach taken towards surgery. Resection is typically chosen large enough to ensure the removal of EZ and ensure seizure freedom (please refer to supplementary text T5 for more discussion).”

In response to the reviewer’s comment as well as that of Reviewer 3, we have also added some more explanations about the basics of our method in the Methods under a section titled “FAST-IRES Principles, Limitations and Parameters”, as well,

“The key principles of FAST-IRES can be summarized into 4 main ideas. Firstly, minimizing source edges with an L1-norm regularization term. The L1-norm of the spatial gradient, which is the difference between the amplitude of two neighboring sources, i.e. the edges, is minimized. This allows the solution to make sudden changes or jumps at limited locations or edges. This is due to the nature of L1-norm where sudden changes of amplitude are allowed at limited number of vector elements as opposed to L2-norm which enforces smooth changes. The L₁-norm basically allows a piecewise homogeneous solution as changes in amplitude in the activated region, are discouraged (to minimize edges). Secondly, minimizing the solution’s L1-norm also enforces sources to have zero background, discouraging constant backgrounds (a constant background has a zero gradient and is not perceived by the edge sparsity term so it cannot be excluded by the edge-sparsity enforcing term alone). Thirdly, a parameter (α) needs to be defined to balance the two terms of the regularization terms. As mentioned before, an L-curve approach can objectively achieve this goal. Fourthly, a series of iterative re-weightings are performed, in which, the two terms of the regularization function are weighted. These weights are updated at each iteration based on the obtained solution. These weights systematically and without the intervention of an operator, slim down the solution to an extended source that fits the measurements.

The reason why these iterations will not result in an overly focused solution, is because of the edge minimization term. This term prefers extended sources and potentially prefers a constant value so that there are no variations in the solution, but it has to allow changes as the measurements have to be fitted as well. In this manner a balance is reached, and the continuation of iterations will not result in overly focused solutions. The weights also ensure that the solution and its edges are sparse and use solution amplitude (and edge amplitude) to guide the algorithm to converge to an extended solution, more easily.

The number of iterations in our algorithm (for this iterative re-weighting scheme), and most algorithms we are aware of, are difficult to pre-determine, but, the process can continue until solutions converge; that is, the relative change of the solution, normalized by the solution at the previous iteration, does not exceed a pre-set value such as 0.0001 (default value used in our codes). Our experience with simulation and data suggests that within a few iterations, the solution converges and stops changing any further, and continuing the iterations will not shrink the solution making it overly focal; consequently, even if a different tolerance was chosen, e.g. 10^{-6} , the solution would not change much (refer to Fig. S12). Thus, we do not believe, that iteration numbers after the solution converges, affect the extent of the sources much, if any at all.

The weights are initiated, for solution and gradient, as identity matrices, as no a priori knowledge of sources is available. The weights' initialization does not affect the solutions. The choice of weights being reciprocal to the inverse of source amplitude, is not based on heuristics, and is a direct result of approximating the "L₀-norm" with a logarithm function (explained and detailed in ^{63,85}).

To assess the effect of L-curve on FAST-IRES estimates, it must be acknowledged that the parameter α affects source estimates as it provides a balance between the two terms of the regularization. However, this parameter can be tuned using an L-curve approach. The L-curve approach is based on a Pareto optimality idea that both terms of the regularization should be as small as possible, basically selecting the α value corresponding to the knee of the L-curve. This approach in itself is not subjective. However, it is possible that due to noise or other unpredictable factors, the curve changes a little. This is a possibility that cannot be ruled out. We have provided an example in Fig. S13 showing that even changing the α by a factor of 10 near the bend of the L-curve does not affect the solution much, indicating that the algorithm is robust against variability in choosing α , as iterations can still fix such variabilities, gradually. As discussed previously, the presented framework provides an objective measure to select α , and this, in itself, is not problematic. As long as these choices are warranted within the model's framework and are theory-driven, and do not affect the subsequent tuning of other parameters in the model, they are not subjective. We believe, that our proposed algorithm is robust to moderate variations of α and employing the L-curve method can systematically reduce the ranges of α that needs to be considered; hence it does not appear to seriously affect source estimates.

It is obvious, that this approach, like any other, is susceptible to vary with noise (as our simulations also indicate that source extent estimation does not exactly fall on a line, albeit with very high Pearson's correlation values). Changing the solution size and shape a little bit will not affect the regularization terms and may still fit the measurements well enough, so the solution is not unique within a given noise level. Thus, depending on the measurements noise level, we can have errors in estimating the source extent, hence the variability observed in our simulation results and general decline in the algorithm's performance with increasing noise. These are the basic ideas and key parameters of the proposed approach and some of the potential limitations of FAST-IRES. Our algorithm provides a relatively simple (compared to more elaborate Hierarchical Bayesian methods ²⁶, for instance), yet effective framework to estimate underlying sources' extent."

Reviewer #2 (Remarks to the Author):

This paper provides a noninvasive source imaging methodology to localize spatio-temporally organized brain sources from EEG. The methodology is mainly based on the authors' earlier work, Fast Spatio-Temporal Iteratively Reweighted Edge Sparsity (FAST-IRES), for source localization. The authors apply their methods to a large cohort of patients with epilepsy, in both ictal and inter-ictal conditions, and validate the noninvasive estimates of the epileptogenic zone with those obtained by iEEG and expert annotations.

Overall, the paper is well-written, the methodology is sound, and the contributions are significant. I have a number of specific comments that would like the authors to address:

-- We thank the reviewer for her/his favorable comment. Our previous work, titled IRES, was an algorithm developed for static processes or single topographical scalp potential maps, while our current work, FAST-IRES, which is built upon the previous IRES approach, represents a new algorithm allowing spatio-temporal imaging of source extent. We have carefully addressed the reviewer's comments and made substantial revision of the manuscript, as detailed below. These changes are reflected throughout our manuscript and supplementary materials; particularly, the title of our manuscript was modified and the "Connectivity Imaging" section and its corresponding results were transferred to the supplementary materials as suggested by the reviewer (with cautionary notes on potential limitations). A section titled "FAST-IRES Principles, Limitations and Parameters" was also added to the main body of our manuscript, and additional notes were added in the Discussion and Methods section.

Major comments:

1) The title of the paper is too general given the scope of the contributions. First, while the proposed algorithms are only validated on data from epilepsy, there is no mention of epilepsy in the title. I understand that the algorithms can potentially be applied to different paradigms (e.g., sensory or cognitive experiments), but without the authors validating their algorithms on at least one more non-epilepsy paradigm, the current title would be too general. Second, the notion of "brain networks" in the title is a bit misleading, as the authors later on show that the results using DTF analysis (Fig. 5) do not significantly improve over the simpler ictal imaging (Fig. 4) (Also see my comments 2 and 3 below on the focal assumptions and connectivity analysis). I suggest that the authors use a more specific title that conveys the contributions accurately. A suggestion is: "Noninvasive Electromagnetic Imaging of Spatio-Temporally Distributed Epileptogenic Brain Sources".

-- We thank the reviewer for this suggestion. We have changed the title to "Noninvasive Electromagnetic Imaging of Spatio-Temporally Distributed **Epileptogenic Brain Sources**" as the reviewer suggested.

2) On page 7, the authors give reference to [36] as evidence showing that activity of areas

up to 400 mm² can be thought of as uniform. However, this conclusion is misleading. In fact, this reference argues for the converse: that a "detectable" evoked response by EEG/MEG requires the uniform activity of 40-400 mm² area of the cortex (not that the actual neural activity is 40-400 mm² in extent). In [36], it is indeed mentioned that these estimates are much larger than the actual tonotopic and amplitopic structures of the auditory cortex. The authors need to clarify that while this assumption may hold for epilepsy (where a large area of cortex exhibits uniform activity), it is not suitable for sensory modalities such as vision, audition and somatosensation where areas as small as 10 micro mm² get activated (e.g., tonotopy).

A similar statement appears on page 17: "This assumption is backed up by physiological recordings and studies, suggesting that in many species including human beings, a considerable amount of cortical tissue is uniformly activated during different brain functions and states." which needs to also be fixed accordingly. This also relates to comment 1 in regard to the title: brain networks in the spatial scales of < 40-400 mm² (e.g., within the somatosensory or auditory cortex) may not be revealed using this methodology.

-- Thank you and point well taken. Our original intent was to claim that extended cortical areas, with relatively synchronous activity, have to be activated in large-scale phenomenon that we are typically studying, so the focally extended source modeling would be a suitable reflection of this fact. We have changed these two sections as follows to more accurately reflect our view:

"This property, **most likely**, is not specific to epileptic sources³⁸ and has been suggested in **other large-scale phenomena, where extended cortical areas have to be synchronously activated to produce detectable signals at scalp-level measurements such as EEG and MEG** ^{39,40}. **Note that fine micro-scale organizations of the brain activity, are not perceivable in EEG/MEG or any surface recordings** ³⁹ and we are making no claim in this work to be able to recover such sources. The idea of a focally extended source is to model large-scale brain signals and organizations that are typically recorded in surface measurements such as EEG and MEG."

And,

"This assumption is backed up by physiological recordings and studies, suggesting that, a considerable amount of cortical tissue is activated during different brain functions and states; **it is such large-scale phenomena that EEG/MEG sensors record during epilepsy-related brain activity (micro-scaled brain organization are not considered)** ^{39,40}. Our proposed framework, which is constituted on these fundamental properties is therefore applicable to study **other** normal and pathological **large-scale** brain networks that are inherently distributed spatio-temporal processes."

3) While the DTF analysis sounds interesting, there is a major concern regarding its validity. Inverse solutions of EEG/MEG can never fully segregate the actual contribution of the sources. That is, the inverse operator, say F , acts on ϕ , which itself is given as $K \cdot A$. Therefore, unless F fully cancels K , and recovers A exactly (which is never the case) and ICA ideally factors all the temporal components with no leakage, the source estimates are a "mixture" of all true sources. As such, performing a causality analysis between source estimates (that are mixtures of all the true sources) is not reliable, as

each source estimate contains elements of all the other sources as well! In my opinion, while the DTF analysis is interesting, it does not add to the already rich contributions of the authors, especially given that the connectivity analysis did not provide significantly better results than the simpler ictal imaging analysis. I suggest that the authors move the connectivity analysis to the supplementary material, and discuss the limitation I mentioned above in the context of source imaging.

Also, the statement on page 16 "This indicates that while the connectivity imaging approach provides a much more objective framework, it did not improve the ictal imaging results, significantly." is not precise. The connectivity approach relies on DTF analysis as well as correlation-based clustering which in turn pose extra assumptions on the analysis (e.g., AR order selection or number of clusters). Also, there may be situations that a unique causal source node is not evident from the connectivity analysis: consider 3 sources A, B, and C. If the DTF infers that $A \rightarrow B \rightarrow C \rightarrow A$, then none of these nodes can be selected as the EZ, and we are back in the ictal imaging case. I suggest that the authors revise this statement to "This indicates that while the connectivity imaging approach provides a richer analytical framework, it did not improve the ictal imaging results, significantly."

-- We thank the reviewer for this comment. As briefly mentioned before, we consider that once the nodes of a large-scale network (with observable signals in EEG), are determined and the time-course of activities of these nodes are extracted, any network analysis such as DTF can be performed to study the network properties. Our simulation results (further enhanced in response to reviewer 1) indicate that time-courses of activity can be estimated fairly accurately. Additionally, in response to reviewer 3 we looked at the correlation of our intra-cranial recordings and estimated time-courses of activity from our method and found that relatively high correlations can be achieved (Figure R13 and Table R7). We do acknowledge that such analyses do not convey a final verdict on this issue as the recordings were not simultaneous, however, source imaging techniques are known to help remove the volume conduction effect substantially, albeit, we cannot claim thoroughly.

In response to the point brought up about circular causality, if the strength of connection between A and C are different, then the node with the dominant connection can be chosen as the main driving node or EZ (basically the node with higher outflow). However, we agree that some confusions could potentially remain in such examples. Our aim was to show that once a spatio-temporal estimate of underlying sources is provided, time series analysis tools such as DTF can be applied to ultimately benefit from a richer analytical framework. Following the reviewer's suggestion, we have moved these results to the supplementary with a sufficient discussion of possible limitations of DTF analysis, in a section named "Connectivity Imaging". We have changed the aforementioned sentence based on the reviewer's suggestion.

In the section titled "Connectivity Imaging" in Results section we, now, have:

"FAST-IRES provides a spatio-temporal estimate of underlying brain sources. This implies that in addition to the location and spatial extent of underlying brain activities, the time-course of such activities can be estimated, as well. Subsequent network or connectivity analyses can be performed based on FAST-IRES results. We have outlined and performed a connectivity analysis based on our FAST-IRES results

which bears novel ideas of how to potentially implement such analyses in tandem with the proposed framework. The motivation, details and potential limitations of this study is presented in the supplementary text T3 and Fig. S4-S7 and Table S6. Please refer to the supplementary materials for a detailed review and discussion of this proposed analysis.”

And in the supplementary text T2 under “Connectivity Imaging”, we have a sub-title named “A Cautionary Note on Potential Limitations” to discuss the potential limitations of this analysis,

“While source imaging reduces the effect of volume conduction and is shown to delineate underlying brain sources ^{6,7,14}, it is not easy to ascertain that this goal has been achieved. FAST-IRES simulation results indicate that this is a strong possibility. Our initial experiment correlating superficial and deep iEEG recordings with estimated time-courses from FAST-IRES (these findings must be considered with caution as we have explained in the manuscript) indicates that this might be a reasonable assumption for FAST-IRES. However, more work has to be done to reach a clear verdict on this issue. Thus, it is possible that connectivity analyses, DTF analysis, might have been affected by residual effects of volume conduction that might not have been thoroughly eliminated by our proposed inverse approach.

It is also important to note that these connectivity analyses are not to be confused with resting-state connectivity analyses performed in fMRI studies or electrophysiological studies. The term connectivity imaging, in this work, refers to directional connectivity of epilepsy networks using DTF analysis.”

4) The authors repeatedly mention "subjective thresholding" (e.g., pages 5, 6, 13, 14, 15 and 23) as a sub-optimal method of obtaining source or network estimates, which I fully agree with. However, it must be noted that any source localization or connectivity analysis algorithm requires "subjective" choices at some level, albeit not in the form of thresholding. For instance, the parameter α in Eq. (1) is chosen by the L-curve approach as a form of cross-validation. By changing the cross-validation cost function (instead of the usual squared error), one may find different results. Conversely, one can use thresholding but pick the threshold level "objectively" through cross-validation.

The subjective choice in the case of FAST-IRES is tuning by the L-curve approach (as opposed to other cross-validation criteria), or the use of AIC instead of FPE in case of DTF modeling, or the correlation-based clustering approach used to create hyper-nodes (instead of say, hierarchical clustering). Also, how is N_c (number of time basis functions or ICA components) chosen in FAST-IRES? From the explanation on page 25, it seems like some of the ICs that resembled artifacts were visually (i.e., subjectively) removed. Even if the ICs are chosen based on a data-driven method (with pre-defined parameters), it adds another subtle layer of subjectivity. So, I suggest that the authors clarify on pages 5-6, 13-15 and 23 that the objectivity in their approach is based on their modeling and theoretical assumptions, as opposed to subjective thresholding which is not model- or theory-driven.

-- We thank the reviewer for the excellent point, elegantly worded, that subjectivity which is not model or theory-driven is problematic, as all algorithms have to make choices at some level. We want to further add, that being able to propose frameworks (models,

theories, etc.) that are further capable of justifying or tuning hyper-parameters of a model are also an important feature. For instance, the choice of the L-curve is based on the idea of pareto optimality and trying to minimize both terms of the objective function simultaneously and itself is a choice. However, it is a choice that provides a consistent and robust method to choose a hyper-parameter. On the other hand, choices such as subjective thresholding are purely based on operator's experience and judgement. Thus, a framework that enables such consistent choices is also important. FAST-IRES also has assumptions and has had to make choices, but we believe these choices are consistent and objective within the adopted framework.

We have addressed this issue in a section titled "FAST-IRES Principles, Limitations and Parameters" in the Methods section of the paper as well as the Discussion and modified some of our claims brought up by the reviewer.

To answer the reviewer's question on FAST-IRES hyper-parameters other than L-curve, which provides one possible alternative to choose the hyper-parameter α , the other mentioned variables are related to the post-hoc analysis of FAST-IRES estimates. While choosing Akaike information criterion (AIC) or Akaike's final prediction error (FPE) criterion does make some changes in the estimated order, verifying and possibly improving these hyper-parameters is easy and can be done systematically. We have used hierarchical clustering to form the hyper-node in our post-hoc connectivity analysis and the number of nodes were selected to be the same as the number of time-basis functions (TBFs). To answer the reviewer's comments on the number of independent components (ICs) selected when forming the TBFs for the FAST-IRES, i.e. N_c , we basically select every IC that shows significant correlation with the EEG trace; for spikes for instance, this would be ICs that have a significant peak at the center of the epoch when breaking the ICs into the same number of spike epochs and averaging it over these epochs, or the onset of seizure in ictal data analysis; basically selecting ICs that are time-locked to epileptic events. Whenever in doubt, the ICs were included to avoid discarding useful ICs. Noisy ICs such as eye artefacts, movement artefacts, muscle artefact which have specific topographical features or abnormal temporal spikes in the IC time-course were removed. Such removal can be done automatically, i.e. objectively, in principle, although it was not attempted in this work. We agree with the reviewer that some level of subjectivity is inherent in every algorithm, but the reproducibility, consistency and robustness that different methods exert is not equal between different methods and a meritorious feature of this work, while acknowledging that some minor portions and elements of our work can be implemented in a more rigorous manner in the future.

Based on these comments we have modified our claims and also included a section titled "FAST-IRES Principles, Limitations and Parameters" in the Methods section of the paper to discuss these potential limitations and the ideas of subjectivity based on model/theory as opposed to naïve subjectivity, as follows.

In the Discussion section we have the following notes:

"It is apt to be reminded that our proposed approach like any model, is based on a theoretical framework and some model assumptions. For instance, we chose an L_1 optimization framework to formulate the inverse problem while another method might formulate the problem within a Bayesian framework of analysis. These choices are inevitable and are not the kind of "subjectivity" that must be avoided, as they are the product of a model-driven effort. However, based on the choices and assumptions different models make, subsequent steps of the analyses they must perform and the characteristics of the

results they obtain, will differ. For instance, while the coherent maximum entropy on the mean (cMEM) algorithm²⁵ produces focally extended sources and successfully employs Otsu's threshold to distinguish between background activity and signal, it still needs this post-hoc addendum to achieve this goal. This is due to the manner in which this problem is formulated and while determining the source extent is resolved objectively by Otsu's thresholding algorithm, it shows that every model-driven choice and assumption affects later processes, which might make one algorithm preferable to another.

FAST-IRES has three significant advantages compared to existing approaches. Firstly, it can, in itself and through the process of iterative re-weighting, distinguish signal and background without the need of any post-hoc analysis. Secondly, its hyper-parameters are easy to tune and track. For instance, the hyper-parameter α which balances between sparsity and edge-sparsity of the solution can be tuned by the L-curve approach (refer to supplementary materials for a detailed mathematical treatment of the subject). This easy and intuitive method of tuning a hyper-parameter in itself is a major advantage of our proposed method compared to some more mathematically based assumptions, such as independence of variables in some Bayesian methods or hyper-parameters decided by trial and error or expert opinion (an insight into the L-curve approach and its relation to Pareto optimality is presented elsewhere⁶³). While our proposed method also needs to make such model-driven choices, it still maintains the aforementioned merits compared to other algorithms. Thirdly, our approach does not pre-parcellate the brain^{19,24,25}, in a data-driven or pre-determined manner, to estimate spatial extent. A more detailed discussion of basic ideas of how FAST-IRES operates and what are some of the assumptions it is based on and the hyper-parameters it has to tune are presented in the Methods section under "FAST-IRES Principles, Limitations and Parameters".

And in a section titled "FAST-IRES Principles, Limitations and Parameters" we have,

"The key principles of FAST-IRES can be summarized into 4 main ideas. Firstly, minimizing source edges with an L1-norm regularization term. The L1-norm of the spatial gradient, which is the difference between the amplitude of two neighboring sources, i.e. the edges, is minimized. This allows the solution to make sudden changes or jumps at limited locations or edges. This is due to the nature of L1-norm where sudden changes of amplitude are allowed at limited number of vector elements as opposed to L2-norm which enforces smooth changes. The L1-norm basically allows a piecewise homogeneous solution as changes in amplitude in the activated region, are discouraged (to minimize edges). Secondly, minimizing the solution's L1-norm also enforces sources to have zero background, discouraging constant backgrounds (a constant background has a zero gradient and is not perceived by the edge sparsity term so it cannot be excluded by the edge-sparsity enforcing term alone). Thirdly, a parameter (α) needs to be defined to balance the two terms of the regularization terms. As mentioned before, an L-curve approach can objectively achieve this goal. Fourthly, a series of iterative re-weightings are performed, in which, the two terms of the regularization function are weighted. These weights are updated at each iteration based on the obtained solution. These weights systematically and without the intervention of an operator, slim down the solution to an extended source that fits the measurements.

The reason why these iterations will not result in an overly focused solution, is because of the edge minimization term. This term prefers extended sources and potentially prefers a constant value so that there are no variations in the solution, but it has to allow changes as the measurements have to be fitted as well. In this manner a balance is reached, and the continuation of iterations will not result in overly focused solutions. The weights also ensure that the solution and its edges are sparse and use solution amplitude (and edge amplitude) to guide the algorithm to converge to an extended solution, more easily.

The number of iterations in our algorithm (for this iterative re-weighting scheme), and most algorithms we are aware of, are difficult to pre-determine, but, the process can continue until solutions converge; that is, the relative change of the solution, normalized by the solution at the previous iteration, does not exceed a pre-set value such as 0.0001 (default value used in our codes). Our experience with simulation and data suggests that within a few iterations, the solution converges and stops changing any further, and continuing the iterations will not shrink the solution making it overly focal; consequently, even if a different tolerance was chosen, e.g. 10^{-6} , the solution would not change much (refer to Fig. S12). Thus, we do not believe, that iteration numbers after the solution converges, affect the extent of the sources much, if any at all.

The weights are initiated, for solution and gradient, as identity matrices, as no a priori knowledge of sources is available. The weights' initialization does not affect the solutions. The choice of weights being reciprocal to the inverse of source amplitude, is not based on heuristics, and is a direct result of approximating the "L₀-norm" with a logarithm function (explained and detailed in ^{63,85}).

To assess the effect of L-curve on FAST-IRES estimates, it must be acknowledged that the parameter α affects source estimates as it provides a balance between the two terms of the regularization. However, this parameter can be tuned using an L-curve approach. The L-curve approach is based on a Pareto optimality idea that both terms of the regularization should be as small as possible, basically selecting the α value corresponding to the knee of the L-curve. This approach in itself is not subjective. However, it is possible that due to noise or other unpredictable factors, the curve changes a little. This is a possibility that cannot be ruled out. We have provided an example in Fig. S13 showing that even changing the α by a factor of 10 near the bend of the L-curve does not affect the solution much, indicating that the algorithm is robust against variability in choosing α , as iterations can still fix such variabilities, gradually. As discussed previously, the presented framework provides an objective measure to select α , and this, in itself, is not problematic. As long as these choices are warranted within the model's framework and are theory-driven, and do not affect the subsequent tuning of other parameters in the model, they are not subjective. We believe, that our proposed algorithm is robust to moderate variations of α and employing the L-curve method can systematically reduce the ranges of α that needs to be considered; hence it does not appear to seriously affect source estimates.

It is obvious, that this approach, like any other, is susceptible to vary with noise (as our simulations also indicate that source extent estimation does not exactly fall on a line, albeit with very high Pearson's correlation values). Changing the solution size and shape a little bit will not affect the regularization terms and may still fit the measurements well enough, so the solution is not unique within a given noise level. Thus, depending on the measurements noise level, we can have errors in estimating the source extent, hence the variability observed in our simulation results and general decline in the algorithm's performance with increasing noise. These are the basic ideas and key parameters of the proposed approach and some of the potential limitations of FAST-IRES. Our algorithm provides a relatively simple (compared to more elaborate Hierarchical Bayesian methods ²⁶, for instance), yet effective framework to estimate underlying sources' extent."

We have further discussed these issues in the "Independent Component Analysis (ICA) and Component Selection" of Methods section,

"The ICA component selection, and consequently the number of ICs, is based on objective measures, in principle. Additionally, as each ICA component will be a TBF element, the spatial distribution of that TBF element will be solved for that temporal component individually, not affecting the spatial distribution and extent of other components; so, missing a component will not affect the spatial estimates of other components. In this sense, the choice of TBF does not affect the extent of the source. It is,

however, possible that missing a component might affect the overall estimate of more complex sources such as seizures. That is why the component selection is not arbitrary. Selected components are time-locked to events observed on the scalp, i.e. spike-peak and seizure onset. This automatically removes some noisy components, however, to speed up the process noisy components such as eye blinks, movement and muscle artifacts, etc. were removed by visual inspection. These components are easy to detect and non-controversial to eliminate, as is common with artifact removal in most EEG pre-processing pipelines (refer to Fig. S14 for an example of denoising effects of ICA analysis). It is possible, in principle, to automate this whole process to minimize the human inter-action portion of this process, however, this is a project of its own and not within the scope of the current work. When doubtful if a component is signal-related or not, the component was included to avoid potential loss of signals of interest (potentially introducing more noise into the system). It is acknowledged that this might affect overall estimation results and as a result ICA must be applied carefully.”

Minor comments:

-- We appreciate the detailed comments and have addressed all of them.

1) Page 3: In the sentence "...can be formulated as a globally distributed spatio-temporal network problem.", the words "formulated" and "problem" are not appropriate. I suggest changing it to: "...can be modeled as a globally distributed spatio-temporal network."

-- Thank you for the suggestion, we have changed the sentence according to the suggestion.

2) Page 4: Electro/magnetoencephalograms are the traces obtained by electro/magnetoencephalography. So, in defining the EEG and MEG acronyms, I suggest using "Electroencephalography" and "Magnetoencephalography", given the context.

-- Agreed and revised accordingly.

3) Page 4: While the low spatial resolution of EEG is mainly attributed to volume conduction effects, MEG is blind to the conductance profile of the tissue. So, the statement that "Unfortunately, the spatial resolution of traditional scalp EEG/MEG measurements is limited, due to the volume conduction effect." is not correct. Technically speaking, if observation noise is zero, and there are enough EEG or MEG sensors, the spatial resolution of EEG and MEG can be arbitrarily high (i.e., solving $Kx = \phi$ for x , where the dimension of ϕ is much more than x), modulo the blindness of MEG to the gyri. So, I suggest that the authors change this sentence to: "Unfortunately, the spatial resolution of traditional scalp EEG/MEG measurements is limited, due to low SNR and having a small number of sensors compared to the number of sources".

-- There is much debate on the spatial resolution of EEG/MEG as related to volume conduction effect, SNR, etc. We have revised the sentence as the reviewer suggested, to avoid confusion.

4) Page 6: I suggest changing "...equally important" to "...equally importantly...". Also I suggest changing "non-subjective" to "objective".

-- Thanks, updated accordingly.

5) Page 8: I suggest changing "...is paramount to success both quantitatively, and for the interpretability of results." to "...is paramount to quantitative assessment and interpretability of the results."

-- Thanks and updated accordingly.

6) Page 10: In the statement "The implications of this observation are discussed later in the manuscript.", please mentioned what part of manuscript you are referring to (Ictal Imaging vs. Spike Imaging?).

-- The reviewer is correct. We have fixed this issue by providing a clear reference to the appropriate location in the manuscript. More explanations cannot be given at this point of the manuscript, as ictal imaging results have to be presented first.

"The implications of this observation are discussed later in the manuscript in the "Ictal Imaging vs. Spike Imaging" section."

7) Page 11: I suggest changing "...foundational for its successful study." to "...foundational for its successful analysis."

-- Thanks and updated accordingly.

8) Page 12: I suggest changing "...demonstrating that our estimates are neither an underestimate nor an overestimate of the "true" underlying EZ" to "demonstrating that our inferred sources neither underestimate nor overestimate the "true" underlying EZ".

-- Thanks and updated accordingly.

9) Page 14: I suggest changing "involve multiple columns of j" to "involve multiple time basis functions". First, j is not defined here, and second, the problem is when one spatial region corresponds to multiple rows of the TBF matrix. So, it is useful to make it explicit in words.

-- Thanks and updated accordingly.

10) how is the distance between the SOZ electrodes and the source estimates defined (on page 30)? Is this the distance between the SOZ electrodes and the peak activity of a focal region? or the boundary of a focal region? Please clarify.

-- This definition is the distance of SOZ electrode to boundary of the estimated source, although most of our SOZ electrodes fell within the extended source and resulted in 0 LE (some examples are provided in Fig. R11). We have clarified this in the manuscript now.

Reviewer #3 (Remarks to the Author):

This article proposed by Sohrabpour and colleagues presents an impressive contribution in the context of high density EEG source localization in the context of interictal spikes and seizure source localization, while evaluating their results using a method previously proposed by their group, FAST-IRES method. The method consists in assessing the underlying time course of neuronal activity using ICA to define Temporal Basis Function, while optimizing a sparse criterion (L1 based) on both the amplitude of the sources, therefore allowing the reconstruction of locally spatially extended sources, which is indeed highly important in the context of epilepsy source imaging. While more details on the method and few simulations examples are provided in supplementary material, the core of this study is to apply this method on a very large dataset of epilepsy patients (37), for whom interictal spikes and seizures were localized for most of them. All patients underwent surgery with an average of 18 months postsurgical follow up, and the resected area was considered as the ground truth for source localization evaluation. Moreover 16 patients underwent intracranial EEG (iEEG) investigations, for which localization error between source localization and the Seizure Onset Zone channel.

Whereas there is a clear interest for such a detailed study and results are really encouraging, several important major issues should be carefully addressed before deciding whether this study could be considered for publication in Nature Communication.

-- We thank the reviewer for her/his favorable comments and critical review. As the reviewer has mentioned, our current approach, FAST-IRES, which is a novel spatio-temporal analysis algorithm, is built upon on an older idea on static inverse imaging which only dealt with single time-point topographical scalp potentials. In the revision, we have clarified to convey the technical innovation of FAST-IRES algorithm which we report in this work, and also performed new analyses based on the reviewer's suggestions. We have substantially revised the manuscript to address the reviewer's comments and concerns, and believe that the manuscript has been improved and strengthened significantly. Based on the reviewer's suggestions we have added multiple sections to the main body and supplementary materials of the manuscript, particularly in the Introduction, Results, Methods and Discussion sections. Additionally, a total of 10 new figures and 6 tables were also added to our manuscript and supplementary materials. These changes are detailed in our point-to-point response to the reviewer's comments, in the following.

Major concerns:

- The introduction and discussion are relatively long describing what is the standard procedure of presurgical investigation of patients with epilepsy, using non-invasive technique and notable EEG source imaging. Several items of such description are largely known by the clinical community and should probably be summarized. More importantly, the description of their proposed method and results, in comparison with other studies published in the field, using EEG or MEG, using interictal spikes or seizures, is quite brief and limited. It would be quite important to develop this important issue to position this work in regards with other similar studies. Several important references, especially regarding methods dealing with recovering the spatial extent of the sources in spikes and seizures, have been missed.

-- We thank the reviewer for this comment. As the readership of the journal might not all be familiar with the routine clinical practices of processing epilepsy data, a comprehensive overview was presented. As the reviewer suggested, we have shortened some of our introductory materials to be more concise and to the point. Additionally, as the reviewer suggested we have added more references and explanation to the already existing algorithms that have analyzed spike and seizure signals. These changes are reflected in the Introduction and Discussion as follows:

We have added further explanations into the Introduction as follows,

“Although recent attempts have been made to address this issue, by estimating extended sources¹⁷⁻²³, these methods still require the application of **post-hoc** thresholds in order to distinguish between the true underlying brain activity and irrelevant background activity. These thresholds may lead to biases which work within datasets with known ground-truth, but can produce highly variable results in novel circumstances or exploratory analysis. **Some of these recently proposed methods generate relatively extended sources which better distinguish background activity from desired brain activity (compared to conventional approaches)^{19,24-26}, which in return, enables them to use more objective thresholding schemes such as Otsu’s thresholding technique²⁷ to discard background activity. However, the dependence of these algorithms on post-hoc addendums suggests that the modeling framework needs improvement. While, such algorithms are more prone to be useful for clinical applications compared to their conventional counterpart, there is much room for improvement. Furthermore, the aforementioned algorithms, either employ pre-parcellation of the cortex to estimate extended sources or extensively employ Bayesian techniques to achieve this goal and require tuning several hyper-parameters.”**

And in the Discussion section we have,

“It is apt to be reminded that our proposed approach like any model, is based on a theoretical framework and some model assumptions. For instance, we chose an L_1 optimization framework to formulate the inverse problem while another method might formulate the problem within a Bayesian framework of analysis. These choices are inevitable and are not the kind of “subjectivity” that must be avoided, as they are the product of a model-driven effort. However, based on the choices and assumptions different models make, subsequent steps of the analyses they must perform and the characteristics of the results they obtain, will differ. For instance, while the coherent maximum entropy on the mean (cMEM) algorithm²⁵ produces focally extended sources and successfully employs Otsu’s threshold to distinguish between background activity and signal, it still needs this post-hoc addendum to achieve this goal. This is

due to the manner in which this problem is formulated and while determining the source extent is resolved objectively by Otsu's thresholding algorithm, it shows that every model-driven choice and assumption affects later processes, which might make one algorithm preferable to another.

FAST-IRES has three significant advantages compared to existing approaches. Firstly, it can, in itself and through the process of iterative re-weighting, distinguish signal and background without the need of any post-hoc analysis. Secondly, its hyper-parameters are easy to tune and track. For instance, the hyper-parameter α which balances between sparsity and edge-sparsity of the solution can be tuned by the L-curve approach (refer to supplementary materials for a detailed mathematical treatment of the subject). This easy and intuitive method of tuning a hyper-parameter in itself is a major advantage of our proposed method compared to some more mathematically based assumptions, such as independence of variables in some Bayesian methods or hyper-parameters decided by trial and error or expert opinion (an insight into the L-curve approach and its relation to Pareto optimality is presented elsewhere⁶³). While our proposed method also needs to make such model-driven choices, it still maintains the aforementioned merits compared to other algorithms. Thirdly, our approach does not pre-parcellate the brain^{19,24,25}, in a data-driven or pre-determined manner, to estimate spatial extent. A more detailed discussion of basic ideas of how FAST-IRES operates and what are some of the assumptions it is based on and the hyper-parameters it has to tune are presented in the Methods section under "FAST-IRES Principles, Limitations and Parameters".

- The discussion should also include a more clinically oriented discussion of the results. From the patient selection table (Suppl Table S1), it seems obvious that, among the very large population studied, most patients were actually characterized by Temporal Lobe Epilepsy. Moreover, in TLE, EEG source localization is usually showing better performance than MEG (mainly radially oriented generators). The paper would therefore benefit from a more accurate clinical description of the patients investigated (Patients selection section) and of a description/discussion of the results depending on the type of epilepsy (TLE, TLE with hippocampal sclerosis only, extra-temporal epilepsy, MRI lesion, ...).

-- We thank the reviewer for this comment. Generally speaking, temporal lobe epilepsy cases are more abundant in adult population and hence more represented in our database. We have not seen any statistically significant difference between temporal and extra-temporal lobe cases when comparing ictal and inter-ictal analysis results. As the reviewer suggested, we have added a more precise description of our patients in the "Patient Information" section of our manuscript. We have also added extra ANOVA and statistical analyses to investigate if any significant differences or groupings can be observed in our ictal or inter-ictal data analysis, and no such effect was observed. The only exception we found was in the spike analysis results between patients whose MRI were normal or indicated lesions. Upon further investigation it was observed that among the 11 patients who had MIT visible lesions, about 73% (8 out of 11) were patients who had multiple spike types in their EEG. As discussed in the manuscript, multiple inconsistent spike types affect the spike analysis results. Our data suggest that patients with MRI visible lesions were more likely to have multiple spike types which makes seizure data analysis more relevant for these patients. We believe that while spike analysis, in its own right, performs as equally well as ictal analysis, in cases where multiple spike types are present, if results are inconsistent more evidence is required to proceed, ictal analysis

provides this extra information. We have included the detailed statistical results of the analysis in the supplementary materials (Tables R2 and R3) and included a discussion of these results in the Discussion.

“Furthermore, we performed analysis of variance (ANOVA) for our results and different sub-populations of our patients, i.e. temporal-lobe vs extra-temporal-lobe patients, sclerosis vs. non-sclerosis patients, and patients with visible lesions in their MRI vs normal-MRI patients, without finding any significant differences between any sub-population group other than seizure-free and non-seizure-free groups. The only exception was in the spike analysis results of patients who had a clear lesion in their MRI and patients who had normal MRI. When investigated further, it was revealed that (in our database), from the 11 patients with MRI-visible lesions, 8 were among the patients who had multiple spike types in their EEG recordings. As a result, we believe that this observed difference is an indirect effect of the inconsistent spike types and not the existence of lesions. It is good to note that although the results of our temporal-lobe epilepsy patients were better than extra-temporal lobe epilepsy patients, this difference proved not to be statistically significant (detailed analyses are presented in Table S10 and Table S11 in the supplementary materials).”

Table R2. ANOVA Analysis of Imaging Results Among Clinical Conditions								
One-way ANOVA Analysis for Ictal and Inter-ictal Source Imaging Results (Geometric Mean of Precision and Recall)								
	Temporal vs. Extra-temporal Lobe		Mesial Sclerosis vs. Non-sclerosis		Lesion in the MRI vs. Normal MRI		Seizure-free vs. Non-seizure-free	
	F-statistics	p-value	F-statistics	p-value	F-statistics	p-value	F-statistics	p-value
Seizure	2.56	0.12	0.1	0.76	2.59	0.12	0.12	0.73
Spike	0.2	0.66	0.26	0.61	5.14	0.03 *	1.94	0.18
4-way ANOVA Analysis for Ictal and Inter-ictal Source Imaging Results (Geometric Mean of Precision and Recall)								
Seizure	1.29	0.27	0.05	0.82	1.86	0.19	0.01	0.93
Spike	0.03	0.87	0.01	0.94	3.96	0.06	1.38	0.25

Table R3. Rank-sum Statistical Test of Imaging Results for Different Clinical Conditions (Geometrical Mean of Precision and Recall)									
	Temporal	Extra-temporal	p-value	Sclerosis	Non-Sclerosis	p-value	Lesion in MRI	Normal MRI	p-value
Seizure	0.72±0.15	0.61±0.15	0.11	0.71±0.15	0.69±0.15	0.74	0.65±0.19	0.74±0.09	0.36
Spike	0.57±0.34	0.5±0.29	0.3	0.52±0.39	0.58±0.29	0.89	0.43±0.33	0.68±0.27	0.03 *
p-value	0.29	0.69	xxx	0.38	0.3	xxx	0.11	0.97	xxx

- Illustrations provided in the paper (except 1 video), only consist in temporal lobe epilepsy cases, it would have been interesting to see the performance of the method on extra-temporal epilepsy cases.

-- We thank the reviewer for bringing up this point. We have updated the main figures 4 and 5 in the main body, to include extra-temporal examples. We are also including a few more examples comparing spike and seizure analysis in extra-temporal patients here for the reviewer (throughout the examples provided in the following figures). We did not observe a statistically significant difference between temporal and extra-temporal patients in our group, although temporal patients did achieve higher mean values for the geometric mean of the normalized overlaps. This statistical analysis might have been limited by the fewer number of extra-temporal cases (compared to temporal cases), which is a factor we could not control. We have included this in the Discussion. We have cautioned about the statistics and provided these examples as some extra cases of extra-temporal patient data analysis results for the benefit of readers (throughout the paper and the supplementary materials).

“It is good to note that although the results of our temporal-lobe epilepsy patients were better than extra-temporal lobe epilepsy patients, this difference proved not to be statistically significant (detailed analyses are presented in Table S10 and Table S11 in the supplementary materials).”

- The time period of few seconds selected for the localization of the SOZ seems relatively long. Whereas this could be relevant for temporal seizures, localization frontal seizures that are known to propagate quite rapidly might be biased by such selection.

-- We agree with the reviewer that couple of seconds after ictal onset, seizure activity can propagate from focal onset, specifically in extra-temporal cases. We would like to clarify that we did not use all the 3-5 seconds that was analyzed for imaging ictal activity, to determine the seizure onset zone. As indicated in our paper a shorter window of 1 second after seizure onset was used to determine the SOZ. This is because seizure propagation after seizure onset, can quickly spread to other regions of the brain. In order to show this, we have changed the window size we used to obtain the SOZ estimate in two patients; if window sizes around the 1-second window are used then no significant change in the estimated region is observed, but, if the window length is increased, even for the temporal-lobe example presented here, the seizure might have already propagated to the contralateral side. In fact, this is a common phenomenon in temporal lobe cases studied in this paper. Additionally, as indicated previously, no significant difference was observed between temporal and extra-temporal lobe cases in our data. Furthermore, given that most of our seizures indicated strong spectral power peaks in low frequencies as low as 1/2-4 Hz, a window of 0.5-1 second was chosen to ensure at least one cycle of oscillation from the dominant frequency was included in the window of analysis, yet as discussed, we did not increase the window length extensively as there is the risk of seizure propagation following seizure onset. We have more detailed discussions about seizure propagation in temporal lobe cases, in response to one of the reviewer's following comments. A good portion of our temporal lobe cases show propagation of seizure to the contralateral side or secondary generalization activity, which

indicates that our temporal lobe cases are not typically more constrained than extra-temporal lobe cases.

The figure and the related discussions are presented in Fig. R4 (also Fig. S16 in the supplementary materials) and a more detailed discussion of the importance of considering appropriate window length for determining the SOZ is also included in the Methods section of the manuscript to bring the readers' attention to this important issue brought up by the reviewer.

“To further assess the effect of this window size on the proposed analyses, different window sizes were tested (refer to Fig. S16 for two examples). Window sizes cannot be too long, as there is a risk of propagation of ictal activity and given that the dominant frequency of the analyzed seizures is typically within the 1-4 Hz range, including short windows will not capture full oscillations of low-frequency waves and might negatively affect results.”

Figure R4. Effect of Window Size on Epileptogenic Zone Estimations. Two cases are presented in this figure, a parietal-lobe case on the top row and temporal-lobe on the bottom row. For each case, the ipsilateral and contralateral view are presented to provide a view of how much seizure activity may or may not propagate following seizure onset. After a few seconds following the seizure onset, ictal activity may propagate to other brain regions, thus the window of analysis, i.e. the average activity of the dominant ictal frequency, cannot be too long as indicated by the temporal-lobe case presented here.

- From a methodological point of view, the authors are mentioning briefly in the introduction few other source localization methods able to recover the underlying spatial

extents of the sources, while arguing that these methods are less clinically relevant because they depend on “subjective thresholds”. This is not completely true, since some quantitative approach for thresholding have been proposed for some methods, while for other it has been shown that because of the use of sparse constraints, spatial maps with localized with great contrast compared to other more standard linear methods (MNE, dSPM, sLORETA), therefore providing highly clinically relevant results. On this issue, the authors should also clarify how several parameters of their proposed methods, such as the number of ICA components, the number of iterations and stopping criteria, the L-curve and the way the weighting factor for both the source and their gradient were initialized and evolved from iteration to iteration. These are several important methodological parameters that could have an impact on the resulting spatial extent of the generators. This important issue, in comparison to other methods proposed in the literature, should be clarified and at least discussed.

-- We agree with the reviewer that certain recent methods proposed for source imaging obtain extended solutions (some have been cited in our manuscript). We also agree that these, and a lot of existing conventional methods are applicable to clinical applications. However, we believe that there is a need to develop a principled approach for imaging distributed source extent from EEG/MEG that does not depend on post-hoc addendums to determine the extent of underlying sources and is developed in a manner in which its hyper-parameters can be tuned, easily, objectively and robustly. Our method provides a simple, yet efficient algorithm that provides extended solutions that estimate the extent of the underlying source well; additionally, the iterative re-weighting scheme introduced in our approach automatically takes care of background noise, softly removing the background noise to provide an estimate of the source extent (what is referred to as support estimation in mathematical terms). This idea is quite central and unique. The algorithm, in itself and without post-hoc interventions, provides the extent estimation systematically, going through iterations that converge quickly within a few iterations. Our proposed algorithm does not rely on post-hoc addendums such as Otsu’s threshold to delineate background activity from signals of interest. This is a major difference between our algorithm and other more recently proposed algorithms. This is not a minor coincidence, but because of the way the problem is set up. We will try to emphasize the differences of our proposed algorithm with more recent algorithms from the Maximum Entropy on the Mean (MEM) family algorithms, specifically coherent-MEM (cMEM), and 4-ExSo-MUSIC algorithm to provide a feasible context for our discussion [14]–[16]. Before we begin, however, we would like to point out two points. Firstly, any model is based on some assumptions and has hyper-parameters that need to be fit. This in itself is not problematic, and quite the norm. Secondly, not all models are equal in terms of how their hyperparameters can be tuned; some are more intuitive and can be easily tuned (in a robust manner) and some are more mathematical and typically there are not easy ways to tune them, except for using subjective thresholds or general methods such as cross validation. Model assumptions do, however, affect the characteristics of the solution, i.e. how smooth or focal it is for instance, and what post-hoc analyses can be performed on them. For instance, applying Otsu’s method on results obtained from Minimum-norm type algorithms will not yield suitable estimates of the extent as it inherently does not distinguish between noise and signal very sharply, because of using L2-norm, which is

known to overly smooth the results. Our focus in the paper was to distinguish our algorithm apart from some of the conventional methods that suffer from these issues. We believe that while our algorithm like any other algorithms, is based on some assumptions and must tune some hyperparameters, at the same time, such parameters can be tuned easily and our approach, generally, has fewer number of assumptions and is developed in a simpler, yet effective, framework. We agree with the reviewer that some of the more recent algorithms have better characteristics (compared to conventional methods) and are clinically relevant, but we still believe our algorithm has clear advantages and benefits. In the following we have tried to achieve two tasks; one is to briefly compare our approach to the two aforementioned algorithms, and the other is to discuss some of the choices and assumptions our algorithm is based on and the reviewer has brought up, to clarify our approach's distinction. We have included this discussion in our paper for readers' benefit and also to provide a clearer and hopefully less biased view of other algorithms that exist and are contributing meaningfully to the field.

Both cMEM and 4-ExSo-MUSIC rely on some sort of pre-parcellation of the cortical surface (called clusters in cMEM and circular disks in 4-ExSo-MUSIC). While, such parcellations are done in a data-driven fashion, it introduces extra parameters that need to be tuned. For instance, in 4-ExSomUSIC there is an iterative approach of starting with a coarse grid and then iteratively finding the more optimal disk size, based on 4th order statistics. This is a time-consuming and highly complex process [15]. For cMEM the equivalent would be to determine the neighboring spatial scale, i.e. how much to extend the seed-points derived from projected singular value decomposition of the data into the source space. It is claimed that an $s=4$ corresponding to parcels of size 2.5 cm^2 is suitable for data analysis [14], [15], based on some previous exploration of data [14]. This spatial scale parameter, basically, limits the size of the sources that can be estimated by nature (sources smaller than a parcel). Additionally, other than heuristic approaches, it is not yet perceivable how can this parameter be tuned. For instance, in Fig. 4 of [15] sources with extents smaller than the parcel size show huge variability in their estimates indicating that a much less accurate estimate can be obtained for these sources (compare this to our simulations). We are not claiming that in real data, a parcellation of kernel sizes of 2.5 cm^2 would not be enough, but simply that such an assumption is inherent in these methods (and our approach is free from such constraints). FAST-IRES on the other hand has no such assumptions, and assumes no need for parcellation of the cortex, a priori or data-driven. We do use the surface triangular elements to model neighborhood dipoles on the cortex, which typically have areas of $1.5\text{-}3 \text{ mm}^2$, but this is not parcellation. The algorithm then assumes that the spatial gradient, i.e. edges or jumps of the solution, should be minimized with an L1-norm regularization (to allow for jumps so sources can switch on and off). It employs an iterative approach, which is a set of weights that update after each iteration, proportional to the inverse of the solution amplitude at each location, to gradually turn off dipoles at cortex location that have lower amplitude. The number of iterations in our algorithm, and most algorithms we are aware of, are difficult to pre-determine, but, we can continue until solutions converge, that is, the change in the solution's amplitude from one iteration to the next normalized by the previous solution's norm, does not exceed a pre-set value such as 0.0001 (used in our codes). Our experience with simulation and data suggests that within a few iterations, the solution converges and stops changing much further, and continuing the iterations will not shrink

the solution to the point that it becomes overly focal. As a result, even if a different tolerance was chosen, e.g. 10^{-6} , the solution would not change much. Thus, in all fairness, we do not believe, that iteration numbers after the solution converges, which can be checked easily, affect the extent of the sources much, if any at all. We initiate these weights, for solution and gradient, always as identity matrices, as we have no a priori knowledge of sources, so initializing the weights in this manner does not affect our solution at all. The choice of weights being reciprocal to the inverse of source amplitude, is not a heuristic one per se, and is a direct result of approximating the “ L_0 -norm” with a logarithm function (explained and detailed in [5], [6]). This is because, L_1 -norm values do not roll off towards a constant value as input goes to infinity and continue to grow to infinity, while, the L_0 pseudo norm assumes the value of 1 for all values except zero; the logarithm function better approximates this process. Potentially, choices other than logarithm function can be used and the same framework can be employed, only the formula for updating the weights will change [24] (the weights are derived based on mathematical principles and are not heuristic). As mentioned before, each algorithm must make such choices and assumptions on the model-level but such choices are inevitable.

To recapitulate the previous paragraph, FAST-IRES does not have any pre-set limit on source size neither does it use pre-parceling techniques, data-driven or not, to estimate source extent. Iteration numbers, cannot be pre-determined but can be stopped based on solution convergence which basically guarantees that continuing the iterations will not affect source estimates (if the solution does not change from one iteration to the next, the weights will not change, thus it would be the same optimization problem as the step before, and the same solutions will be obtained, as all other parameters are set, throughout the iterations, and weighting is the only parameter that changes from iteration to iteration). Please refer to Fig. R5 for an example (also added to the supplementary materials as Fig. S12). Identity matrices are used for weight initialization (solution and gradient) as no prior knowledge is assumed and consequently do not affect the source estimates negatively.

Figure R5. Effect of Iterations on Estimations. While the iterative re-weighting scheme is key in distinguishing signal from background noise in the FAST-IRES framework, it usually converges within a few iterations. Continuing iterations will not shrink the solution to zero or an overly focused source.

The ICA component selection, and consequently the number of ICs, is based on objective measures, in principle. Additionally, as each ICA component will be a TBF element, the spatial distribution of that TBF element will be solved for that temporal component individually, not affecting the spatial distribution and extent of other components, so missing a component will not affect the spatial estimates of other components. In this sense, the choice of TBF does not affect the extent of the source. It is, however, possible that missing a component might affect the overall estimate of more complex sources such as seizures. That is why our component selection is not arbitrary. We select components that are time-locked to events observed on the scalp, i.e. spike-peak and seizure onset. This automatically removes some noisy components, however, to speed up the process we did visually inspect our components to remove obvious noisy components such as eye blinks, movement and muscle artifacts, etc. which are easy to detect and non-controversial to eliminate, as is common with artifact removal in most EEG pre-processing pipelines. It is possible, in principle, to automate this whole process to minimize the human interaction portion of this process, however, this is a project of its own and not within the scope of our current work. When doubtful if a component is signal-related or not, the component was included to avoid potential loss of signals of interest (potentially introducing more noise into the system). We do acknowledge that this might

affect overall estimation results and have added these discussions and cautionary notes in the manuscript. It is good to note, that cMEM for instance, uses component selection in its singular value decomposition (SVD) step prior to finding seeds for the parcellation and uses an arbitrary, yet reasonable, threshold of 95% to determine the signal subspace, so to speak. We believe, automating the component selection in our method would be a more straightforward process than determining a threshold for SVD component selection (in cMEM, for instance). In any case, we do acknowledge that at some level some choices have to be made which are subjective in nature, the point is at what level this would be and how easy will it be for subsequent parameters, that are affected by such choices, to be tuned robustly and easily.

Another difference that our proposed approach has in comparison to cMEM and 4-ExSo-MUSIC is the application of post-hoc thresholding techniques to clearly determine source extents. It is shown in [15] that cMEM is superior to 4-ExSo-MUSIC in determining the extent size so we will focus on cMEM in this section. FAST-IRES through its iterations converges to an extended solution with no need to threshold the solution explicitly, using data-driven approaches such as Otsu's threshold [23] or subjective thresholding, as opposed to cMEM. While cMEM's is set up, by its reference distribution [15], [25], in a way to switch patches on and off with some probability distribution that is derived from data, it still needs to apply some sort of post-hoc threshold to distinguish baseline and signal activity [15]. This reference distribution has an on/off switch, in a sense, to determine if a cluster, i.e. spatial parcel on the cortex, is activated or not; and when activated, models the activity of that parcel with a Gaussian distribution. This choice, indicates that the activity levels and source amplitudes can change gradually, and while a parcel is switched on, it may contribute minimally to the data fitting and have small amplitudes, hence, a post-hoc thresholding, typically Otsu's method can be used to alleviate the situation. This indicates the need of this algorithm for post-hoc intervention without the algorithm itself being able to directly achieve extent estimation. FAST-IRES does not need such an intervention, so while subjective thresholding is not an issue for either of the algorithms, our algorithm already and automatically takes care of this thresholding process through its iterative re-weighting scheme, as explained previously. Due to such factors, cMEM is not capable of providing statistical thresholding for static source maps [15] (and has to rely on subjective thresholds), while our approach, in fact does. We believe that this is another merit our proposed framework possesses. This does not imply that other methods cannot be used or should not be used in clinical practice, but merely that, our algorithm has the most advanced features in determining source extent without post-hoc intervention. We have clarified these points in the manuscript and supplementary materials.

To address the effect of L-curve on our estimates, we do and have acknowledged that the parameter α affects source estimates as it provides a balance between the two terms of the regularization. However, this parameter can be tuned using the well-established L-curve approach. The L-curve approach is based on a Pareto optimality idea that both terms of the regularization should be as small as possible, basically resulting in the α value corresponding to the knee to be selected. This approach in itself is not subjective. However, it is possible that due to noise or other unpredictable factors, the curve changes a little. This is possible, as it is possible with other techniques such as cross-validation, and has been discussed in the manuscript. We have provided an

example in Fig. R6 showing that even changing the α by, approximately a factor of 10, near the bend of the L-curve does not affect the solution much, indicating that the algorithm is robust against variability in choosing α , as iterations can still fix such variabilities, gradually. As long as these choices are warranted within the model's framework and are theory-driven, and don't affect the subsequent tuning of other parameters in the model, they are not subjective. We believe, that our proposed algorithm is robust to small variations of α and employing the L-curve method can systematically reduce the ranges of α we have to consider; hence it does not seriously affect source estimates. We have discussed these points and added cautionary notes about these choices and differences between theory-driven choices as opposed to more subjective choices in the manuscript and believe that, thanks to the reviewer's comments, we have provided a clearer and more comprehensive view of subjectivity and objectivity of our method and other valuable contributions in the field.

Figure R6. Example of an L-curve and Choice of α on Estimations. The L-curve helps us choose the α parameter that balances between the two terms of the optimization problem, in a systematic manner. While, choice of α matters in determining the source extent, as the example provides, a reasonable range of values are possible and will still yield excellent solutions.

These discussions have been summarized in the Methods and Discussion section of the main body.

This has been reflected in Discussion section as:

“It is apt to be reminded that our proposed approach like any model, is based on a theoretical framework and some model assumptions. For instance, we chose an L_1 optimization framework to formulate the inverse problem while another method might formulate the problem within a Bayesian framework of analysis. These choices are inevitable and are not the kind of “subjectivity” that must be avoided, as they are the product of a model-driven effort. However, based on the choices and assumptions different models make, subsequent steps of the analyses they must perform and the characteristics of the results they obtain, will differ. For instance, while the coherent maximum entropy on the mean (cMEM) algorithm²⁵ produces focally extended sources and successfully employs Otsu’s threshold to distinguish between background activity and signal, it still needs this post-hoc addendum to achieve this goal. This is due to the manner in which this problem is formulated and while determining the source extent is resolved objectively by Otsu’s thresholding algorithm, it shows that every model-driven choice and assumption affects later processes, which might make one algorithm preferable to another.

FAST-IRES has three significant advantages compared to existing approaches. Firstly, it can, in itself and through the process of iterative re-weighting, distinguish signal and background without the need of any post-hoc analysis. Secondly, its hyper-parameters are easy to tune and track. For instance, the hyper-parameter α which balances between sparsity and edge-sparsity of the solution can be tuned by the L-curve approach (refer to supplementary materials for a detailed mathematical treatment of the subject). This easy and intuitive method of tuning a hyper-parameter in itself is a major advantage of our proposed method compared to some more mathematically based assumptions, such as independence of variables in some Bayesian methods or hyper-parameters decided by trial and error or expert opinion (an insight into the L-curve approach and its relation to Pareto optimality is presented elsewhere⁶³). While our proposed method also needs to make such model-driven choices, it still maintains the aforementioned merits compared to other algorithms. Thirdly, our approach does not pre-parcellate the brain^{19,24,25}, in a data-driven or pre-determined manner, to estimate spatial extent. A more detailed discussion of basic ideas of how FAST-IRES operates and what are some of the assumptions it is based on and the hyper-parameters it has to tune are presented in the Methods section under “FAST-IRES Principles, Limitations and Parameters”.

And in a section titled “FAST-IRES Principles, Limitations and Parameters” as,

“The key principles of FAST-IRES can be summarized into 4 main ideas. Firstly, minimizing source edges with an L_1 -norm regularization term. The L_1 -norm of the spatial gradient, which is the difference between the amplitude of two neighboring sources, i.e. the edges, is minimized. This allows the solution to make sudden changes or jumps at limited locations or edges. This is due to the nature of L_1 -norm where sudden changes of amplitude are allowed at limited number of vector elements as opposed to L_2 -norm which enforces smooth changes. The L_1 -norm basically allows a piecewise homogeneous solution as changes in amplitude in the activated region, are discouraged (to minimize edges). Secondly, minimizing the solution’s L_1 -norm also enforces sources to have zero background, discouraging constant backgrounds (a constant background has a zero gradient and is not perceived by the edge sparsity term so it cannot be excluded by the edge-sparsity enforcing term alone). Thirdly, a parameter (α) needs to be defined to balance the two terms of the regularization terms. As mentioned before, an L-curve approach can objectively achieve this goal. Fourthly, a series of iterative re-weightings are performed, in which, the two terms of the regularization function are weighted. These weights are updated at each iteration based on the obtained solution. These weights systematically and without the intervention of an operator, slim down the solution to an extended source that fits the measurements.

The reason why these iterations will not result in an overly focused solution, is because of the edge minimization term. This term prefers extended sources and potentially prefers a constant value so that there are no variations in the solution, but it has to allow changes as the measurements have to be fitted as well. In this manner a balance is reached, and the continuation of iterations will not result in overly focused solutions. The weights also ensure that the solution and its edges are sparse and use solution amplitude (and edge amplitude) to guide the algorithm to converge to an extended solution, more easily.

The number of iterations in our algorithm (for this iterative re-weighting scheme), and most algorithms we are aware of, are difficult to pre-determine, but, the process can continue until solutions converge; that is, the relative change of the solution, normalized by the solution at the previous iteration, does not exceed a pre-set value such as 0.0001 (default value used in our codes). Our experience with simulation and data suggests that within a few iterations, the solution converges and stops changing any further, and continuing the iterations will not shrink the solution making it overly focal; consequently, even if a different tolerance was chosen, e.g. 10^{-6} , the solution would not change much (refer to Fig. S12). Thus, we do not believe, that iteration numbers after the solution converges, affect the extent of the sources much, if any at all.

The weights are initiated, for solution and gradient, as identity matrices, as no a priori knowledge of sources is available. The weights' initialization does not affect the solutions. The choice of weights being reciprocal to the inverse of source amplitude, is not based on heuristics, and is a direct result of approximating the "L₀-norm" with a logarithm function (explained and detailed in ^{63,85}).

To assess the effect of L-curve on FAST-IRES estimates, it must be acknowledged that the parameter α affects source estimates as it provides a balance between the two terms of the regularization. However, this parameter can be tuned using an L-curve approach. The L-curve approach is based on a Pareto optimality idea that both terms of the regularization should be as small as possible, basically selecting the α value corresponding to the knee of the L-curve. This approach in itself is not subjective. However, it is possible that due to noise or other unpredictable factors, the curve changes a little. This is a possibility that cannot be ruled out. We have provided an example in Fig. S13 showing that even changing the α by a factor of 10 near the bend of the L-curve does not affect the solution much, indicating that the algorithm is robust against variability in choosing α , as iterations can still fix such variabilities, gradually. As discussed previously, the presented framework provides an objective measure to select α , and this, in itself, is not problematic. As long as these choices are warranted within the model's framework and are theory-driven, and do not affect the subsequent tuning of other parameters in the model, they are not subjective. We believe, that our proposed algorithm is robust to moderate variations of α and employing the L-curve method can systematically reduce the ranges of α that needs to be considered; hence it does not appear to seriously affect source estimates.

It is obvious, that this approach, like any other, is susceptible to vary with noise (as our simulations also indicate that source extent estimation does not exactly fall on a line, albeit with very high Pearson's correlation values). Changing the solution size and shape a little bit will not affect the regularization terms and may still fit the measurements well enough, so the solution is not unique within a given noise level. Thus, depending on the measurements noise level, we can have errors in estimating the source extent, hence the variability observed in our simulation results and general decline in the algorithm's performance with increasing noise. These are the basic ideas and key parameters of the proposed approach and some of the potential limitations of FAST-IRES. Our algorithm provides a relatively simple (compared to more elaborate Hierarchical Bayesian methods ²⁶, for instance), yet effective framework to estimate underlying sources' extent. "

We have further discussed these issues in the “Independent Component Analysis (ICA) and Component Selection” of Methods section,

“The ICA component selection, and consequently the number of ICs, is based on objective measures, in principle. Additionally, as each ICA component will be a TBF element, the spatial distribution of that TBF element will be solved for that temporal component individually, not affecting the spatial distribution and extent of other components; so, missing a component will not affect the spatial estimates of other components. In this sense, the choice of TBF does not affect the extent of the source. It is, however, possible that missing a component might affect the overall estimate of more complex sources such as seizures. That is why the component selection is not arbitrary. Selected components are time-locked to events observed on the scalp, i.e. spike-peak and seizure onset. This automatically removes some noisy components, however, to speed up the process noisy components such as eye blinks, movement and muscle artifacts, etc. were removed by visual inspection. These components are easy to detect and non-controversial to eliminate, as is common with artifact removal in most EEG pre-processing pipelines (refer to Fig. S14 for an example of denoising effects of ICA analysis). It is possible, in principle, to automate this whole process to minimize the human inter-action portion of this process, however, this is a project of its own and not within the scope of the current work. When doubtful if a component is signal-related or not, the component was included to avoid potential loss of signals of interest (potentially introducing more noise into the system). It is acknowledged that this might affect overall estimation results and as a result ICA must be applied carefully.”

- The author should clarify what was the space of possible spatial extents explored by their method during the iterative approach, what are the key parameters allowing to recover such spatial extent and finally, whether these extents could be in agreement with the large spatial extents involved during epileptic activity, as suggested from intra-cranial recordings.

-- We have partially discussed some of the distinctions of our proposed framework with existing algorithms in response to the reviewer’s previous comment, during which we mentioned that our algorithm does not employ any parcellation or cortical segmentations for estimating the extent of sources. It just uses the basic surface triangulations of the cortical surface that are used in the BEM model to calculate the lead-field matrix and solve the forward problem. Using these surface elements, the neighborhood among cortical surfaces can be determined easily as each triangle has exactly 3 sides and consequently exactly 3 neighbors (the size of surface elements is typically between 1.5-3 mm²). Therefore, we do not have any explicit limit on the size of the underlying source’s extent and any source even as small as a surface element, is theoretically allowed. As our computer simulation shows we simulated sources as small as 0.7 mm (radius) in extent up to 40 mm (radius) in extent with reasonable outcome and our clinical data also suggest that large sources comparable to resection size can be recovered using our proposed approach (referring to figures in the manuscript and Figures R1 and R2 in this response letter).

The key parameters that our algorithm uses to estimate source extent is as follows:

- i. Minimizing source edges with an L1-norm regularization term. We calculate the spatial gradient, by taking the difference between the amplitude of two neighboring sources, i.e. the edges, and minimize its L₁-norm. This allows the solution to make sudden changes or jumps at limited locations or edges

which is due to the nature of L_1 -norm where sudden changes of amplitude are allowed at limited number of vector element as opposed to L_2 -norm which enforces smooth changes. This, in effect, allows a piecewise homogeneous solution as amplitude changes inside the activated region are discouraged (to minimize edges).

- ii. Minimizing the solution's L_1 -norm also enforces sources to have zero background, discouraging constant backgrounds (a constant background has a zero gradient and is not perceived by the edge sparsity term so such constant values cannot be excluded by the edge-sparsity enforcing term, alone).
- iii. A parameter (α) needs to be defined to balance the two terms of the regularization terms. As mentioned before, an L-curve approach can objectively achieve this goal (detailed discussions were presented in response to reviewer's previous comment).
- iv. A series of iterative re-weightings are performed, in which, the two terms of the regularization function are weighted. These weights are updated at each iteration based on the obtained solution (details of how the weights were related to the inverse of source amplitude were presented and discussed in response to reviewer's previous comment). These weights systematically and without the intervention of an operator, slim down the solution to an extended source that fits the measurements.

The reason why these iterations will not make the source overly focused, is because of the edge minimization term. This term prefers extended sources and potentially prefers a constant value so that there are no variations in the solution, but it has to allow changes in amplitude, as the measurements have to be fit as well (a constant value will not fit the measurements well). In this manner a balance is reached, and the continuation of iterations will not result in overly focused solutions (as illustrated in the example in Fig. R5). The weights also ensure that the solution and its edges are sparse by using solution amplitude (and edge amplitude) to move the algorithm to converge to an extended solution, more easily. It is obvious, that this approach, like any other, is susceptible to vary with noise (as our simulations also indicate that source extent estimation does not exactly fall on a line, albeit with very high Pearson's correlation values). Changing the solution size and shape a little bit will not affect the regularization terms and may still fit the measurements well enough, so the solution is not unique (we are still solving an inverse problem). Thus, depending on noise level present in the measurements, we can have errors in estimating the source extent, hence the variability observed in our simulation results and general decline in the algorithm's performance with increasing noise. These are the basic ideas and key parameters of our approach. As discussed before, our algorithm provides a relatively simple (compared to more elaborate Hierarchical Bayesian methods), yet effective framework to estimate underlying sources' extent. We have included this explanation in a section titled "FAST-IRES Principles, Limitations and Parameters" in the Methods to provide a more balanced view of our approach, its strengths and potential shortcomings.

"To assess the effect of L-curve on FAST-IRES estimates, it must be acknowledged that the parameter α affects source estimates as it provides a balance between the two terms of the regularization.

However, this parameter can be tuned using an L-curve approach. The L-curve approach is based on a Pareto optimality idea that both terms of the regularization should be as small as possible, basically selecting the α value corresponding to the knee of the L-curve. This approach in itself is not subjective. However, it is possible that due to noise or other unpredictable factors, the curve changes a little. This is a possibility that cannot be ruled out. We have provided an example in Fig. S13 showing that even changing the α by a factor of 10 near the bend of the L-curve does not affect the solution much, indicating that the algorithm is robust against variability in choosing α , as iterations can still fix such variabilities, gradually. As discussed previously, the presented framework provides an objective measure to select α , and this, in itself, is not problematic. As long as these choices are warranted within the model's framework and are theory-driven, and do not affect the subsequent tuning of other parameters in the model, they are not subjective. We believe, that our proposed algorithm is robust to moderate variations of α and employing the L-curve method can systematically reduce the ranges of α that needs to be considered; hence it does not appear to seriously affect source estimates.

It is obvious, that this approach, like any other, is susceptible to vary with noise (as our simulations also indicate that source extent estimation does not exactly fall on a line, albeit with very high Pearson's correlation values). Changing the solution size and shape a little bit will not affect the regularization terms and may still fit the measurements well enough, so the solution is not unique within a given noise level. Thus, depending on the measurements noise level, we can have errors in estimating the source extent, hence the variability observed in our simulation results and general decline in the algorithm's performance with increasing noise. These are the basic ideas and key parameters of the proposed approach and some of the potential limitations of FAST-IRES. Our algorithm provides a relatively simple (compared to more elaborate Hierarchical Bayesian methods ²⁶, for instance), yet effective framework to estimate underlying sources' extent. ”

- Overall the evaluation of the accuracy of the results is based on spatial overlapping measurements between the results of the proposed source localization technique (spikes, seizure, connectivity during seizure) with the resected area, using validation metrics of precision and recall. It was not clear to me whether the source space was a cortical surface grid or a 3D grid. Indeed if sources are estimated along the cortical surface, then this evaluation with the resection area would require a surface to volume transformation that was not mentioned. Moreover, it was not clearly explained how was the resected region segmented, and co-registered with the pre-surgical MRI. The authors should also discuss whether their evaluation could also be biased by possible co-registration errors, especially if non-linear co-registration was considered between pre-op and post-op MRIs.

-- We thank the reviewer for this comment, and have clarified this in the main body and included an extra figure in the supplementary materials to further clarify this issue (Figure S17). Our solutions are constrained to the cortex and we have projected the resection volume to cortical surface. Basically, in the software Curry, once the post-surgical MRI images of a patient are analyzed, the surgical resection can be segmented out as a set of surface points. This surface is close to the cortical surface, but due to the convoluted nature of the cortex these points are not exactly on the cortex, so these points were projected onto the surface to determine which surface elements, triangular elements, were resected. This procedure is illustrated in Fig. R7.

It is important to note that our image co-registrations do not involve any warping, as we do not project our model to a single head model as is the case in many fMRI

studies. We form individual patient head models and co-register pre-surgical and post-surgical MRI images for each patient individually. The co-registration process is performed in Curry and involves matching the anatomical landmarks of the two images (landmarks such as nasion, preauricular pits, inion, etc.). The transformation is basically a volumetric-based linear transformation maximizing the mutual information between the two registered set of images, that matches these two images together. We have taken extra care to make sure these images match by checking if same anatomical features can be seen at the same slice (Curry allows reviewing co-registered images simultaneously). Co-registration errors are usually within a slice if any (changing one slice at a time, we can see if we are observing the same feature in the two co-registered images, or have we passed a little). Typically the error is in the order of image resolution or about 1-3 mm, so we expect such errors to have minimal effect on our performance measures as resection sizes are at least a couple of centimeters wide. We have added this discussion to the paper as well.

“Estimates were computed on the cortical surface, so surgical surfaces extracted from post-operational MRI had to be projected to the cortical surface for NORs, i.e. precision and recall, to be calculated (Fig. S17 provides two examples).”

Figure R7. Example of Projecting Resection Surface to Cortical Surface. Surgical volume surfaces extracted from post-surgical MRI (co-registered to pre-surgical MRI) were carefully projected to the cortical surface, where our solution space is located, so that subsequent analyses (calculating precision and recall) could be performed.

• When comparing seizure localization to spike localization, the fact that several spike types were averaged (weighted average) even if coming from completely different topographies is seriously biasing spike localization results, which have been shown to perform quite well in other EEG or MEG studies. **In clinical practice, one should really consider the spikes that are of primary importance and this will depend of a global clinical picture.** Similarly, some patients might also exhibit seizures starting from the right or from the left independently. **Did the author record such cases; the duration of the recording should be included for every patient.** Could it be that the recording was not sufficiently long to record seizures from both side, in case of **bilateral temporal epilepsy patients** (if any?). To provide a less biased evaluation of spike versus seizure localization, only spikes showing a scalp topography in agreement with the seizure should be considered in the quantitative analysis (Fig. 6 and statistical tests in suppl. material).

-- We thank the reviewer for bringing this issue up for discussion. It is important to emphasize that we did not average spike topographies before solving the inverse problem. Different spike types were averaged separately, source imaged separately, and their performance, i.e. normalized overlap ratios and localization error, calculated separately; and only after that, were these results averaged. This was done in an attempt to consider the overall effect of spike imaging results so that inconsistent spikes could also be compared to seizure analysis results. We however respectfully disagree with the comment that inconsistent spikes should be ignored. Since inconsistent spikes represent challenging cases for epilepsy source localization, our goal is to provide a solution based on data as much as possible. It is reported that in long EEG sessions, the possibility of recording bilateral inter-ictal discharges is quite high [26]. We had, in the previous version of our manuscript discussed the effect of inconsistent spikes on overall spike performance, and showed that in non-seizure-free cases were inconsistent spikes were observed much more than seizure-free patients, did we observe a difference between seizure and spike analysis, thus, concluding that spike analysis was performing poorly compared to seizure analysis, solely, due to inconsistent spikes. That is why we recommended that if inconsistent spikes are observed in a patient, seizure analysis is to be considered and performed. This is a common practice, that is to refer to seizures, surface EEG or intra-cranial EEG, to confirm diagnosis (certainly other non-electrophysiological clinical information may be considered in clinical practice, but the goal of this work is to provide data-driven epilepsy source localization that may be applied in routine clinical practice). Our analysis was to show that, when only electrophysiological recordings such as spikes and seizures are available, we can perform source imaging to determine the epileptogenic tissue but in case of inconsistent spikes our data suggest that seizure imaging can provide additional benefit. We never claimed that spike analysis is not beneficial or inadequate. As a result, we believe that these spikes cannot and should not be disregarded in our analyses. We have however, provided statistical analysis of comparing seizure results and consistent-spikes and shown that there is no statistically significant difference between the two groups. We have presented the statistics of this test in Table R4 and Figure R8 and also emphasized this point again that spike analysis is not as good as seizure analysis only because of inconsistent spikes and that no difference is seen between seizure analysis and consistent-spikes analysis results.

However, it is necessary to note that defining “consistent” spikes was a post-hoc procedure that needed clinical outcome, while ictal analysis made no use of such information. This further emphasizes the conclusion that we have reported, that inconsistent spikes are reason for further investigation and ictal source imaging, as our data suggests, is a suitable, important and accurate means to this end.

Figure R8. Geometrical Mean Differences of “Consistent” Spike and Ictal Imaging Analysis Results. There is not statistical differences observed between “consistent” spike and seizure imaging results in total, for seizure-free patient group and non-seizure-free patient group. “Consistent” spikes are defined as spikes that are ipsilateral to surgical resection.

Performance Metric		All Patients	ILAE 1-2	ILAE 3-6
Geometric Mean (normalized - 0 to 1)	Seizure	0.70 ± 0.15	0.71 ± 0.16	0.69 ± 0.15
	Spike	0.63 ± 0.33	0.62 ± 0.32	0.64 ± 0.37
Statistical Tests’ Results (P-value)	Rank-sum Test	0.87	0.79	0.53
	T-Test	0.28	0.29	0.73

In response to reviewer’s comment about bilateral and propagating seizures, it must be reported that indeed in 13 of our patients (more than a third of all our patients – 11 of which are temporal lobe cases) seizures either secondarily generalized (after a focal onset) or propagated to the contralateral side a few seconds after seizure onset, which is exactly why we analyzed seizures in the first 0.5-1 second after onset where seizures are still contained near the onset focus. The duration of our seizure recordings is typically between 90-120 seconds and the full duration of seizures are recorded (as patients were under video-EEG monitoring and stayed in the monitoring unit for 5-7 days typically) and analyzed for ICA (will explain in further detail in reviewer’s next comment). As a matter of

fact, even our ICA analysis showed contralateral components which correlated well with the seizure onset, so these components were selected and included for our analysis. It was also seen that if we looked at longer time windows, sometimes as early as 2-3 seconds after seizure onset, the activity would be seen on the contralateral side, as well. Thus, we did not exclude contralateral seizures from our analysis (similar to inter-ictal spikes). In Fig. R9 we can see some of the ICs selected in a patient and the source imaging results of this patient in the beginning of seizure and later during the seizure (note how seizure propagates to the contralateral side).

There are multiple studies suggesting that the existence of multiple spike types might be correlated with unfavorable surgical results [27], [28]. Even in presence of multiple spike types, the existence of a unilateral seizure foci indicated better surgical outcomes [29], [30]. However, if only one type of spike is observed, seizure analysis yields similar conclusion as spike analysis [31]. As a result, we think that our study has, more analytically and quantitatively, verified these results as well.

We have included this table and figure in the supplementary materials and also emphasized this in the Discussion as well to make sure that spike analysis is not discredited totally, as we think it is quite valuable and important for clinical evaluation of epilepsy. We thank the reviewer for bringing this point to our attention so that we could clarify this point thoroughly to avoid any potential confusion.

“To further investigate the difference observed between spike and seizure analysis, we compared “consistent” spikes to seizures in our data base. Consistent spikes were defined as spikes that were ipsilateral to the resection side. Our results indicated that no significant differences could be perceived between consistent spikes and seizures at any level, i.e. total population level, seizure-free cohort or non-seizure-free patients. Results and statistics are presented in Fig. S9 and Table S8. Noting, that this post-hoc analysis was only possible after the availability of clinical results, implying that in practice defining and rejecting inconsistent spikes is a very difficult task or impossible, if further information, such as ictal recordings, are not available.”

And,

“This is possibly why we observed a markedly significant difference between ILAE3-6 patients’ spike and seizure analysis while no such difference was observed among ILAE 1-2 patients in the two groups. Additionally, our post-hoc analysis of consistent spikes also supports this hypothesis as consistent spikes bore no statistically significant difference with ictal imaging results. Thus, it is fair to conclude, that inconsistent spikes are ground for further investigation, and as a result seizure imaging must be performed in these patients to unequivocally determine the affected side and the EZ. This has been observed in the clinical literature; if patients have bitemporal activation, in temporal lobe epilepsy cases for instance, it is more likely to expect unfavorable results and chances of seizure-freedom are reduced significantly^{72,73}. However, our data suggest that if the patient has consistent spikes, spike imaging might provide accurate information about the EZ on its own, without foreseeable benefits from seizure imaging. This indicates that spike imaging is a useful technique for imaging epileptogenic tissue but in cases of multiple inconsistent spike types, other relevant clinical information such as seizures, need to be considered and analyzed carefully.”

Figure R9. An Example of Ictal Propagation in a Temporal Lobe Epilepsy Case. In this example we can clearly see that ICA determined components on both hemispheres (imaged components in source space correspond to the time-course of activity of the presented spatial components, i.e. the TBF elements). As time passes the ictal activity can propagate to the contralateral side as quick as a few seconds after seizure onset.

- How many ICA components were selected for every spike or seizure? Was every seizure analyzed independently using ICA or where the seizure concatenated and at the end how were results combined to provide one seizure localization result for every patient. In line with this idea, even if ICA is indeed well cleaning the signal, at the end the algorithm is performing only one “averaged spike” source localization. Therefore, at the end, authors are considering only one source localization result for every spike type, therefore lacking statistics. **Several authors have proposed the use of single spike source localization, or sub-averaged of clusters of spike, in order to provide statistics over several spike localization.** In this context, the idea of proposing a measure of spike-to-spike reliability in the source space (after single spike source localization) could significantly improve source localization results.

-- We thank the reviewer for bringing this point up for discussion. The number of ICA components for each seizure and spike are presented in Table R5 for reviewer’s information. We did not concatenate seizures for analysis and analyzed seizures independently, calculated our measures for each seizure independently and then averaged the seizure analysis results to achieve one data point for each patient. As explained in the reviewer’s previous comment we also got one single result per patient for spikes, as a result each patient has one data point for seizure and one datapoint for spikes, so no favor is given to seizures or spikes in statistical analysis. As we have explained and presented some results in response to reviewer’s previous comment, we

do not see difference between spike and seizure analysis when consistent-spikes are analyzed so we do not believe that any statistical bias is present in our analysis, we have emphasized this in our current manuscript again to make sure there are no confusions.

We did follow reviewer's suggestion and performed sub-averaging for multiple patients to see if our average spike analysis is biased. We followed Aydin et al. in this analysis [32]. It is important to note that our analysis is not the same as dipole localization analysis, in which small changes in noise could have an effect on the location of the dipole (which is a non-linear inverse algorithm and more susceptible to initialization and local minima). In these methods, the distribution of dipoles due to noise would constitute a distribution of dipoles that would be indicative of the irritative zone. Our method is fundamentally different and provides an extended solution which is not very susceptible to noise (refer to simulation results in Fig. R2), and as such we don't think sub-averaging will affect our analysis much. To this point we present the analysis results of 3 of our patients here. In these patients we sub-averaged the spikes for a total of 1000 times (number of spikes pooled were 3 less than total number of spikes to ensure that 1000 random draws are possible and that the most number of spikes are averaged to achieve high and comparable SNR to averaged spikes). We made sure that each draw was unique, so each selection set (each draw of the 1000 draws) was different from one another. For each of these 1000 sub-averaged spikes we calculated the geometrical mean of precision and recall, i.e. our performance metric, and compared to the same value for the total average-spike. As it can be seen in Fig. R10 and Table R6, our average spike results are well within the sub-average distribution, slightly better than the mean and median, indicating that using the average spike has not biased our results as random draws of the same set of spikes with comparable numbers, do not yield statistically significant results. This means that our average spike result does not underperform and statistically speaking is performing quite well, close to the mean and median of the distribution (indicating that we have no obvious bias in our estimate as bootstrapping, in limit, should provide the actual distribution of a random variable).

We have added a section called "Sub-averaging Approach to Spike Imaging" in the supplementary materials to reflect these results in more detail and also emphasized this in the body of our manuscript to make sure readers are aware of the importance of sub-averaging in spike analysis and that our results are not deviating from the distribution of sub-averaging results.

"In order to ensure that spike imaging analyses were not biased we followed Aydin et al. ¹⁵ in employing a sub-averaging procedure. In conventional dipole fitting studies ^{16,17}, it is observed that due to spike variability, a distribution of fitted dipoles can be achieved, which provides an estimate of the irritative zone, based on the premise that the different observed spikes are arising from the irritative zone, hence the localized dipoles can reveal this underlying zone in the brain.

As the SNR of single spikes can be low and affect the source imaging quality, substantially, Aydin et al. proposed a sub-average procedure that selects a sub-set of spikes randomly (similar to bootstrapping) to improve SNR and yet preserve variability to image the underlying irritative zone using dipole fitting methods.

Our source imaging approach is different from dipole fitting as it already provides an extended solution and is robust against noise. However, we followed a similar approach and randomly drew 1,000 sets of spikes from the set of recorded spikes in 3 patients. These selection sets were unique, no two sets

of the 1,000 draws were the same and were randomly generated by MATLAB. After averaging the spikes for each randomly drawn selection set, the sub-averaged spike was input into the FAST-IRES algorithm and the epileptogenic tissue was estimated for each of the 1,000 draws. The NORs were calculated for each of these 1,000 estimates and the geometric mean was calculated and compared the solution of the averaged spike (all spikes averaged; the procedure followed in this study). The results are presented in Fig. S10 and Table S9. It was observed that the averaged spike results were slightly exceeding the mean and median of this randomly drawn distribution (refer to Fig. S10 and Table S9). Indicating, that average spike analysis is not performing poorly and is not biasing our results.

The average and standard deviation of the estimated solutions for the 1,000 draws are also presented in Fig. S10 for reference. It is observed that the average spike results are selected within the most consistent parts of the solutions, i.e. spatial locations with high average and low standard deviation. Thus, we conclude that sub-averaging proved our results to be unbiased and robust and did not provide additional benefits, within our framework of analysis. ”

And in the Results,

“Additionally, we employed a sub-average technique suggested by Aydin et al.⁶⁰ to ensure that our spike analysis was not biased. The details of implementation and obtained results from this analysis is presented in the supplementary text T4 and results are presented in Fig. S10 and Table S9.”

And in Discussion,

“To ensure that spike imaging results were not biased, we employed sub-averaging routine proposed by Aydin et al.⁶⁰. The intuition behind this approach is to maximize signal SNR by averaging spikes, yet, preserving spike variability by not averaging all the spikes at once and only averaging together randomly drawn sets of spikes (from the total spike population). This simultaneously improves SNR and preserves spike-to-spike variability. In earlier source imaging studies where dipole localization was employed, variability of spikes, resulted in dipoles being localized to different brain regions, hence, resulting in a distribution of dipoles that could reveal the extent of the irritative zone. Our proposed approach is different as it is robust to noise and provides extended solutions which can count for the variability of spikes; moreover, if spikes were different, they would have been categorized as a different spike type and would have been analyzed separately (in our current pipeline). Our analysis also showed that average spike results slightly exceed the mean and median of these random sub-averaged populations (compared the geometric mean of averaged spike to the random sub-averaged population). It is important to emphasize that sub-averaging techniques are important to consider when performing spike analyses, specifically, if spike numbers are limited, but in our proposed framework, no significant improvement was observed. ”

Table R5. Number of TBF components in Each Patient for Seizure and Spike Analysis

Pt. #	No. of Spikes Analyzed	No. of Seizures Analyzed	Number of TBF components for Spike Analysis	Number of TBF components for Seizure Analysis
1	120	3	3	(3, 4, 3)
2	40	2	2	(5, 5)
3	5	3	4	(3, 2, 3)
4	40	3	6	(2, 2, 7)
5	14	3	3	(3, 3, 3)
6	21	2	3	(6, 4)
7	5	3	3	(4, 4, 5)

8	xx	1	xx	4
9	7	3	1	(3, 6, 3)
10	30	3	3	(4, 3, 3)
11	9	2	1	(3, 6)
12	21	1	2	4
13	5	4	2	(4, 3, 4)
14	30	3	5	(3, 5, 4)
15	25	3	4	(5, 6)
16	12	3	3	(4, 5, 5)
17	80	3	6	(6,7,4)
18	15	--	2 spike types – (2, 2)	--
19	20	3	5	(3, 4, 4)
20	29	--	2 spike types – (4, 2, 3)	--
21	39	3	2 spike types – (1, 2)	(6, 2, 4)
22	16	3	2 spike types – (2, 1)	(6, 4, 4)
23	11	3	3	(8, 6, 5)
24	8	3	2 spike types – (1, 1)	(4, 3, 5)
25	21	1	3 spike types – (2, 6, 1)	(4, 4)
26	34	4	2 spike types – (2, 2)	(4, 4, 5, 6)
27	68	1	2 spike types – (3, 2)	2
28	68	2	2 spike types – (3, 3)	(5, 4)
29	29	3	3 spike types – (2, 3, 3)	(4, 5, 4)
30	18	3	2 spike types – (3, 2)	(5, 6, 3)
31	14	4	2 spike types – (1, 2)	(4, 4, 2, 4)
32	8	1	3	2
33	61	3	2 spike types – (1, 2, 1, 4)	(5, 7, 8)
34	16	1	2	8
35	75	1	2 spike types – (3, 2)	4
36	13	2	5	(11, 7)

Figure R10. Sub-averaging Analysis in 3 Examples. The average and standard deviation among the 1000 solutions of the random sub-averaged spikes are presented on the left. As it can be seen on our estimates on the right, our solutions conform to the most consistent part of these distribution, i.e. where average results are high in amplitude and standard deviation is low. The histograms depict the probability distribution of the geometrical means of these 1000 draws (for quantitative results kindly refer to Table R6).

Table R6. Statistics for Sub-averaging Results of Spikes in 3 Patients (Geometrical Mean of Precision and Recall)

Case #	Mean \pm std [Median]	Average Spike Analysis	Worse? ($p < 0.05$)
Case # 1	0.89 \pm 0.03 [0.89]	0.92	No
Case # 2	0.63 \pm 0.18 [0.69]	0.69	No
Case # 3	0.79 \pm 0.16 [0.84]	0.93	No

• The proposed method and results on connectivity are of great interest and quite promising. However I would further clarify and rename these section “Connectivity studies during seizures”, in order to make sure not to confuse with the concept of resting state connectivity analysis, which is quite important in our community. Because the method is completely driven by ICA and components able to extract signal with good SNR, I am not convinced this approach could be used in the context of resting state connectivity, but to study the flow of information during the seizure this is completely fair. However, in this regards, the problem of volume conduction that still exists after source localization, was not addressed (briefly in suppl material) and should be discussed.

-- We thank the reviewer for the favorable comment and agree with the reviewer that the distinction between this type of connectivity analysis and resting state analysis must be emphasized. Due to the reviewer’s comment and reviewer 2 comment and the addition of new analyses to our manuscript, we have currently moved the connectivity analysis to the supplementary materials. If source imaging is done carefully, volume conduction effects can be minimized as shown by many studies [11], [33], [34]. However, it is difficult to claim that it has been removed completely. Thus, based on the reviewer’s recommendation we have emphasized this point in the supplementary section regarding connectivity analysis and we have also mentioned this in the main body. Although, this is not within the scope of this work, but we did, based on the reviewer’s minor comment in the following, compare intra-cranial time-courses and source imaging time-courses in one patient and showed that high correlation (over 60%) can be observed between them, suggesting that source imaging can reduce volume conduction tremendously. We acknowledge that further work is to be done in this regard, and have discussed this finding and its implications in the manuscript and supplementary materials as well.

In the section titled “Connectivity Imaging” in Results section we, now, have:

“FAST-IRES provides a spatio-temporal estimate of underlying brain sources. This implies that in addition to the location and spatial extent of underlying brain activities, the time-course of such activities can be estimated, as well. Subsequent network or connectivity analyses can be performed based on FAST-IRES results. We have outlined and performed a connectivity analysis based on our FAST-IRES results which bears novel ideas of how to potentially implement such analyses in tandem with the proposed framework. The motivation, details and potential limitations of this study is presented in the supplementary text T3 and Fig. S4-S7 and Table S6. Please refer to the supplementary materials for a detailed review and discussion of this proposed analysis.”

And in the supplementary text T2 under “Connectivity Imaging”, we have a sub-title named “A Cautionary Note on Potential Limitations” to discuss the potential limitations of this analysis,

“While source imaging reduces the effect of volume conduction and is shown to delineate underlying brain sources ^{6,7,14}, it is not easy to ascertain that this goal has been achieved. FAST-IRES simulation results indicate that this is a strong possibility. Our initial experiment correlating superficial and deep iEEG recordings with estimated time-courses from FAST-IRES (these findings must be considered with caution as we have explained in the manuscript) indicates that this might be a reasonable assumption for FAST-IRES. However, more work has to be done to reach a clear verdict on this issue. Thus, it is possible that connectivity analyses, DTF analysis, might have been affected by residual effects of volume conduction that might not have been thoroughly eliminated by our proposed inverse approach.

It is also important to note that these connectivity analyses are not to be confused with resting-state connectivity analyses performed in fMRI studies or electrophysiological studies. The term connectivity imaging, in this work, refers to directional connectivity of epilepsy networks using DTF analysis.”

Minor concerns

- There are few typos throughout the manuscript that should be considered.

-- We have carefully revised our manuscript and corrected and re-written multiple sentences and paragraphs. We hope this has fixed the typos the reviewer had noticed before.

- The concept of field of view, or scope, in the context of electrophysiology using invasive and non-invasive techniques is not very clear. It should be clarified or possibly another term could also be proposed to better describe this property.

-- We have replaced this term with spatial coverage which is more often used in this context and creates less confusion.

- For measurement of Localization Error with one channel SOZ, it would be important to clarify what part of the source localization was considered to measure this distance, the main peak, the whole extended area? Were there cases of large intensity source localization results non covered of iEEG implantation?

-- Since our solutions provide an estimate of the extent, and also provides a fairly uniform amplitude over the estimated extent we defined the localization error as the distance between the electrodes and the whole extended area. For most of our patients the SOZ electrode(s) are already within or around the region of maximum amplitude, we have included some examples in Fig. R11 for reviewer’s reference.

If we understand the reviewer’s question, she/he is asking about major sources of activity that may not have been covered by iEEG electrodes; as mentioned to the reviewer’s prior comments we have explained that in 13 patients, seizures propagated either to bilateral side or secondary generalization was observed, which was why we used the early seizure onset portion to determine the epileptogenic brain tissue. However, early after onset, the solutions covered the SOZ implanted electrodes, pretty well, as indicated

by the LE results. We have clarified the definition of the LE in the text and also included these figures in the supplementary materials for further reference.

“The distance is calculated to the boundary of the estimated EZ. The amplitude of estimated solutions by FAST-IRES do not vary much within the activated region, and SOZ electrodes cover (or are close to) regions where solutions’ amplitudes reach maximum (refer to Fig. S18 for a few examples).”

Figure R11. SOZ Electrodes and Estimated Solutions Relative Positioning. In three typical examples we are showing how SOZ electrodes cover our estimated SOZs and are close to the maximum activity (our solution due to the edge-sparsity enforcement in FAST-IRES is quite uniform).

- It could be of interest to illustrate how much ICA is able to reduce the impact of motion and muscle artifact during the seizure, which is a primary importance.

-- We have provided an example, showing how much the ICA analysis can remove noisy artefacts from an actual seizure recorded in Fig. R12, included also in the supplementary materials for reader’s reference.

“These components are easy to detect and non-controversial to eliminate, as is common with artifact removal in most EEG pre-processing pipelines (refer to Fig. S14 for an example of denoising effects of ICA analysis).”

Figure R12. Denoising Effect of ICA. A typical example of how ICA can clean up EEG ictal recordings. Due to movement, eye rolling, eye blinks and many other artefacts, ictal recordings are not easy to process in their raw form and source imaging based on raw data may not be accurate enough, so, a careful pre-processing is recommended.

- For all video, a time scale should be added to the underlying time course to get a better idea of the scale of the signal (even if time samples are mentioned, a temporal scale is quite important). For the video illustrating reconstruction in deep mesial structure, it would be of great interest to compare the time course localized by the methods to real recordings, during typical spikes or seizures (even if these data were not recording simultaneously, it should be possible to find example that are likely to be quite similar). This would be important to judge the quality of the source reconstruction results in deeper structure.

-- We thank the reviewer for this comment. We have added time-bars to our videos per the reviewer's suggestion. We have also looked at the correlation of time-courses between estimated solution from EEG and iEEG recordings for the patient in Video S3 (which the reviewer is referring to). Fig. R13 and Table R7 show the details of our analysis. We looked at the average time-course of activity from SOZ electrodes located deeply in the mesial temporal lobe (near hippocampus) and anterior electrodes. We also

extracted the time-course of activity from cortical areas around these electrodes and compared the traces of these activities together by calculating the correlation between these traces. We repeated this procedure for all the three analyzed seizures (on which seizure imaging was performed) and three seizures selected from the intra-cranial recording. On average a correlational value of over 0.61 was obtained for these traces. The detailed results are presented in Table R7. In Fig. R13 the bold line indicates the mean of time-course activity and the shaded region depicts the standard deviation of the traces among different electrodes and source points. We have included these traces to Video S3 and included Fig. R13 and Table R7 to the supplementary materials. We have also included a cautionary note as to how these results are preliminary due to the fact that simultaneous recordings were not available and that for full validation of these results, simultaneous EEG and iEEG recordings should be analyzed, an aim beyond the scope of the current work. Additionally, two recent publications in Nature Communications [35], [36], estimated the time-course of activity for deep sources (from EEG/MEG) and correlated these findings to find correlation values similar to what we have found here. These studies, benefitted from simultaneous implanted electrodes and scalp recordings, and have shown that under certain conditions, it is possible to estimate the time-course of deep sources with moderate accuracy from scalp measurements. We have also emphasized these recent references in our manuscript:

“Additionally, we calculated the correlation of estimated time-courses of activity, within the lateral portion of the anterior temporal lobe and the deep structures near the hippocampus and para-hippocampal cortex, and compared them to intra-cranial traces recorded in this patient. Our results show a correlational value of 0.61 between estimated time-courses and iEEG recordings in these two regions (Video S3 depicts these results as well). We have further discussed the implications and limitations of this finding in the discussion and presented some more details in the supplementary materials.”

And, in a section titled “Comparing Estimated FAST-IRES Time-courses with Intra-cranial Traces”,

“In order to assess how close our estimated time-courses resemble “true” time courses, i.e. the time-course of activity of underlying sources, an additional analysis was performed in one patient. In this analysis the estimated time-course of activity from FAST-IRES were averaged in the lateral anterior temporal region and the deep para-hippocampal cortex, where intra-cranial electrodes were placed and were ultimately determined as SOZ. These averaged time-courses were correlated to the averaged intra-cranial traces of these two regions and an average correlation of 0.6 was achieved for both regions. This process was repeated for all the 3 seizures that were analyzed in this patient and for every EEG seizure, the extracted time-courses were compared to 3 randomly selected seizures recorded by the iEEG electrodes. The analysis was done in the 1-5 Hz which is the dominant frequency of this patient’s ictal activity. The details of this analyses are presented in Table S13 and Fig. S19, in the supplementary materials, as well as Video S3.

While these results are positive and in line with our simulation results (that underlying sources’ time-courses of activity can be estimated reliably), they are limited by the fact that intra-cranial recordings and EEG recordings were not recorded simultaneously. While these seizures are typical seizures and on average should represent similar spatio-temporal processes, further investigation are necessary to determine the validity of this observation - a work we intend to undertake in our future endeavors.

There is some preliminary evidence that such accurate estimations are possible. In two recent publications ^{5,6}, estimated time-courses of activity from scalp measurements were compared to deep intra-cranial recordings and moderate correlations (comparable to our results) were shown. These two studies benefitted from simultaneous intra-cranial and scalp recordings and are highly suggestive of the possibility to delineate underlying sources’ activity even from deep tissues within the brain.”

Figure R13. Comparing Estimated Time-courses from EEG to Intra-cranial Recordings. In this temporal lobe epilepsy patient, where deep mesial electrodes were implanted, the estimated time-courses from FAST-IRES solution near the SOZ electrodes were averaged and compared to the intra-cranial traces. We looked at anterior regions and deep, near the hippocampus and para-hippocampal cortex regions, separately as the signal looks quite different and found a high correlation between these traces as indicated in the right panel. For the two regions, i.e. deep mesial cortical electrodes near the hippocampus (HC) and the antero-lateral temporal (LAT) surface electrodes, the intra-cranial recorded traces, in blue, are compared to the estimated time-course of activity from FAST-IRES estimates, in red.

Table R7. Statistics for Correlation between Estimated Traces from FAST-IRES and Intra-cranial EEG Traces

	Anterior Temporal Region (Surface)			Para-hippocampal Region (Deep)		
EEG/iEEG	Seizure #1	Seizure #2	Seizure #3	Seizure #1	Seizure #2	Seizure #3
Seizure #1	0.63	0.57	0.64	xx	xx	Xx
Seizure #2	0.68	0.56	0.62	0.57	0.68	0.75
Seizure #3	0.65	0.58	0.59	0.63	0.57	0.6
Mean ± std	0.61 ± 0.04			Mean ± std	0.61 ± 0.07	

Seizure # refers to which seizure results from the EEG recordings were compared to which seizure in the iEEG recording. In EEG Seizure #1, the deep region was not fully recovered so no value for correlation is available (xx).

- On a minor note regarding simulation results in suppl. material, it would have been quite interesting to see the behavior of the method in the presence of realistic noise exhibiting correlations in space and time, as opposed to white noise.

-- We agree with the reviewer and in response to this comment and another comment from reviewer 1, we have included a full new Monte Carlo simulation with realistic noise. The results of this simulation are included in the body of the paper, as shown in Fig. R1 and Fig. R2. The description of these new results in the main body of our manuscript is reflected as follows, in the Results section under “Numerical Simulation Study”:

“In order to thoroughly investigate the performance of our framework we performed a Monte Carlo simulation in which random locations on the cortex were selected and an extended source was simulated at that location, and the EEG signals generated by these source configurations were calculated. A combination of real EEG noise (scalp potential recordings of actual human subjects during rest with no evident epileptic activity) and white Gaussian noise was added to the generated scalp potentials. Multiple noise levels were generated resulting in signal-to-noise ratios (SNRs) of 0, 5, 10, and 20 dB (1,150 simulations were performed for each noise condition). The sources were then estimated using our proposed algorithm and the estimated sources’ extent was plotted against the simulated sources’ extent to show how well source extent can be estimated by our proposed algorithm. Fig. 2 presents the results of our simulation along a typical example of a mesio-temporal source that has been localized in all SNR conditions. The estimated time-course of activity for this source matches the simulated time-course, nicely. A significantly high Pearson’s correlation value of 0.88 was found for these results. More details about the simulation protocol, Pearson’s correlational values and the related statistics, and more examples of estimated sources are presented in the supplementary materials (Table S3-S5, Fig. S1-S3 and Video S1).”

As well as in the supplementary materials text T2:

“In order to provide a thorough evaluation of FAST-IRES a Monte Carlo simulation was performed. Random locations were selected on the cortex as seed points, around which an extended source was created (115 location on the cortex). The extent size of these sources ranged from 0 mm to around 40 mm (each location was randomly assigned an extent). Once the extended sources were placed on a fine cortical model (66,490 triangular surface elements) the forward problem was solved to generate the scalp potentials of these sources and different levels of noise was added to this simulated EEG to obtain SNR levels of 0, 5, 10, and 20 dB. The noise was a combination of realistic correlated EEG noise (from non-epileptic periods of EEG recordings) and additive white Gaussian noise (equal power to realistic EEG noise). For each simulation case at an SNR, 10 noise realizations were analyzed (accruing to a total of 1,150 cases for each SNR level). The inverse was then subsequently calculated for these noisy scalp maps on a coarse cortical grid (31,530 triangular surface elements). The extent of the estimated sources was compared against the simulated sources extent and plotted against each other (Fig. 2 in the main body of this paper). Calculating the Pearson’s correlational value and its related p-value, we found statistically significant and high correlational values, proving that our method robustly estimates underlying sources’ extents (refer to Table S3 for quantitative and more detailed results).

It must be mentioned that as is evident in the results presented in Fig. 2 of the main text, our estimates show the stochastic nature of our algorithm (as in almost any source imaging algorithm), as our estimates do not fall exactly on a straight line. This variation is partly due to noise and partly due to the geometrical complexities of the cortex, specifically for larger sources. Fig. S1 shows some simulation examples.”

- Fig. 5b: there might be a typo in the title which should refer to “Connectivity Imaging Estimates” as opposed to “Spike Imaging Estimate”

-- We thank the reviewer for bringing this error to our attention. We have corrected this typographical error and as discussed before, the connectivity analysis is moved to the supplementary materials, hence, now this figure is referred to as Fig. S6.

References

- [1] A. M. Lascano *et al.*, “Yield of MRI, high-density electric source imaging (HD-ESI), SPECT and PET in epilepsy surgery candidates,” *Clinical Neurophysiology*, vol. 127, no. 1, pp. 150–155, 2016.
- [2] V. Brodbeck *et al.*, “Electroencephalographic source imaging: a prospective study of 152 operated epileptic patients,” *Brain*, vol. 134, no. 10, pp. 2887–2897, 2011.
- [3] A. Sohrabpour, Y. Lu, P. Kankirawatana, J. Blount, H. Kim, and B. He, “Effect of EEG electrode number on epileptic source localization in pediatric patients,” *Clinical Neurophysiology*, vol. 126, no. 3, pp. 472–480, 2015.
- [4] G. Lantz, R. G. de Peralta, L. Spinelli, M. Seeck, and C. M. Michel, “Epileptic source localization with high density EEG: how many electrodes are needed?,” *Clinical neurophysiology*, vol. 114, no. 1, pp. 63–69, 2003.
- [5] E. J. Candès, M. B. Wakin, and S. P. Boyd, “Enhancing sparsity by reweighted ℓ_1 minimization,” *Journal of Fourier analysis and applications*, vol. 14, no. 5–6, pp. 877–905, 2008.
- [6] A. Sohrabpour, Y. Lu, G. Worrell, and B. He, “Imaging brain source extent from EEG/MEG by means of an iteratively reweighted edge sparsity minimization (IRES) strategy,” *NeuroImage*, vol. 142, pp. 27–42, 2016.
- [7] C. Cai, K. Sekihara, and S. S. Nagarajan, “Hierarchical multiscale Bayesian algorithm for robust MEG/EEG source reconstruction,” *NeuroImage*, vol. 183, pp. 698–715, Dec. 2018.
- [8] N. von Ellenrieder, L. Beltrachini, P. Perucca, and J. Gotman, “Size of cortical generators of epileptic interictal events and visibility on scalp EEG,” *NeuroImage*, vol. 94, pp. 47–54, Jul. 2014.
- [9] Z.-L. Lü and S. J. Williamson, “Spatial extent of coherent sensory-evoked cortical activity,” *Exp Brain Res*, vol. 84, no. 2, pp. 411–416, Apr. 1991.
- [10] S. Murakami and Y. Okada, “Invariance in current dipole moment density across brain structures and species: Physiological constraint for neuroimaging,” *NeuroImage*, vol. 111, pp. 49–58, May 2015.
- [11] A. Sohrabpour, S. Ye, G. A. Worrell, W. Zhang, and B. He, “Noninvasive electromagnetic source imaging and granger causality analysis: An electrophysiological Connectome (eConnectome) approach,” *IEEE Transactions on Biomedical Engineering*, vol. 63, no. 12, pp. 2474–2487, 2016.

- [12] F. Babiloni *et al.*, “Estimation of the cortical functional connectivity with the multimodal integration of high-resolution EEG and fMRI data by directed transfer function,” *Neuroimage*, vol. 24, no. 1, pp. 118–131, 2005.
- [13] Y. Lu, L. Yang, G. A. Worrell, and B. He, “Seizure source imaging by means of FINE spatio-temporal dipole localization and directed transfer function in partial epilepsy patients,” *Clinical Neurophysiology*, vol. 123, no. 7, pp. 1275–1283, 2012.
- [14] R. A. Chowdhury, J. M. Lina, E. Kobayashi, and C. Grova, “MEG source localization of spatially extended generators of epileptic activity: comparing entropic and hierarchical Bayesian approaches,” *PLoS One*, vol. 8, no. 2, p. e55969, 2013.
- [15] R. A. Chowdhury *et al.*, “Complex patterns of spatially extended generators of epileptic activity: Comparison of source localization methods cMEM and 4-ExSo-MUSIC on high resolution EEG and MEG data,” *NeuroImage*, vol. 143, pp. 175–195, Dec. 2016.
- [16] G. Birot, L. Albera, F. Wendling, and I. Merlet, “Localization of extended brain sources from EEG/MEG: The ExSo-MUSIC approach,” *NeuroImage*, vol. 56, no. 1, pp. 102–113, May 2011.
- [17] H. Becker *et al.*, “SISSY: An efficient and automatic algorithm for the analysis of EEG sources based on structured sparsity,” *NeuroImage*, vol. 157, pp. 157–172, Aug. 2017.
- [18] A. Bolstad, B. D. Van Veen, and R. Nowak, “Space-time event sparse penalization for magneto-/electroencephalography,” *NeuroImage*, vol. 46, no. 4, pp. 1066–1081, 2009.
- [19] A. Gramfort, D. Strohmeier, J. Haueisen, M. S. Hämäläinen, and M. Kowalski, “Time-frequency mixed-norm estimates: Sparse M/EEG imaging with non-stationary source activations,” *NeuroImage*, vol. 70, pp. 410–422, 2013.
- [20] S. Haufe, V. V. Nikulin, A. Ziehe, K.-R. Müller, and G. Nolte, “Combining sparsity and rotational invariance in EEG/MEG source reconstruction,” *NeuroImage*, vol. 42, no. 2, pp. 726–738, 2008.
- [21] T. Limpiti, B. D. Van Veen, and R. T. Wakai, “Cortical patch basis model for spatially extended neural activity,” *IEEE Transactions on Biomedical Engineering*, vol. 53, no. 9, pp. 1740–1754, 2006.
- [22] M. Zhu, W. Zhang, D. L. Dickens, and L. Ding, “Reconstructing spatially extended brain sources via enforcing multiple transform sparseness,” *NeuroImage*, vol. 86, pp. 280–293, 2014.
- [23] N. Otsu, “A threshold selection method from gray-level histograms,” *IEEE transactions on systems, man, and cybernetics*, vol. 9, no. 1, pp. 62–66, 1979.
- [24] D. Wipf and S. Nagarajan, “Iterative Reweighted ℓ_1 and ℓ_2 Methods for Finding Sparse Solutions,” *IEEE Journal of Selected Topics in Signal Processing*, vol. 4, no. 2, pp. 317–329, 2010.
- [25] C. Grova, J. Daunizeau, J.-M. Lina, C. G. Bénar, H. Benali, and J. Gotman, “Evaluation of EEG localization methods using realistic simulations of interictal spikes,” *Neuroimage*, vol. 29, no. 3, pp. 734–753, 2006.
- [26] E. Ergene, J. J. Shih, D. E. Blum, and N. K. So, “Frequency of Bitemporal Independent Interictal Epileptiform Discharges in Temporal Lobe Epilepsy,” *Epilepsia*, vol. 41, no. 2, pp. 213–218, 2000.
- [27] M. Y. Chung, T. S. Walczak, D. V. Lewis, D. V. Dawson, and R. Radtke, “Temporal Lobectomy and Independent Bitemporal Interictal Activity: What Degree of Lateralization Is Sufficient?,” *Epilepsia*, vol. 32, no. 2, pp. 195–201, Apr. 1991.
- [28] A. Hufnagel, M. Dümpelmann, J. Zentner, O. Schijns, and C. E. Elger, “Clinical Relevance of Quantified Intracranial Interictal Spike Activity in Presurgical Evaluation of Epilepsy,” *Epilepsia*, vol. 41, no. 4, pp. 467–478, Apr. 2000.
- [29] Y. Aghakhani, X. Liu, N. Jette, and S. Wiebe, “Epilepsy surgery in patients with bilateral temporal lobe seizures: A systematic review,” *Epilepsia*, vol. 55, no. 12, pp. 1892–1901, Dec. 2014.

- [30] A. Hufnagel *et al.*, “Prognostic Significance of Ictal and Interictal Epileptiform Activity in Temporal Lobe Epilepsy,” *Epilepsia*, vol. 35, no. 6, pp. 1146–1153, Nov. 1994.
- [31] C. Vollmar, I. Stredl, M. Heinig, S. Noachtar, and J. Rémi, “Unilateral temporal interictal epileptiform discharges correctly predict the epileptogenic zone in lesional temporal lobe epilepsy,” *Epilepsia*, vol. 0, no. 0.
- [32] Ü. Aydin *et al.*, “Combined EEG/MEG Can Outperform Single Modality EEG or MEG Source Reconstruction in Presurgical Epilepsy Diagnosis,” *PLOS ONE*, vol. 10, no. 3, p. e0118753, Mar. 2015.
- [33] L. Astolfi *et al.*, “Assessing cortical functional connectivity by linear inverse estimation and directed transfer function: simulations and application to real data,” *Clinical neurophysiology*, vol. 116, no. 4, pp. 920–932, 2005.
- [34] H. B. Hui, D. Pantazis, S. L. Bressler, and R. M. Leahy, “Identifying true cortical interactions in MEG using the nulling beamformer,” *NeuroImage*, vol. 49, no. 4, pp. 3161–3174, 2010.
- [35] M. Seeber, L.-M. Cantonas, M. Hoevels, T. Sesia, V. Visser-Vandewalle, and C. M. Michel, “Subcortical electrophysiological activity is detectable with high-density EEG source imaging,” *Nature Communications*, vol. 10, no. 1, p. 753, Feb. 2019.
- [36] F. Pizzo *et al.*, “Deep brain activities can be detected with magnetoencephalography,” *Nature Communications*, vol. 10, no. 1, p. 971, Feb. 2019.

Reviewers' Comments:

Reviewer #1:

Remarks to the Author:

That was a long response to reviewers! I mainly requested that estimated and "true" values of spatial extent for empirical data were presented. The correlation isn't very high, but good enough to justify publication.

Reviewer #2:

Remarks to the Author:

The authors have done an excellent job addressing my comments/suggestions, as well as those by the other two reviewers. The revised manuscript is significantly improved in terms of presentation, scope, clarity and placement of the contributions in context. I appreciate the authors' sincere and laborious undertaking in revising the manuscript, and have no further comments.

Reviewer #3:

None

Response to Reviewers

We sincerely thank the reviewers for their critical and constructive comments. We believe that the manuscript has significantly improved and strengthened because of reviewers' comments. We also thank the reviewers for their favorable comments, and we are glad to see that the reviewers' comments/concerns have been thoroughly addressed.

Reviewer #1 (Remarks to the Author):

That was a long response to reviewers! I mainly requested that estimated and "true" values of spatial extent for empirical data were presented. The correlation isn't very high, but good enough to justify publication.

-- We thank the reviewer for her/his favorable comment. We are glad that the reviewer is satisfied.

Reviewer #2 (Remarks to the Author):

The authors have done an excellent job addressing my comments/suggestions, as well as those by the other two reviewers. The revised manuscript is significantly improved in terms of presentation, scope, clarity and placement of the contributions in context. I appreciate the authors' sincere and laborious undertaking in revising the manuscript, and have no further comments.

-- We thank the reviewer for her/his favorable comment. We are glad to see that your comments have been addressed.